# REPRESENTATION AND BIAS IN MULTILINGUAL NLP: INSIGHTS FROM CONTROLLED EXPERIMENTS ON CONDITIONAL LANGUAGE MODELING

## ABSTRACT

Inspired by the phenomenon of performance disparity between languages in machine translation, we investigate whether and to what extent languages are equally hard to "conditional-language-model". Our goal is to improve our understanding and expectation of the relationship between language, data representation, size, and performance. We study one-to-one, bilingual conditional language modeling through a series of systematically controlled experiments with the Transformer and the 6 languages from the United Nations Parallel Corpus. We examine character, byte, and word models in 30 language directions and 5 data sizes, and observe indications suggesting a script bias on the character level, a length bias on the byte level, and a word bias that gives rise to a hierarchy in performance across languages. We also identify two types of sample-wise non-monotonicity — while word-based representations are prone to exhibit Double Descent, length can induce unstable performance across the size range studied in a novel meta phenomenon which we term *erraticity*. By eliminating statistically significant performance disparity on the character and byte levels by normalizing length and vocabulary in the data, we show that, in the context of computing with the Transformer, there is no complexity intrinsic to languages other than that related to their statistical attributes and that performance disparity is not a necessary condition but a byproduct of word segmentation. Our application of statistical comparisons as a fairness measure also serves as a novel rigorous method for the intrinsic evaluation of languages, resolving a decades-long debate on language complexity. While all these quantitative biases leading to disparity are mitigable through a shallower network, we find room for a human bias to be reflected upon. We hope our work helps open up new directions in the area of language and computing that would be fairer and more flexible and foster a new transdisciplinary perspective for DL-inspired scientific progress.

## 1 INTRODUCTION

With a transdisciplinary approach to explore a space at the intersection of Deep Learning (DL) / Neural Networks (NNs), language sciences, and language engineering, we report our undertaking in **use-inspired basic research** — with an application-related phenomenon as inspiration, we seek **fundamental scientific understanding** through empirical experimentation. This is *not* an application or machine translation (MT) paper, but one that strives to evaluate and seek new insights on language in the context of DL with a consideration to contribute to our evaluation, segmentation, and model interpretation practice in multilingual Natural Language Processing (NLP).

**Our *inspiration*: performance disparity in MT**   The use case that inspired our investigation is the disparity of MT results reported in Junczys-Dowmunt et al. (2016). Of the 6 official languages of the United Nations (UN) — Arabic (AR), English (EN), Spanish (ES), French (FR), Russian (RU), and Chinese (ZH), results with target languages AR, RU, and ZH seem to be worse than those with EN/ES/FR, regardless of the algorithm, may it be from phrased-based Statistical MT (SMT/Moses

(Koehn et al., 2007)) or Neural MT (NMT).[1] The languages have the same amount of line-aligned, high-quality parallel data available for training, evaluation, and testing. This prompts the question: are some languages indeed harder to translate from or to?

**Problem statement: are all languages equally hard to Conditional-Language-Model (CLM)?** A similar question concerning (monolingual) language modeling (LMing) was posed in Cotterell et al. (2018) and Mielke et al. (2019) along with the introduction of a method to evaluate LMs with multiway parallel corpora (multitexts) in information-theoretic terms. To explicitly focus on modeling the complexities that may or may not be *intrinsic* to the languages, we study the more fundamental process of CLMing without performing any translation. This allows us to eliminate confounds associated with generation and other evaluation metrics. One could think of our effort as estimating conditional probabilities with the Transformer, with a bilingual setup where perplexity of one target language ($l_{\text{trg}}$) is estimated given the parallel data in one source language ($l_{\text{src}}$), where $l_{\text{src}} \neq l_{\text{trg}}$. We focus on the very basics and examine the first step in our pipeline — input representation, holding everything else constant. Instead of measuring absolute cross-entropy scores, we evaluate the relative differences between languages from across 5 magnitudes of data sizes in 3 different representation types/levels. **We consider *bias* to be present when performance disparity in our Transformer models is statistically significant.**

## 1.1 SUMMARY OF FINDINGS AND CONTRIBUTIONS

In investigating performance disparity as a function of size and data with respect to language and representation on the Transformer in the context of CLMing, we find:

1. in a bilingual (one-to-one) CLMing setup, there is **neutralization of source language instances**, i.e. there are no statistically significant differences between source language pairs. Only pairs of target languages differ significantly (see Table 1).
2. We identify 2 types of **sample-wise non-monotonicity** on each of the primary representation levels we studied:
   (a) Double Descent (Belkin et al., 2019; Nakkiran et al., 2020): on the word level, for all languages, performance at $10^2$ lines is typically better than at $10^3$ before it improves again at $10^4$ and beyond. This phenomenon can also be observed in character models with ZH as a target language as well as on the word level with non-neural n-gram LMs;
   (b) *erraticity*: performance is irregular and exhibits great variance across runs. We find sequence length to be predictive of this phenomenon. We show that this can be rectified by data transformation or hyperparameter tuning. In our study, erraticity affects AR and RU on the byte level where the sequences are too long with UTF-8 encoding and ZH when decomposed into strokes on the character level.
3. In eliminating performance disparity through lossless data transformation on the character and byte levels, we resolve language complexity (§ 4 and App. J). We show that, in the context of computing with the Transformer, unless word-based methods are used, there is no linguistic/morphological complexity applicable or necessary. There is **no complexity that is intrinsic to a language aside from its statistical properties**. Hardness in modeling is relative to and bounded by its representation level (**representation relativity**). On the character and byte levels, hardness is correlated with statistical properties concerning sequence length and vocabulary of a language, irrespective of its linguistic typological, phylogenetic, historical, or geographical profile, and can be eliminated. On the word level, hardness is correlated with vocabulary, and a complexity hierarchy arises through the manual preprocessing step of word tokenization. This complexity/disparity effected by word segmentation cannot be eliminated due to the fundamental qualitative differences in the definition of a "word" being one that neither holds universally nor is suitable/consistent for fair crosslinguistic comparisons. We find clarification of this expectation of disparity necessary because more diligent error analyses need to be afforded instead of simply accepting massively disparate results or inappropriately attributing under-performance to linguistic reasons.
4. Representational units of **finer granularity** can help close the gap in performance disparity.
5. Bigger/overparameterized models can **magnify/exacerbate the effects of differences in data statistics**. Quantitative biases that lead to disparity are mitigable through numerical methods.

---

[1]We provide a re-visualization of these grouped in 6 facets by target language in Figure 4 in Appendix A.

**Outline of the paper** In § 2, we define our method and experimental setup. We present our results and analyses on the primary representations in § 3 and those from secondary set of controls in § 4 in a progressive manner to ease understanding. Meta analyses on fairness evaluation, non-monotonic behavior, and discussion on biases are in § 5. Additional related work is in § 6. We refer our readers to the Appendices for more detailed descriptions/discussions and reports on supplementary experiments.

## 2 METHOD AND DEFINITIONS

**Controlled experiments as basic research for scientific understanding** Using the United Nations Parallel Corpus (Ziemski et al., 2016), the data from which the MT results in Junczys-Dowmunt et al. (2016) stem, we perform a series of controlled experiments on the Transformer, holding the hyperparameter settings for all 30 one-to-one language directions from the 6 languages constant. We control for size (from $10^2$ to $10^6$ lines) and language with respect to representational granularity. We examine 3 primary representation types — character, byte (UTF-8), and word, and upon encountering some unusual phenomena, we perform a secondary set of controls with 5 alternate representations — on the character level: Pinyin and Wubi (ASCII representations for ZH phones and character strokes, respectively), on the byte level: code page 1256 (for AR) and code page 1251 (for RU), and on the word level: Byte Pair Encoding (BPE) (Sennrich et al., 2016), an adapted compression algorithm from Gage (1994). These symbolic variants allow us to manipulate the statistical properties of the representations, while staying as "faithful" to the language as possible. We adopt this symbolic data-centric approach because we would like to more directly interpret the confounds, if any, that make language data different from other data types. We operate on a smaller data size range as this is more common in traditional domain sciences and one of our higher goals is to bridge an understanding between language sciences and engineering (the latter being the dominant focus in NLP). We run statistical tests to identify the strongest correlates of performance and to assess whether the differences between the mean performance of different groups are indeed significant. **We are concerned *not* with the absolute scores, but with the *relations* between scores from different languages and the generalizations derived therefrom.**

**Information-theoretic, fair evaluation with multitexts** Most sequence-to-sequence models are optimized using a cross-entropy loss (see Appendix B for definition). Cotterell et al. (2018) propose to use "renormalized" perplexity (PP) to evaluate LMs fairly using the total number of bits divided by some constant. In our case, we choose instead a simpler method of using an "unnormalized" PP, directly using the total number of bits needed to encode the development (dev) set, which has a constant size of 3,077 lines per language.

**Disparity/Inequality** In the context of our CLMing experiments, we consider there to be "disparity" or "inequality" between languages $l_1$ and $l_2$ if there are significant differences between the performance distributions of these two languages with respect to each representation. Here, by performance we mean the number of bits required to encode the held-out data using a trained CLM. With 30 directions, there are 15 pairs of source languages $(l_{src1}, l_{src2})$ and 15 pairs of target languages $(l_{trg1}, l_{trg2})$ possible. To assess whether the differences are significant, we perform unpaired two-sided significance tests with the null hypothesis that the score distributions for the two languages are not different. Upon testing for normality with the Shapiro-Wilk test (Shapiro & Wilk, 1965; Royston, 1995), we use the parametric unpaired two-sample Welch's t-test (Welch, 1947) (when normal) or the non-parametric unpaired Wilcoxon test (Wilcoxon, 1945) (when not normal) for the comparisons. We use the implementation in R (R Core Team, 2014) for these 3 tests. To account for the multiple comparisons we are performing, we correct all p-values using Bonferroni's correction (Benjamini & Heller, 2008; Dror et al., 2017) and follow Holm's procedure[2] (Holm, 1979; Dror et al., 2017) to identify the pairs of $l_1$ and $l_2$ with significant differences after correction. We report all 3 levels of significance ($\alpha \leq 0.05, 0.01, 0.001$) for a more comprehensive evaluation.

**Experimental setup** The systematic, identical treatment we give to our data is described as follows with further preprocessing and hyperparameter details in Appendices B and C, respectively. The distinctive point of our experiment is that the training regime is the same for all (intuition in App. O.1).

---

[2]using implementation from `https://github.com/rtmdrr/replicability-analysis-NLP`

After filtering length to 300 characters maximum per line in parallel for the 6 languages, we made 3 subsets of the data with 1 million lines each — one having lines in the order of the original corpus (dataset A) and two other randomly sampled (without replacement) from the full corpus (datasets B & C). Lines in all datasets are extracted in parallel and remain fully aligned for the 6 languages. For each run and each representation, there are 30 pairwise directions (i.e. one $l_{\text{src}}$ to one $l_{\text{trg}}$) that result from the 6 languages. We trained all 150 (for 5 sizes) 6-layer Transformer models for each run using the SOCKEYE Toolkit (Hieber et al., 2018). We optimize using PP and use early stopping if no PP improvement occurs after 3 checkpoints up to 50 epochs maximum, taking the best checkpoint. Characters and bytes are supposed to mitigate the out-of-vocabulary (OOV) problem on the word level. In order to assess the effect of modeling with finer granularity more precisely, all vocabulary items appearing once in the train set are accounted for (i.e. full vocabulary on train, as in Gerz et al. (2018a;b)). But we allow our system to categorize all unknown items in the dev set to be unknown (UNK) so to measure OOVs (open vocabulary on dev (Jurafsky & Martin, 2009)). To identify correlates of performance, we perform Spearman's correlation (Spearman, 1904) with some basic statistical properties of the data (e.g. length, vocabulary size ($|V|$), type-token-ratio, OOV rate) as metrics — a complete list thereof is provided in Appendix F. For each of the 3 primary representations — character, byte, and word, we performed 5 runs total in 5 sizes ($10^2$-$10^6$ lines) (runs A0, B0, C0, A1, & A2) and 7 more runs in 4 sizes ($10^2$-$10^5$ lines) (A3-7, B1, & C1), also controlling for seeds. For the alternate/secondary representations, we ran 3 runs each in 5 sizes ($10^2$-$10^6$ lines) (A0, B0, & C0).

## 3   EXPERIMENTAL RESULTS OF PRIMARY REPRESENTATIONS

Subfigures 1a, 1b, and 1c present the mean results across 12 runs of the 3 primary representations — character, byte, and word, respectively. The x-axis represents data size in number of lines and y-axis the total conditional cross-entropy, measured in bits (Eq. 1 in Appendix B). Each line connects 5 data points corresponding to the number of bits the CLMs (trained with training data of $10^2$, $10^3$, $10^4$, $10^5$, and $10^6$ lines) need to encode the target language dev set given the corresponding text in the source language. These are the same data in the same 30 language directions and 5 sizes with the same training regime, just preprocessed/segmented differently. This confirms **representation relativity** — languages (or any objects being modeled) need to be evaluated relative to their representation. "One size does not fit all" (Durrani et al., 2019), our conventional way of referring to "language" (as a socio-cultural product or with traditional word-based approaches, or even for most multilingual tasks and competitions) is too coarse-grained (see also Fisch et al. (2019) and Ponti et al. (2020)).

Subfigures 1d, 1e, and 1f display the corresponding information sorted into facets by target language, source languages represented as line types. Through these we see more clearly that results can be grouped rather neatly by target language (cf. figures sorted by source language in Appendix H) — as implicit in the Transformer's architecture, the decoder is unaware of the source language in the encoder. As shown in Table 1 in § 5 summarizing the number of source and target language pairs with significant differences, there are **no significant differences across any source language pairs**. The Transformer neutralizes source language instances. This could explain why transfer learning or multilingual/zero-shot translation (Johnson et al., 2017) is possible at all on a conceptual level.

In general, for character and byte models, most language directions do seem to converge at $10^4$ lines to similar values across all target languages, with few notable exceptions. There are some fluctuations past $10^4$, indicating further tuning of hyperparameters would be beneficial due to our present setting possibly working most favorably at $10^4$. On the character level, target language ZH ($\text{ZH}_{trg}$) shows a different learning pattern throughout. And on the byte level, $\text{AR}_{trg}$ and $\text{RU}_{trg}$ display non-monotonic and unstable behavior, which we refer to as *erratic*. Word models exhibit Double Descent across the board (note the spike at $10^3$), but overall, difficult/easy languages stay consistent, with AR and RU being the hardest, followed by ES and FR, then EN and ZH. A practical takeaway from this set of experiments: in order to obtain more robust training results, use bytes for ZH (as suggested in Li et al. (2019a)) and characters for AR and RU (e.g. Lee et al. (2017)) — also if one wanted to avoid any "class" problems in performance disparity with words. Performance disparity for these representations is reported in Table 1 under "CHAR", "BYTE", and "WORD". Do note, however, that the intrinsic performance of ZH with word segmentation is not particularly subpar. But this often does not correlate with its poorer downstream tasks results (recall results from Junczys-Dowmunt et al. (2016)). Since the notion of word in ZH is highly contested and

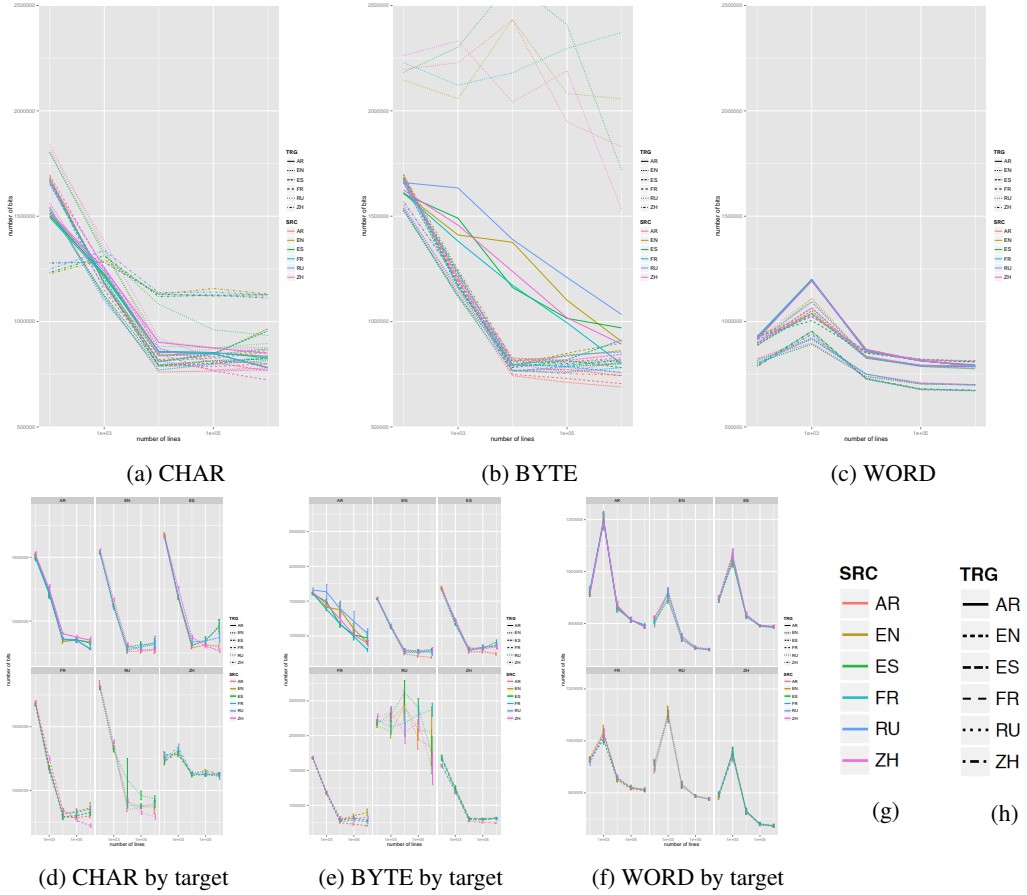

(a) CHAR
(b) BYTE
(c) WORD

(d) CHAR by target
(e) BYTE by target
(f) WORD by target

(g)
(h)

Figure 1: Number of bits (the lower the better) as a function of data size plotted for all 30 directions. Subfigures 1d, 1e, and 1f depict the corresponding information as in 1a, 1b, and 1c (showing mean across 12 runs), respectively, but sorted in 6 facets by target language and with error bars. Legend in Subfigure 1g shows the correspondence between colors and source languages, in Subfigure 1h between line types and target languages. (These figures are also shown enlarged in Appendix G.)

ambiguous — 1) it is often aimed to align with that in other languages so to accommodate manual feature engineering and academic theories, 2) there is great variation among different conventions, 3) native ZH speakers identify characters as words — there are reasons to rethink this procedure now that fairer and language-independent processing in finer granularity is possible (cf. Li et al. (2019b) as well as Duanmu (2017) for a summary on the contested nature of wordhood in ZH). A more native analysis of ZH, despite being considered a high-resource language, has not yet been recognized in NLP.

## 4 UNDERSTANDING THE PHENOMENA WITH ALTERNATE REPRESENTATIONS

To understand why some languages show different results than others, we carried out a secondary set of control experiments with representations targeting the problematic statistical properties of the corresponding target languages. (An extended version of this section is provided in Appendix P.)

**Character level** We reduced the high $|V|$ in ZH with representations in ASCII characters — Pinyin and Wubi. The former is a romanization of ZH characters based on their pronunciations and the latter an input algorithm that decomposes character-internal information into stroke shape and ordering and matches these to 5 classes of radicals (Lunde, 2008). We replaced the ZH data in these formats *only on the target side* and reran the experiments involving $ZH_{trg}$ on the character level. Results in Figure 2 and Table 1 show that the elimination of disparity on character level is possible if ZH is represented

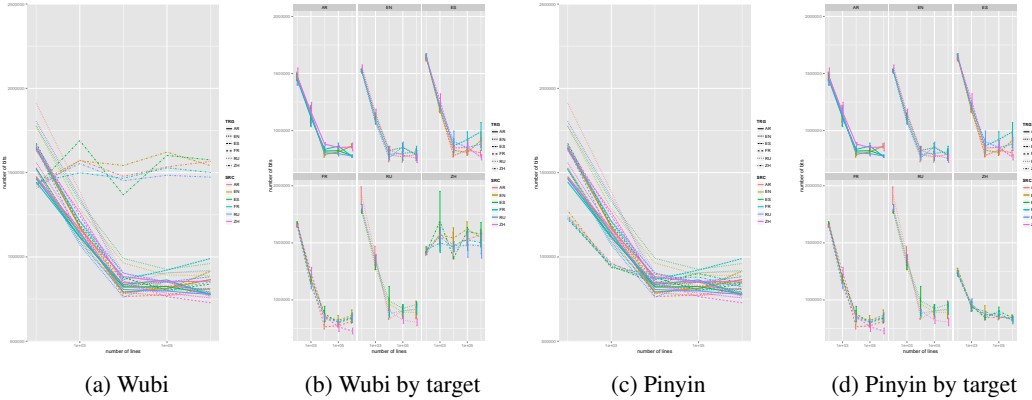

(a) Wubi     (b) Wubi by target     (c) Pinyin     (d) Pinyin by target

Figure 2: Character-level remedies for ZH: Wubi vs. Pinyin.

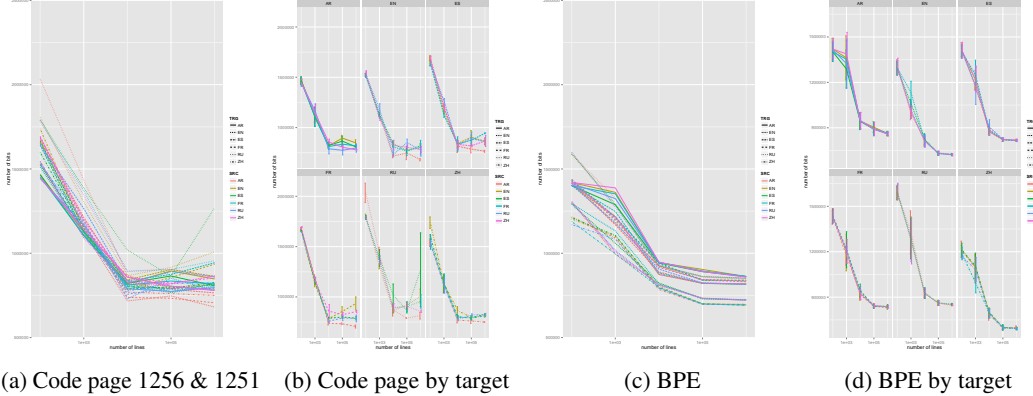

(a) Code page 1256 & 1251    (b) Code page by target    (c) BPE    (d) BPE by target

Figure 3: Byte-level (Subfigures 3a & 3b) remedies with code page 1256 for target AR and 1251 for target RU, and word-level (Subfigures 3c & 3d) remedy with BPE for all languages.

through Pinyin (transliteration), as in Subfigure 2c. But models with ZH logographic scripts display a behaviorial tendency unlike those with other (phonetic) alphabetic scripts (Subfigure 2a). Work published thus far using Wubi with the Transformer seems to have needed some form of architectural modification (Gao et al., 2020) or a different architecture altogether (Nikolov et al., 2018; Zhang et al., 2019), suggesting a possible script bias (to be further discussed in § 5 under "Basis for biases").

**Byte level** Length is the most salient statistical attribute that makes AR and RU outliers. To shorten their sequence lengths, we tested with alternate encodings on $AR_{trg}$ and $RU_{trg}$ — code page 1256 and 1251, which provide 1-byte encodings specific to AR and RU, respectively. Results are shown in Subfigures 3a and 3b. Not only is erraticity resolved, the number of 15 possible target language pairs with significant differences reduces from **8** with the UTF-8 byte representation to **0** (Table 1 under "$ARRU_t$"), indicating that we eliminated disparity with this optimization heuristic. Since our heuristic is a lossless and reversible transform, it shows that **a complexity that is intrinsic and necessary in language**[3] **does not exist** in computing, however diverse they may be, as our 6 are, from the conventional linguistic typological, phylogenetic, historical, or geographical perspectives. Please refer to Appendix J for our discussion on language complexity.

**Word level** The main difference between word and character/byte models is length not being a top contributing factor correlating with performance, but instead $|V|$ is. This is understandable as word segmentation neutralizes sequence lengths. To remedy the OOV problem, we use BPE, which learns a fixed vocabulary of variable-length character sequences (on word level, as it presupposes word

---

[3]aside from its statistical properties related to length and vocabulary. "Language" here refers to language represented through all representations.

Table 1: Number of language pairs out of 15 with significant differences, with respective p-values. $ARRU_t$ refers to AR & RU being optimized only on the target side; whereas $ARRU_{s,t}$ denotes optimization on both source and target sides (relevant for directions AR-RU and RU-AR).

| | p-value | CHAR | | Pinyin | | Wubi | | BYTE | | $ARRU_t$ | | $ARRU_{s,t}$ | | WORD | | BPE | |
|---|---|---|---|---|---|---|---|---|---|---|---|---|---|---|---|---|---|
| | | src | trg | src | trg | src | trg | src | trg | src | trg | src | trg | src | trg | src | trg |
| | 0.05 | 0 | 7 | 0 | 4 | 0 | 8 | 0 | 9 | 0 | 4 | 0 | 4 | 0 | 11 | 0 | 10 |
| | 0.01 | 0 | 5 | 0 | 2 | 0 | 6 | 0 | 8 | 0 | 3 | 0 | 4 | 0 | 8 | 0 | 8 |
| ☞ | 0.001 | 0 | 3 | **0** | **0** | 0 | 5 | 0 | 8 | **0** | **0** | 0 | 2 | 0 | 8 | 0 | 7 |

segmentation) from the training data. It is more fine-grained than word segmentation and is known for its capability to model subword units for morphologically complex languages (e.g. AR and RU). We use the same vocabulary of 30,000 as specified in Junczys-Dowmunt et al. (2016). This reduced our averaged OOV token rate by 89-100% across the 5 sizes. The number of language pairs with significant differences reduced to 7 from 8 for word models, showing how **finer-grained modeling has a positive effect on closing the disparity gap**.

## 5  META-RESULTS, ANALYSIS, AND DISCUSSION

**Performance disparity**    Table 1 lists the number of language pairs with significant differences under the representations studied. Considering how it is **possible** for our character and byte models to effect no performance disparity for the same languages on the same data, this indicates that disparity is not a necessary condition. In fact, the customary expectation that languages ought to perform differently stems from our word segmentation practice. Furthermore, the order of AR/RU > ES/FR > EN/ZH (Figure 1c) resembles the idea of morphological complexity. Considering there are character-internal meaningful units in languages with logographic script such as ZH (cf. Zhang & Komachi (2018)) that are rarely captured, studied, or referred to as "morphemes", this goes to show that linguistic morphology, along with its complexity, as is practiced today[4] and that which has occurred in the NLP discourse thus far, has only been relevant on and is bounded to the "word" level. The definition of word, however, has been recognized as problematic for a very long time in the language sciences (see Haspelmath (2011) and references therein from the past century). Since the conventional notion of word, which has been centered on English and languages with alphabetic scripts, has a negative impact on languages both morphologically rich (see Minkov et al. (2007), Seddah et al. (2010), inter alia), AR and RU in our case, as well as morphologically "frugal" (Koehn, 2005), as in ZH, finer-grained modeling with characters and bytes (or n-gram variants/pieces thereof) is indeed a more sensible option and enables a greater variety of languages to be handled with more simplicity, fairness, independence, and flexibility.

While the lack of significant differences between pairs of source languages would signify neutralization of source language instances, it does not mean that source languages have no effect on target. For our byte solutions with code pages, we experimented also with source side optimization in the directions that involve AR/RU as source. This affected the distribution of the disparity results for that representation — with 2 pairs being significantly different (see Table 1 under "$ARRU_{s,t}$"). We defer further investigation on the nature of source language neutralization to future work.

**Sample-wise Double Descent (DD)**    Sample-wise non-monotonicity/DD (Nakkiran et al., 2020) denotes a degradation followed by an improvement in performance with increasing data size. We notice word models and character models with $ZH_{trg}$, i.e. models with high target $|V|$, are prone to exhibit a spike at $10^3$. A common pattern for these is the **ratio of target training token count to number of parameters** falls into $O(10^{-4})$ for $10^2$ lines, $O(10^{-3})$ at $10^3$, $O(10^{-2})$ at $10^4$, and $O(10^{-1})$ for $10^5$ lines and so on. But for more atomic units such as alphabetic (not logographic) characters (may it be Latin, Cyrillic, or Abjad) and for bytes, this progression instead begins at $O(10^{-3})$ at $10^2$ lines. Instead of thinking this spike of $10^3$ as irregular, we may instead want to

---

[4]But there are no reasons why linguistics or linguistic typology cannot encompass a statistical science of language beyond/without "words", or with continuous representations of characters and bytes. In fact, that could complement the needs of language engineering and the NNs/DL/ML communities better.

think of this learning curve as shifted by 1 order of magnitude to the right for characters and bytes and/or the performance at $10^2$ lines for words and ZH-characters due to being overparameterized and hence abnormal. This would also fit in with the findings by Belkin et al. (2019) and Nakkiran et al. (2020) attributing DD to overparameterization. If we could use this ratio and logic of higher $|V|$ to automatically detect "non-atomic" units, ones that can be further decomposed, this observation could potentially be beneficial for advancing other sciences, e.g. biology. From a cognitive modeling perspective, the similarity in behavior of ZH characters and words of other languages can affirm the interpretation of wordhood for those ZH speakers who identify ZH characters as words (see also last paragraph in § 3 and Appendix J). While almost all work attribute DD to algorithmic reasons, concurrent work by Chen et al. (2020) corroborates our observation and confirms that DD arises due to "the interaction between the properties of the data and the inductive biases of learning algorithms". Other related work on DD and its more recent development can also be found in their work.

We performed additional experiments testing our setting on the datasets used by the Nakkiran et al. (2020) and testing our data on a non-neural LM. Results support our findings and are provided in Appendix K. Number of model parameters can be found in Appendix L.

**Erraticity**  We observe another type of sample-wise non-monotonicity, one that signals irregular and unstable performance across data sizes and runs. Within one run, erraticity can be observed directly as changes in direction on the y-axis. Across runs, large variance can be observed, even with the same dataset (see Figure 18 in Appendix M). Erraticity can also be observed indirectly through a negative correlation between data size and performance. Many work on length bias in NMT have focused on solutions related to search, e.g. Murray & Chiang (2018). Our experiments show that a kind of length bias can surface already with CLMing, without generation taking place. If the connection between erraticity and length bias can indeed be drawn, it could strengthen the case for global conditioning (Sountsov & Sarawagi, 2016). (See Appendix M for more discussion and results.)

**Script bias, erraticity, word bias — are these necessary conditions?**  To assess whether the observed phenomena are particular to this one setting, we performed one run with dataset A in 4 sizes with the primary representations on 1-layer Transformers (see Appendix N). We observed no significant disparity across the board. It shows that **larger/overparameterized models can magnify/exacerbate the differences in the data statistics**. That hyperparameter tuning — in this case, through the reduction of the number of layers — can mitigate effects from data statistics is, to the best of our knowledge, a novel insight, suggesting also that a general expectation of monotonic development as data size increases can indeed be held. Our other findings remain consistent (representational relativity, source language neutralization, and DD on word level).

**Bases for biases**  Recall in § 1, we "consider *bias* to be present when performance disparity in our Transformer models is statistically significant". As shown in our data statistics and analysis (Appendices D and P respectively), script bias, length bias wrt erraticity in CLMing, and word bias are all evident in the vocabulary and length information in the data statistics. Hence these disparities in performance are really a result of the Transformer being able to model these **differences in data** at such a magnitude that the differences are statistically significant. The meta phenomenon of erraticity, however, warrants an additional consideration indicative of the **empirical limits of our compute** (cf. Xu et al. (2020)), even when the non-monotonicity is not observed during the training of each model.

In eliminating performance disparity in character and byte models by normalizing vocabulary and length statistics in the data, we demonstrated that performance disparity as expected from the morphological complexity hierarchy is due to word tokenization, not intrinsic or necessary in language. This is the word bias. Qualitative issues in the concept of word will persist and make crosslinguistic comparison involving "words" unfair even if one were to be able to find a quantitative solution to mitigate the OOV issue, the bottleneck in word-based processing. We humans have a choice in how we see/process languages. That some might still prefer to continue with a crosslinguistic comparison with "words" and exert the superiority of "word" tokenization speaks for a view that is centered on "privileged" languages — in that case, **word bias is a human bias**.

And, in eliminating performance disparity across the board with our one-layer models, we show that all quantitative differences in data statistics between languages can also be modeled in a "zoomed-

out"/"desensitized" mode, suggesting that while languages can be perceived as being fundamentally different in different ways in different granularities, they can also be viewed as fundamentally similar.

# 6 ADDITIONAL RELATED WORK

Similar to our work in testing for hardness are Cotterell et al. (2018), Mielke et al. (2019), and Bugliarello et al. (2020). The first two studied (monolingual) LMs — the former tested on the Europarl languages (Koehn, 2005) with n-gram and character models and concluded that morphological complexity was the culprit to hardness, the latter studied 62 languages of the Bible corpus (Mayer & Cysouw, 2014) in addition and refuted the relevance of linguistic features in hardness based on character and BPE models on both corpora in word-tokenized form. Bugliarello et al. (2020) compared translation results of the Europarl languages with BPEs at one data size and concluded that it is easier to translate out of EN than into it, statistical significance was, however, not assessed. In contrast, we ablated away the confound of generation and studied CLMing with controls with a broader range of languages with more diverse statistical profiles in 3 granularities and up to 5 orders of magnitude in data size. That basic data statistics are the driver of success in performance in multilingual modeling has so far only been explicitly argued for in Mielke et al. (2019). We go beyond their work in monolingual LMs to study CLMs and evaluate also in relation to data size, representational granularity, and quantitative and qualitative fairness.

Bender (2009) advocated the relevance of linguistic typology for the design of language-independent NLP systems based on crosslinguistic differences in word-based structural notions, such as parts of speech. Ponti et al. (2019) found typological information to be beneficial in the few-shot setting on the character level for 77 languages with Latin scripts. But no multilingual work has thus far explicitly examined the relation between linguistic typology and the statistical properties of the data, involving languages with diverse statistical profiles in different granularities.

As obtaining training data is often the most difficult part of an NLP or Machine Learning (ML) project, Johnson et al. (2018) introduced an extrapolation methodology to directly model the relation between data size and performance. Our work can be viewed as one preliminary step towards this goal. To the best of our knowledge, there has been no prior work on demonstrating the neutralization of source language instances through statistical comparisons, a numerical analysis on DD for sequence-to-sequence models, the meta phenomenon of a sample-wise non-monotonicity (erraticity) being related to length, or the connection between effects of data statistics and modification in architectural depth.

# 7 CONCLUSION

**Summary** We performed a novel, rigorous relational assessment of performance disparity across different languages, representations, and data sizes in CLMing with the Transformer. Different disparity patterns were observed on different representation types (character, byte, and word), which can be traced back to the data statistics. The disparity pattern reflected on the word level corresponds to the morphological complexity hierarchy, reminding us that the definition of morphology is predicated on the notion of word and indicating how morphological complexity can be modeled by the Transformer simply through word segmentation. As we were able to eliminate disparity on the same data on the character and byte levels by normalizing length and vocabulary, we showed that morphological complexity is not a necessary concept but one that results from word segmentation and is bounded to the word level, orthogonal to the performance of character or byte models. Representational units of finer granularity were shown to help eliminate performance disparity though at the cost of longer sequence length, which can have a negative impact on robustness. In addition, we found all word models and character models with $\text{ZH}_{trg}$ to behave similarly in their being prone to exhibit a peak (as sample-wise DD) around $10^3$ lines in our setting. While bigger/overparameterized models can magnify the effect of data statistics, exacerbating the disparity, we found a decrease in model depth can eliminate these quantitative biases, leaving only the qualitative aspect of "word" and the necessity of word segmentation in question.

**Outlook** Machine learning has enabled greater diversity in NLP (Joshi et al., 2020). Fairness, in the elimination of disparity, does not require big data. This paper made a pioneering attempt to bridge research in DL/NNs, language sciences, and language engineering through a data-centric perspective.

We believe a **statistical** science for NLP as a data science can well complement algorithmic analyses with an empirical view contributing to a more generalizable pool of knowledge for NNs/DL/ML. A more comprehensive study not only can lead us to new scientific frontiers, but also better design and evaluation, benefitting the development of a more general, diverse and inclusive Artificial Intelligence.

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

# APPENDICES

## A RE-VISUALIZATION OF FIGURE 1 IN JUNCZYS-DOWMUNT ET AL. (2016) IN 6 FACETS BY TARGET LANGUAGE

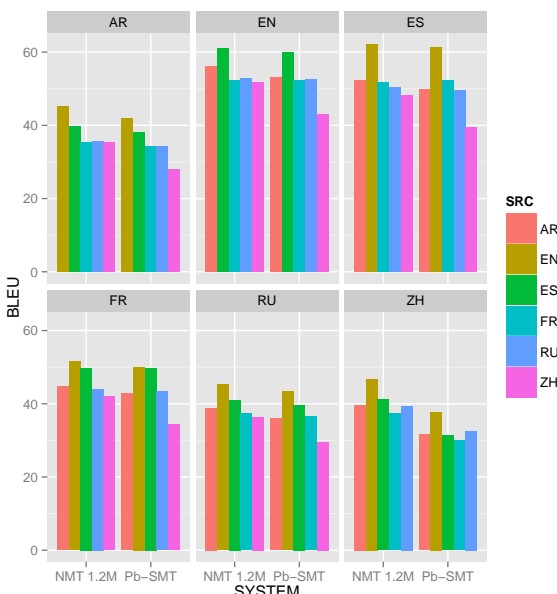

Figure 4: Results of the Moses baseline systems (right group in each facet) and neural models (left) with 1.2 million iterations (1 iteration corresponds to 1 mini-batch) for the 30 directions of the 6-way UN corpus, tokenized (ZH segmented), lowercased, and length filtered to 100 BPE tokens.

## B DATA SELECTION AND PREPROCESSING DETAILS

The UN Parallel Corpus v1.0 (Ziemski et al., 2016) consists of manually translated UN documents from 1990 to 2014 in the 6 official UN languages. Therein is a subcorpus that is fully aligned by line, comprising the 6-way parallel corpus we use. We tried to have as little preprocessing or filtering as necessary to eliminate possible confounds. But as the initial runs of our experiment failed due to insufficient memory on a single GPU with 12 GB VRAM[5], we filtered out lines with more than 300 characters in any language in lockstep with one another for all the 6 languages such that the subcorpora would remain parallel, thereby keeping the material of each language semantically equivalent to one another. 8,944,859 lines for each language were retained as our training data which cover up to the 75th percentile in line length for all 6 languages. In order to monitor the effect of data size, we made subcorpora of each language in 5 sizes by `head`ing the first $10^2, 10^3, 10^4, 10^5, 10^6$ lines[6]. We refer to this as dataset A. In addition, to better understand and verify the consistency of the phenomena observed, we made 2 supplemental datasets by shuffling the 8,944,859 lines two different times randomly and `head`ing the number of lines in our 5 sizes for each language, again in lockstep with one another (datasets B and C).

---

[5]GPUs used for experiments in this paper range from a NVIDIA TITAN RTX (24 GB), NVIDIA GeForce RTX 2080 Ti (11 GB), a GTX Titan X (12 GB), to a GTX 1080 (8 GB). All jobs were run on a single GPU setting. Some word-level experiments involving $AR_{trg}$ or $RU_{trg}$ at $10^6$ had to be run on a CPU as 24 GB VRAM were not sufficient. Models with higher maximum sequence lengths (e.g. byte models) were trained with 24 GB VRAM. Difference in equipment does not necessarily lead to degradation/improvement in scores.

[6]The terms "line" and "sentence" have been used interchangeably in the NLP literature. We use "line" to denote a sequence that ends with a newline character and "sentence" as one with an ending punctuation. Most parallel corpora, such as ours, are aligned by line, as a line may be part of a sentence or without an ending punctuation (e.g. a header/title). Using a standardized unit such as "line" would also be a fairer measure to linguae/scriptiones continuae (languages/scripts with no explicit punctuation).

For character modeling, we used a dummy symbol to denote each whitespace. For byte, we turned each UTF-8-encoded character into a byte string in decimal value, such that each token is a number between 0 and 255, inclusive. For word, we followed (Junczys-Dowmunt et al., 2016) and used the Moses tokenizer (Koehn et al., 2007) as is standard in NMT practice when word tokenization is applied and Jieba[7] for segmentation in ZH.

For Pinyin, we used the implementation from `https://github.com/lxyu/pinyin` in the numerical format such that each character/syllable is followed by a single digit indicating its lexical tone in Mandarin. For Wubi, we used the dictionary from the implementation from `https://github.com/arcsecw/wubi`.

We have implemented all representations such that they would be reversible even when the sequence contains code-mixing.

We used the official dev set as provided in (Ziemski et al., 2016), 3,077 lines per language remained from 4,000 after filtering line length to 300 characters. Data statistics is provided in Appendix D for reference.

The systematic training regime that we give to our language directions are identical for all. For each primary representation type (character, byte, and word), we performed:

- 5 runs in 5 sizes ($10^2 - 10^6$): A0 (seed=13), B0 (13), C0 (9948), A1 (9948), A2 (265), and
- 7 more runs in 4 sizes ($10^2 - 10^5$): A3 (777), A4 (42), A5 (340589), A6 (1000), A7 (83146), B1 (9948), & C1 (13).

For each run and each size, there are 30 pairwise directions (i.e. 1 source language to 1 target language, e.g. AR-EN for Arabic to English) that result from the 6 languages. We trained all 150 jobs for each run and representation using the Transformer model (Vaswani et al., 2017) as supported by the SOCKEYE Toolkit (Hieber et al., 2018) (version 1.18.85), based on MXNet (Chen et al., 2015). A detailed description of the architecture of the Transformer can be found in (Vaswani et al., 2017). The same set of hyperparameters applies to all and its values are listed in Appendix C.

**Notes on training time**  Each run of 30 directions in 5 sizes took approximately 8-12 days for character and byte models. Byte models generally took longer — hence training time is positively correlated with length (concurring with observations by Cherry et al. (2018) as they compared character with BPE models). A maximum length of 300 characters entails a maximum length of *at least* 300 bytes in UTF-8. Each run of word models (30 directions, 5 sizes) took about 6 days (excluding the training of some 7-9 directions out of 30 per run involving $AR_{trg}$ or $RU_{trg}$ at $10^6$ on word level which took about 12-18 hours *each direction* to train on a CPU as these required more space and would run out of memory (OOM) on our GPUs otherwise). These figures do not include the additional probing experiments described in § 4.

**Evaluation metric**  Most sequence-to-sequence models are optimized using a cross-entropy loss, defined as:

$$H(\boldsymbol{t}, \boldsymbol{s}) = -\sum_{i=1}^{N} \log_2 p(t_i \mid \boldsymbol{t}_{<i}, \boldsymbol{s}) \tag{1}$$

where $\boldsymbol{t}$ is the sequence of tokens to be predicted, $t_i$ refers to the $i^{th}$ token in that sequence, $\boldsymbol{s}$ is the sequence of tokens conditioned on, and $N = |\boldsymbol{t}|$. It is customary to report scores as PP, which is $2^{\frac{1}{N} H(\boldsymbol{t}, \boldsymbol{s})}$, i.e. 2 to the power of the cross-entropy averaged by the number of tokens (based on whichever granularity of unit is used for training) in the data. Cotterell et al. (2018) propose to use "renormalized" PP to evaluate LMs fairly through the division of an arbitrary constant. In our case, we choose instead a simpler method of using an "unnormalized" PP, i.e. the total number of bits needed to encode the development (dev) set, which has a constant size of 3,077 lines per language (after length filtering of the same dev set used in Junczys-Dowmunt et al. (2016)) for all various training sizes. As the implementation we used (SOCKEYE (Hieber et al., 2018)) only reports PP, we transform it back to entropy as defined above by noting that $H(\boldsymbol{t}, \boldsymbol{s}) = \log_2 PP(\boldsymbol{t}|\boldsymbol{s}) \times N$.

---

[7]`https://github.com/fxsjy/jieba`

## C  HYPERPARAMETER SETTING

- encoder transformer;
- decoder transformer;
- num-layers 6:6;
- num-embed 512:512;
- transformer-model-size 512;
- transformer-attention-heads 8;
- transformer-feed-forward-num-hidden 2048;
- transformer-activation-type relu;
- transformer-positional-embedding-type fixed;
- transformer-preprocess d; transformer-postprocess drn;
- transformer-dropout-attention 0.1;
- transformer-dropout-act 0.1;
- transformer-dropout-prepost 0.1;
- batch-size 15;
- batch-type sentence;
- max-num-checkpoint-not-improved 3;
- max-num-epochs 50;
- optimizer adam;
- optimized-metric perplexity;
- optimizer-params epsilon: 0.000000001, beta1: 0.9, beta2: 0.98;
- label-smoothing 0.0;
- learning-rate-reduce-num-not-improved 4;
- learning-rate-reduce-factor 0.001;
- loss-normalization-type valid;
- max-seq-len 300 for character, word, and BPE, 672 for all bytes, 688 for Wubi, 680 for Pinyin;
- checkpoint-frequency/interval 4000.
  (For smaller datasets, the end of 50 epochs is often reached before the first checkpoint. Since SOCKEYE only outputs scores at checkpoints, we adjusted the checkpoint frequency as follows to get a score outputted by the end of 50 epochs: 1000 for 100 lines for all character & byte instances, 400 for 100 lines for word and 500 for 100 lines BPE, 3450 for 1000 lines for word & BPE. For the very few cases that this default does not suffice due to bucketing of similar length sequences, we manually set the checkpoint frequency to the last batch.)

# D  DATA STATISTICS

- Number of types, i.e. vocabulary size ($|V|$). Note that Sockeye adds for its calculation 4 additional types: <pad>, , , <unk>.
- Number of tokens. This excludes the 1 EOS/BOS (end-/beginning-of-sentence) marker added by Sockeye to each line.
- Out-of-vocabulary (OOV) type rate (in %), i.e. the fraction of the types in the dev data that is not covered by the types in the training data.
- OOV token rate (in %), i.e. the fraction of tokens in the dev data that is treated as UNKnowns.
- Type-token-ratio (in %), i.e. the ratio between the number of types and tokens in the data. This is a rough proxy for lexical diversity in that a value of 1 would indicate that no type is ever seen twice, and a value very close to 0 would indicate that very few distinct types account for almost all of the data.
- Line length (excl. EOS/BOS marker): mean±standard deviation, and the 0/25/50/75/100-th percentile.

## Statistics for dataset A

*Table of data statistics for dataset A across representations (BYTE, CHAR, WORD, BPE) and languages (AR, EN, ES, FR, RU, ZH, ZH_pinyin, ZH_wubi, AR_cp1256, RU_cp1251), with columns for number of lines 100 / 1,000 / 100,000 / 1,000,000 and measured quantities: Number of TYPES, Number of TOKENS, OOV type rate (%), OOV token rate (%), TTR (%), and Mean line length±std 0/25/50/75/100-th.*

## Statistics for dataset B

## Statistics for dataset C

| Representation / Number of lines | CHAR 100 | CHAR 1,000 | CHAR 10,000 | CHAR 100,000 | CHAR 1,000,000 | BYTE 100 | BYTE 1,000 | BYTE 10,000 | BYTE 100,000 | BYTE 1,000,000 | WORD 100 | WORD 1,000 | WORD 10,000 | WORD 100,000 | WORD 1,000,000 | BPE 100 | BPE 1,000 | BPE 10,000 | BPE 100,000 | BPE 1,000,000 |
|---|---|---|---|---|---|---|---|---|---|---|---|---|---|---|---|---|---|---|---|---|
| **Number of TYPES** | | | | | | | | | | | | | | | | | | | | |
| AR | 97 | 136 | 162 | 261 | 427 | 92 | 123 | 144 | 160 | 176 | 1,165 | 6,823 | 29,296 | 97,380 | 310,632 | 864 | 4,530 | 18,735 | 29,673 | 29,954 |
| EN | 78 | 98 | 126 | 186 | 310 | 81 | 103 | 131 | 159 | 178 | 906 | 4,775 | 13,658 | 42,677 | 141,163 | 802 | 3,484 | 10,800 | 20,766 | 29,215 |
| ES | 84 | 108 | 133 | 186 | 306 | 85 | 112 | 133 | 160 | 175 | 970 | 4,734 | 16,850 | 51,281 | 154,649 | 803 | 3,686 | 12,670 | 26,134 | 29,470 |
| FR | 87 | 113 | 139 | 195 | 335 | 91 | 117 | 134 | 160 | 176 | 991 | 4,603 | 15,586 | 45,335 | 132,493 | 853 | 3,683 | 12,170 | 27,731 | 29,355 |
| RU | 98 | 139 | 167 | 211 | 287 | 100 | 140 | 154 | 166 | 177 | 1,165 | 6,890 | 26,112 | 84,137 | 250,778 | 1,003 | 4,858 | 19,029 | 29,322 | 29,814 |
| ZH | 783 | 1,550 | 2,443 | 3,477 | 4,698 | 128 | 152 | 162 | 177 | 189 | 882 | 4,360 | 14,730 | 46,667 | 139,895 | 1,084 | 3,716 | 11,214 | 27,443 | 29,130 |
| ZH_pinyin | 74 | 107 | 139 | 240 | 413 | 97 | 136 | 162 | 261 | 427 | | | | | | | | | | |
| ZH_wubi | 95 | 129 | 154 | 242 | 408 | 98 | 141 | 167 | 211 | 287 | | | | | | | | | | |
| AR_cp1256 | | | | | | | | | | | | | | | | | | | | |
| RU_cp1251 | | | | | | | | | | | | | | | | | | | | |
| **Number of TOKENS** | | | | | | | | | | | | | | | | | | | | |
| AR | 10,212 | 104,144 | 1,021,136 | 10,231,081 | 102,408,339 | 18,448 | 188,507 | 1,845,136 | 18,482,689 | 185,105,498 | 1,889 | 19,281 | 190,738 | 1,912,669 | 19,137,210 | 3,518 | 25,601 | 210,758 | 2,087,431 | 20,981,760 |
| EN | 12,300 | 123,646 | 1,208,927 | 12,089,449 | 120,933,962 | 12,306 | 123,681 | 1,206,187 | 12,093,117 | 120,971,460 | 2,116 | 21,545 | 212,034 | 2,123,153 | 21,231,465 | 3,369 | 25,111 | 221,264 | 2,160,162 | 21,666,482 |
| ES | 14,125 | 139,663 | 1,369,779 | 13,701,387 | 137,133,440 | 14,387 | 143,237 | 1,395,305 | 13,958,221 | 139,702,853 | 2,472 | 24,600 | 242,672 | 2,430,586 | 24,319,760 | 3,784 | 28,726 | 254,092 | 2,483,995 | 24,805,209 |
| FR | 13,797 | 138,528 | 1,365,360 | 13,661,150 | 136,754,360 | 14,243 | 143,139 | 1,411,451 | 14,159,815 | 141,343,075 | 2,453 | 24,950 | 248,488 | 2,485,118 | 24,864,449 | 3,755 | 28,907 | 258,550 | 2,527,957 | 25,362,724 |
| RU | 13,763 | 136,674 | 1,339,286 | 13,309,690 | 133,671,322 | 25,345 | 251,231 | 2,457,508 | 24,535,771 | 245,388,139 | 2,024 | 20,474 | 201,618 | 2,016,451 | 20,156,467 | 3,813 | 26,772 | 259,794 | 2,772,616 | 21,787,539 |
| ZH | 3,371 | 34,429 | 341,530 | 3,419,448 | 34,200,543 | 9,235 | 95,100 | 938,805 | 9,404,386 | 94,052,681 | 1,824 | 19,120 | 190,485 | 1,907,643 | 19,073,463 | 2,559 | 22,067 | 210,794 | 1,947,300 | 19,373,372 |
| ZH_pinyin | 11,605 | 119,256 | 1,171,605 | 11,732,010 | 117,336,880 | 10,212 | 104,144 | 1,021,136 | 10,231,081 | 102,448,339 | | | | | | | | | | |
| ZH_wubi | 10,071 | 103,944 | 1,022,870 | 10,245,716 | 102,450,581 | 13,763 | 136,674 | 1,339,286 | 13,360,690 | 133,360,690 | | | | | | | | | | |
| **OOV type rate (%)** | | | | | | | | | | | | | | | | | | | | |
| AR | 28.46 | 5.38 | 0.00 | 0.00 | 0.00 | 26.02 | 4.88 | 0.00 | 0.00 | 0.00 | 90.72 | 71.13 | 35.70 | 13.44 | 5.02 | 6.06 | 0.34 | 0.00 | 0.00 | 0.00 |
| EN | 21.65 | 12.37 | 8.25 | 0.00 | 0.00 | 22.35 | 11.76 | 2.06 | 0.00 | 0.00 | 89.47 | 60.56 | 26.27 | 10.43 | 4.32 | 3.62 | 0.49 | 0.18 | 0.04 | 0.00 |
| ES | 25.23 | 14.41 | 10.81 | 5.41 | 0.00 | 23.64 | 10.91 | 0.00 | 0.00 | 0.00 | 90.71 | 63.96 | 27.18 | 9.71 | 3.88 | 4.44 | 0.53 | 0.19 | 0.06 | 0.00 |
| FR | 23.01 | 9.73 | 7.08 | 1.77 | 0.00 | 22.88 | 9.32 | 0.85 | 0.00 | 0.00 | 89.92 | 62.27 | 25.96 | 8.89 | 3.48 | 3.96 | 0.49 | 0.15 | 0.02 | 0.00 |
| RU | 33.56 | 7.53 | 2.05 | 0.00 | 0.00 | 33.13 | 5.56 | 1.39 | 0.00 | 0.00 | 93.11 | 69.66 | 32.15 | 10.91 | 3.81 | 4.82 | 0.29 | 0.01 | 0.02 | 0.00 |
| ZH | 27.55 | 26.87 | 6.73 | 1.42 | 0.35 | 16.99 | 3.92 | 0.65 | 0.00 | 0.00 | 90.34 | 63.48 | 29.89 | 10.31 | 4.09 | 67.98 | 26.22 | 5.03 | 0.55 | 0.10 |
| ZH_pinyin | | 8.16 | 1.02 | 0.00 | 0.00 | 28.46 | 5.38 | 0.00 | 0.00 | 0.00 | | | | | | | | | | |
| ZH_wubi | | 4.17 | 0.83 | 0.00 | 0.00 | 33.56 | 7.53 | 0.00 | 0.00 | 0.00 | | | | | | | | | | |
| **OOV token rate (%)** | | | | | | | | | | | | | | | | | | | | |
| AR | 0.13 | 0.01 | 0.00 | 0.00 | 0.00 | 0.05 | 0.00 | 0.00 | 0.00 | 0.00 | 53.40 | 25.24 | 9.31 | 3.25 | 1.18 | 0.73 | 0.12 | 0.00 | 0.00 | 0.00 |
| EN | 0.03 | 0.00 | 0.00 | 0.00 | 0.00 | 0.03 | 0.00 | 0.00 | 0.00 | 0.00 | 34.26 | 11.63 | 3.41 | 1.36 | 0.51 | 0.30 | 0.03 | 0.02 | 0.00 | 0.00 |
| ES | 0.04 | 0.01 | 0.00 | 0.00 | 0.00 | 0.05 | 0.00 | 0.01 | 0.00 | 0.00 | 32.17 | 12.05 | 3.56 | 1.16 | 0.46 | 0.29 | 0.04 | 0.04 | 0.02 | 0.00 |
| FR | 0.15 | 0.01 | 0.00 | 0.00 | 0.00 | 0.08 | 0.00 | 0.00 | 0.00 | 0.00 | 31.39 | 11.66 | 3.30 | 1.03 | 0.41 | 0.38 | 0.04 | 0.02 | 0.02 | 0.00 |
| RU | 1.15 | 0.00 | 0.18 | 0.04 | 0.00 | 0.08 | 0.00 | 0.01 | 0.00 | 0.00 | 49.48 | 22.44 | 7.33 | 2.33 | 0.79 | 22.01 | 4.75 | 0.81 | 0.09 | 0.01 |
| ZH | 0.07 | 1.25 | 0.00 | 0.10 | 0.00 | 0.10 | 0.16 | 0.02 | 0.00 | 0.00 | 38.65 | 13.96 | 4.52 | 1.42 | 0.56 | | | | | |
| ZH_pinyin | 0.09 | 0.00 | 0.00 | 0.00 | 0.00 | 0.13 | 0.01 | 0.00 | 0.01 | 0.00 | | | | | | | | | | |
| ZH_wubi | | 0.00 | 0.00 | 0.00 | 0.00 | 0.15 | 0.00 | 0.00 | 0.00 | 0.00 | | | | | | | | | | |
| **TTR (%)** | | | | | | | | | | | | | | | | | | | | |
| AR | 0.95 | 0.13 | 0.02 | 0.00 | 0.00 | 0.50 | 0.07 | 0.01 | 0.00 | 0.00 | 61.67 | 35.39 | 14.84 | 5.09 | 1.62 | 24.56 | 17.69 | 8.89 | 1.42 | 0.14 |
| EN | 0.63 | 0.08 | 0.01 | 0.00 | 0.00 | 0.66 | 0.08 | 0.01 | 0.01 | 0.00 | 42.82 | 19.37 | 6.44 | 2.01 | 0.66 | 23.81 | 13.87 | 4.88 | 1.24 | 0.13 |
| ES | 0.63 | 0.08 | 0.01 | 0.00 | 0.00 | 0.64 | 0.08 | 0.01 | 0.01 | 0.00 | 39.24 | 19.34 | 6.94 | 2.11 | 0.64 | 21.22 | 12.83 | 4.91 | 1.10 | 0.12 |
| FR | 0.71 | 0.10 | 0.01 | 0.00 | 0.00 | 0.39 | 0.06 | 0.01 | 0.01 | 0.00 | 40.40 | 18.45 | 6.27 | 1.82 | 0.53 | 22.72 | 12.78 | 4.71 | 1.30 | 0.14 |
| RU | 23.23 | 4.50 | 0.72 | 0.10 | 0.01 | 1.39 | 0.16 | 0.02 | 0.02 | 0.01 | 57.56 | 31.70 | 12.95 | 3.24 | 1.24 | 26.28 | 18.15 | 8.66 | 1.35 | 0.14 |
| ZH | 0.64 | 0.09 | 0.01 | 0.00 | 0.00 | 0.95 | 0.13 | 0.02 | 0.02 | 0.02 | 48.36 | 21.76 | 7.73 | 2.45 | 0.73 | 42.36 | 16.84 | 5.62 | 1.41 | 0.15 |
| ZH_pinyin | 0.34 | 0.12 | | | | 0.71 | 0.10 | | | | | | | | | | | | | |
| ZH_wubi | | | | | | | | | | | | | | | | | | | | |
| **Mean line length ±std 025/50/75/00th** | | | | | | | | | | | | | | | | | | | | |
| AR | 102.32 ± 57.40 5/57/96/152,213 | 104.14 ± 57.83 4/54/106/149,284 | 102.11 ± 56.06 3/53/102/147,293 | 102.31 ± 56.33 1/53/102/147,298 | 102.45 ± 56.46 1/53/102/147,300 | 184.98 ± 105.28 9/101/173/280,383 | 188.51 ± 105.63 6/99/191/270,487 | 184.83 ± 100.15 1/95/184/296/551 | 184.83 ± 100.15 1/95/184/296/551 | 185.11 ± 100.40 1/95/185/296/559 | 18.89 ± 10.27 2/11/18/27,43 | 19.29 ± 10.59 1/10/20/27,50 | 19.07 ± 10.72 1/10/19/27/82 | 19.13 ± 10.82 1/10/19/27,120 | 19.14 ± 10.84 1/10/19/27,143 | 35.18 ± 18.92 4/18/34/50,73 | 25.00 ± 14.42 1/14/26/36,79 | 21.07 ± 12.01 1/11/21/30,86 | 20.67 ± 12.00 1/11/21/29,123 | 20.08 ± 12.10 1/11/21/29,147 |
| EN | 123.00 ± 72.30 9/63/116/187,268 | 123.65 ± 71.12 3/90/128/178,300 | 120.88 ± 69.02 1/61/122/175/300 | 120.89 ± 69.58 1/62/122/175/300 | 120.93 ± 69.60 1/61/122/175/300 | 123.05 ± 71.15 3/60/120/179,487 | 123.68 ± 71.15 9/63/116/187,268 | 120.93 ± 69.04 1/62/122/175/300 | 120.91 ± 72.89 1/62/122/175/300 | 120.97 ± 69.62 1/61/122/175/300 | 21.16 ± 12.24 2/10/21/31,48 | 21.55 ± 12.31 1/11/22/33,55 | 21.20 ± 12.15 1/11/21/30/136 | 21.23 ± 12.17 1/11/21/30/118 | 21.23 ± 12.17 1/11/21/30/136 | 33.09 ± 19.32 3/15/33/49,78 | 25.11 ± 14.82 1/13/26/36,86 | 22.13 ± 12.67 1/12/22/32,88 | 21.60 ± 12.36 1/12/22/31,120 | 21.67 ± 12.42 1/12/22/31,138 |
| ES | 141.25 ± 82.83 11/59/134/215/292 | 139.56 ± 79.20 3/67/145/201,300 | 136.98 ± 78.35 1/70/138/199/300 | 137.01 ± 78.45 1/70/138/199/300 | 137.13 ± 78.47 1/70/138/199/300 | 143.87 ± 84.37 11/60/136/218/298 | 142.24 ± 81.05 3/69/149/206/308 | 139.13 ± 79.72 1/71/140/204/308 | 139.58 ± 79.84 1/71/141/203/316 | 139.70 ± 79.87 1/71/141/203/316 | 24.72 ± 14.39 1/13/24/35,60 | 24.60 ± 13.91 1/12/25/36/60 | 24.27 ± 13.80 1/13/24/35/84 | 24.31 ± 13.87 1/13/24/35,100 | 24.32 ± 13.87 1/13/24/35,139 | 37.84 ± 21.74 4/17/35/56,86 | 28.73 ± 16.27 1/14/30/41,90 | 25.41 ± 14.46 1/13/25/36/90 | 24.84 ± 14.17 1/13/25/36/121 | 24.93 ± 14.22 1/13/25/36/140 |
| FR | 137.07 ± 80.12 6/52/133/215/293 | 138.53 ± 79.49 3/65/141/203/300 | 136.54 ± 78.80 1/68/137/200/300 | 136.61 ± 78.93 1/69/138/199/300 | 136.75 ± 78.96 1/69/138/199/300 | 142.43 ± 82.18 6/64/137/219/303 | 143.14 ± 81.47 3/70/145/209/315 | 141.20 ± 81.56 1/71/142/206/319 | 141.34 ± 81.61 1/71/142/206/316 | 141.34 ± 81.61 1/71/142/206/316 | 24.33 ± 13.79 1/13/24/35,62 | 24.69 ± 14.19 1/12/26/36,63 | 24.85 ± 14.28 1/13/25/36/138 | 24.85 ± 14.31 1/13/25/36/118 | 24.86 ± 14.31 1/13/25/36/118 | 37.55 ± 20.97 1/14/30/42/86 | 28.91 ± 16.41 1/13/30/41,88 | 25.86 ± 14.85 1/13/26/37,88 | 25.28 ± 14.53 1/13/25/36/124 | 25.36 ± 14.58 1/13/26/36/138 |
| RU | 137.63 ± 80.80 4/70/133/215/293 | 136.67 ± 78.77 3/63/139/200/300 | 133.93 ± 77.25 1/67/135/196/300 | 133.70 ± 77.26 1/67/134/195/300 | 133.67 ± 77.25 1/67/134/195/300 | 253.45 ± 150.90 21/126/242/382/550 | 251.23 ± 146.21 4/118/235/366/561 | 245.34 ± 143.17 1/122/247/358/569 | 245.36 ± 143.18 1/122/247/359/560 | 245.34 ± 143.17 1/122/247/358/569 | 20.16 ± 11.44 1/11/20/29/82 | 20.47 ± 11.61 1/10/21/29,55 | 20.16 ± 11.53 1/11/20/29/136 | 20.16 ± 11.53 1/11/20/29/117 | 20.16 ± 11.53 1/11/20/29/136 | 38.13 ± 21.16 3/18/38/56/81 | 26.77 ± 15.36 1/14/27/38/78 | 21.08 ± 12.61 1/12/22/31/87 | 21.73 ± 12.64 1/11/22/31/121 | 21.79 ± 12.64 1/11/22/31/138 |
| ZH | 33.71 ± 18.03 4/19/33/48/170 | 34.43 ± 19.40 2/18/35/49/127 | 34.15 ± 19.06 1/19/34/48/133 | 34.20 ± 19.27 1/19/34/48/220 | 34.20 ± 19.28 1/19/34/48/272 | 92.35 ± 52.42 12/47/93/132/188 | 95.30 ± 54.29 3/70/96/138/248 | 93.88 ± 53.42 3/49/94/135/378 | 94.04 ± 53.75 1/49/94/135/378 | 94.05 ± 53.79 1/48/94/135/504 | 18.24 ± 9.97 2/10/17/26/39 | 19.12 ± 10.87 1/10/30/27/55 | 19.05 ± 10.86 1/10/19/27/248 | 19.08 ± 10.91 1/10/19/27/113 | 19.07 ± 10.92 1/10/19/27/248 | 25.59 ± 13.84 2/14/26/37/52 | 22.07 ± 12.08 1/12/22/33/90 | 19.97 ± 11.43 1/11/20/28/92 | 19.47 ± 11.17 1/11/19/27/115 | 19.57 ± 11.36 1/11/19/28/249 |
| ZH_pinyin | 116.05 ± 67.54 11/58/117/167/283 | 119.26 ± 68.54 3/61/122/173/320 | 117.16 ± 67.25 1/60/118/169/335 | 117.82 ± 67.49 1/60/118/169/584 | 117.86 ± 67.55 1/60/118/169/584 | | | | | | | | | | | | | | | |
| ZH_wubi | 100.71 ± 57.05 13/54/101/139/217 | 103.94 ± 59.48 3/54/105/150/296 | 102.29 ± 57.92 3/54/102/147/297 | 102.46 ± 54.39 1/53/102/146/398 | 102.45 ± 58.31 1/53/103/146/627 | | | | | | | | | | | | | | | |
| AR_cp1256 | | | | | | 102.33 ± 56.00 3/53/102/147/293 | 104.14 ± 57.83 4/54/106/149/284 | 102.11 ± 56.06 3/53/102/147/293 | 102.45 ± 56.33 1/53/102/147/298 | 102.45 ± 56.46 1/53/103/146/300 | | | | | | | | | | |
| RU_cp1251 | | | | | | 133.93 ± 80.80 1/71/130/212/296 | 133.93 ± 77.25 3/54/102/147/297 | 133.67 ± 77.25 1/67/134/195/300 | 133.67 ± 77.25 1/67/135/196/300 | 133.67 ± 77.25 1/67/134/195/300 | | | | | | | | | | |

Statistics for development (dev) set

As a different set of vocabulary is learned from each training dataset and data size, BPE has a distinct dev set for each.

| Representation / Number of lines in train set | CHAR | BYTE | WORD | BPE_A 100 | BPE_A 1,000 | BPE_A 100,000 | BPE_A 1,000,000 | BPE_B 100 | BPE_B 1,000 | BPE_B 100,000 | BPE_B 1,000,000 | BPE_C 100 | BPE_C 1,000 | BPE_C 100,000 | BPE_C 1,000,000 |
|---|---|---|---|---|---|---|---|---|---|---|---|---|---|---|---|
| **Number of TYPES** | | | | | | | | | | | | | | | |
| AR | 130 | 123 | 13,836 | 708 | 3,232 | 8,994 | 12,430 | 865 | 3,991 | 12,954 | 13,000 | 908 | 4,119 | 12,934 | 13,003 |
| EN | 97 | 102 | 7,199 | 656 | 2,567 | 7,461 | 7,556 | 801 | 3,149 | 7,557 | 7,528 | 823 | 3,245 | 7,524 | 7,573 |
| ES | 111 | 110 | 8,551 | 680 | 2,832 | 6,909 | 8,871 | 833 | 3,436 | 8,869 | 8,906 | 834 | 3,424 | 8,871 | 8,900 |
| FR | 113 | 118 | 8,312 | 714 | 2,821 | 6,605 | 8,466 | 827 | 3,407 | 7,689 | 8,705 | 883 | 3,463 | 7,631 | 8,708 |
| RU | 146 | 144 | 12,819 | 878 | 3,769 | 10,085 | 12,788 | 1,024 | 4,438 | 11,052 | 12,054 | 1,038 | 4,509 | 12,882 | 12,958 |
| ZH | 1,976 | 153 | 7,413 | 3,224 | 4,215 | 6,386 | 7,654 | 3,261 | 4,481 | 7,719 | 7,702 | 3,260 | 4,524 | 7,654 | 7,721 |
| ZH_pinyin | 98 | 130 | | | | | | | | | | | | | |
| ZH_wubi | 120 | 146 | | | | | | | | | | | | | |
| AR_cp1256 | 376,679 | 334,358 | | | | | | | | | | | | | |
| RU_cp1251 | 330,734 | 431,538 | | | | | | | | | | | | | |
| **Number of TOKENS** | | | | | | | | | | | | | | | |
| AR | 334,358 | 605,516 | 61,371 | 167,574 | 115,693 | 83,001 | 70,527 | 149,689 | 97,883 | 68,538 | 68,270 | 149,623 | 97,231 | 68,579 | 68,278 |
| EN | 301,222 | 391,260 | 67,629 | 156,826 | 101,782 | 77,089 | 70,339 | 140,256 | 90,871 | 69,633 | 69,348 | 140,377 | 90,360 | 69,633 | 69,341 |
| ES | 443,958 | 452,800 | 78,087 | 170,133 | 113,083 | 88,634 | 81,341 | 155,687 | 103,788 | 80,569 | 80,587 | 154,746 | 103,172 | 80,579 | 80,371 |
| FR | 438,083 | 452,556 | 78,745 | 166,289 | 114,694 | 88,726 | 81,559 | 156,256 | 104,067 | 80,844 | 80,671 | 153,745 | 104,091 | 80,912 | 80,604 |
| RU | 431,538 | 703,214 | 64,180 | 177,818 | 113,628 | 79,763 | 71,081 | 163,319 | 100,294 | 70,200 | 69,991 | 163,806 | 99,978 | 70,196 | 69,982 |
| ZH | 107,990 | 301,085 | 60,013 | 96,745 | 80,231 | 68,129 | 62,810 | 98,775 | 75,636 | 61,867 | 61,882 | 94,127 | 75,718 | 61,916 | 61,823 |
| **TTR (%)** | | | | | | | | | | | | | | | |
| AR | 0.04 | 0.02 | 22.54 | 0.42 | 2.79 | 10.84 | 17.62 | 0.58 | 4.08 | 18.90 | 19.17 | 0.61 | 4.24 | 18.86 | 19.18 |
| EN | 0.03 | 0.02 | 10.64 | 0.42 | 2.52 | 7.78 | 10.74 | 0.57 | 3.47 | 10.85 | 10.93 | 0.59 | 3.59 | 10.81 | 10.02 |
| ES | 0.03 | 0.02 | 10.95 | 0.40 | 2.48 | 7.81 | 10.91 | 0.54 | 3.30 | 11.01 | 11.15 | 0.54 | 3.32 | 11.01 | 11.12 |
| FR | 0.03 | 0.03 | 10.56 | 0.45 | 2.46 | 7.80 | 10.63 | 0.53 | 3.27 | 10.72 | 10.79 | 0.57 | 3.33 | 10.73 | 10.80 |
| RU | 0.03 | 0.02 | 19.97 | 0.49 | 3.32 | 12.64 | 17.99 | 0.63 | 4.42 | 18.35 | 18.51 | 0.63 | 4.51 | 18.35 | 18.52 |
| ZH | 1.83 | 0.05 | 12.35 | 3.33 | 5.25 | 9.37 | 12.19 | 3.48 | 5.92 | 12.48 | 12.46 | 3.46 | 5.97 | 12.36 | 12.49 |
| ZH_pinyin | 0.03 | 0.04 | | | | | | | | | | | | | |
| ZH_wubi | 0.04 | 0.03 | | | | | | | | | | | | | |
| **Mean line length std 0/25/50/75/100th** | | | | | | | | | | | | | | | |
| AR | 108.66 ± 58.01 3/60/110/153/277 | 106.79 ± 105.85 6/107/199/277/503 | 19.95 ± 10.08 1/12/20/27/58 | 54.46 ± 29.51 1/30/54/76/152 | 37.60 ± 20.51 1/22/37/52/125 | 28.07 ± 14.80 1/16/27/37/95 | 22.92 ± 12.56 1/14/22/32/75 | 48.65 ± 26.35 1/27/49/68/156 | 31.80 ± 17.76 1/18/32/44/110 | 23.83 ± 13.15 1/14/24/33/80 | 22.19 ± 12.07 1/13/22/31/71 | 48.63 ± 26.21 1/28/49/67/145 | 31.60 ± 17.53 1/18/31/44/110 | 22.29 ± 12.14 1/13/22/31/73 | 22.19 ± 12.06 1/13/22/31/72 |
| EN | 127.14 ± 68.64 6/86/130/181/289 | 127.16 ± 68.65 6/86/130/181/299 | 21.08 ± 11.81 1/13/22/31/61 | 50.64 ± 27.44 1/27/51/72/136 | 33.08 ± 18.07 1/19/33/46/110 | 25.16 ± 12.42 1/14/25/35/78 | 22.86 ± 12.27 1/13/23/32/71 | 45.58 ± 24.86 1/26/46/65/129 | 29.53 ± 16.30 1/17/29/41/94 | 23.90 ± 12.95 1/14/24/33/74 | 22.54 ± 12.05 1/13/23/31/65 | 45.62 ± 24.67 1/26/46/66/122 | 29.37 ± 16.13 1/17/29/41/92 | 22.63 ± 12.12 1/13/23/31/65 | 22.54 ± 12.05 1/13/23/31/66 |
| ES | 114.28 ± 77.78 5/77/140/207/300 | 140.96 ± 79.21 5/78/149/211/307 | 25.26 ± 13.56 1/15/25/36/63 | 55.30 ± 30.13 1/30/56/79/141 | 37.14 ± 20.13 1/21/37/52/111 | 28.51 ± 15.53 1/17/29/41/85 | 26.44 ± 14.29 1/15/26/37/73 | 50.01 ± 27.66 1/27/51/72/133 | 33.72 ± 18.48 1/19/34/47/103 | 27.00 ± 14.88 1/15/28/39/75 | 26.11 ± 13.94 1/15/26/37/70 | 50.28 ± 27.27 1/28/51/71/133 | 33.53 ± 18.38 1/19/34/47/95 | 26.19 ± 13.99 1/15/26/37/71 | 26.12 ± 13.94 1/14/27/38/94 |
| FR | 142.37 ± 77.81 4/74/145/205/300 | 147.08 ± 80.30 4/77/150/212/310 | 25.59 ± 13.86 1/14/26/36/96 | 54.04 ± 29.55 1/29/55/76/139 | 37.27 ± 20.65 1/20/38/52/119 | 28.84 ± 15.76 1/16/29/40/89 | 26.51 ± 14.35 1/15/27/37/70 | 50.78 ± 27.79 1/27/52/72/135 | 33.82 ± 18.74 1/18/34/47/108 | 27.66 ± 15.11 1/15/28/39/80 | 26.22 ± 14.16 1/14/27/37/69 | 49.97 ± 27.25 1/27/51/70/131 | 33.85 ± 18.80 1/18/34/47/107 | 26.38 ± 14.21 1/14/27/37/69 | 26.21 ± 14.16 1/14/27/37/69 |
| RU | 140.25 ± 76.21 5/75/141/200/300 | 237.79 ± 141.53 7/138/255/370/569 | 20.86 ± 11.25 1/12/21/29/69 | 57.79 ± 32.12 1/31/58/82/185 | 36.93 ± 20.73 1/21/36/52/185 | 25.02 ± 14.48 1/15/26/36/128 | 23.10 ± 12.71 1/13/23/32/93 | 53.68 ± 29.00 1/29/53/75/185 | 32.09 ± 18.42 1/18/32/46/106 | 24.39 ± 13.59 1/14/24/34/112 | 22.75 ± 12.46 1/13/23/32/93 | 53.24 ± 29.18 2/30/53/76/185 | 32.49 ± 18.49 1/18/32/45/161 | 24.41 ± 13.57 1/14/24/34/110 | 22.74 ± 12.45 1/13/23/32/92 |
| ZH | 35.10 ± 18.48 2/21/35/49/125 | 97.85 ± 52.10 4/59/90/138/288 | 19.50 ± 10.42 1/11/20/27/64 | 31.44 ± 16.90 1/18/31/44/100 | 26.07 ± 14.19 1/16/26/36/95 | 22.14 ± 12.02 1/13/22/31/88 | 20.41 ± 11.05 1/12/20/28/74 | 30.68 ± 16.47 1/17/30/43/101 | 24.56 ± 13.42 1/14/24/34/92 | 21.19 ± 11.49 1/12/21/29/71 | 20.09 ± 10.79 1/12/20/28/65 | 30.49 ± 16.45 1/18/30/43/101 | 24.61 ± 13.49 1/14/24/34/93 | 20.12 ± 10.81 1/12/20/28/64 | 20.09 ± 10.79 1/12/20/28/65 |
| ZH_pinyin | 122.32 ± 65.50 6/07/125/173/353 | 109.66 ± 58.01 3/60/110/153/277 | | | | | | | | | | | | | |
| ZH_wubi | 107.49 ± 56.73 4/60/108/151/294 | 140.25 ± 76.21 5/75/141/200/300 | | | | | | | | | | | | | |

# E   SCORE TABLES

Number of bits to encode the dev data for each of the 30 language directions. Shown is mean±std over:

- 12 runs for CHAR, BYTE, and WORD from 100 to 100,000 lines,
- 5 runs for 1,000,000 lines, and
- 3 runs for all sizes involving alternate representations (BPE, Pinyin, Wubi, cp 1256 and cp 1251).

_[Large landscape score table with column groups CHAR, BYTE, WORD, and BPE (each spanning sizes 100, 1,000, 100,000, 1,000,000), and rows for the 30 language directions: AR-EN, AR-ES, AR-FR, AR-RU, AR-ZH, EN-AR, EN-ES, EN-FR, EN-RU, EN-ZH, ES-AR, ES-EN, ES-FR, ES-RU, ES-ZH, FR-AR, FR-EN, FR-ES, FR-RU, FR-ZH, RU-AR, RU-EN, RU-ES, RU-FR, RU-ZH, ZH-AR, ZH-EN, ZH-ES, ZH-FR, ZH-RU; plus additional rows for alternate representations such as AR-ZH_pinyin, EN-ZH_pinyin, ES-ZH_pinyin, FR-ZH_pinyin, RU-ZH_pinyin, AR-ZH_wubi, EN-ZH_wubi, ES-ZH_wubi, FR-ZH_wubi, RU-ZH_wubi, EN-AR_cp1256, ES-AR_cp1256, FR-AR_cp1256, RU-AR_cp1256, ZH-AR_cp1256, AR-RU_cp1251, EN-RU_cp1251, ES-RU_cp1251, FR-RU_cp1251, ZH-RU_cp1251, AR_cp1256-RU_cp1251, RU_cp1251-AR_cp1256.]_

## F CORRELATION STATISTICS

Best correlating metrics, i.e. the union of top 3 metrics for all representations.
For each representation, the **top 3 metrics** are boldfaced.
All correlations are **highly significant** ($p < 10^{-30}$), except for min source length for WORD ($p \approx 0.0001$) and min target length for WORD ($p \approx 0.3861$).

| Metric | CHAR | Pinyin | Wubi | BYTE | $\text{ARRU}_t$ | $\text{ARRU}_{s,t}$ | WORD | BPE |
|---|---|---|---|---|---|---|---|---|
| minimum length (target) | **0.84** | **0.85** | **0.86** | **0.60** | **0.84** | **0.84** | −0.02 | 0.65 |
| minimum length (source) | **0.82** | **0.84** | **0.85** | 0.57 | **0.84** | **0.84** | 0.10 | 0.64 |
| number of tokens (source) | −0.78 | −0.81 | −0.82 | **−0.60** | **−0.81** | **−0.81** | −0.59 | **−0.83** |
| TTR (target) | **0.83** | **0.83** | **0.84** | 0.48 | 0.81 | 0.81 | 0.61 | 0.83 |
| $|V|$ (source) | −0.54 | −0.51 | −0.51 | −0.50 | −0.67 | −0.68 | **−0.63** | **−0.86** |
| data size in lines | −0.80 | −0.83 | −0.83 | −0.59 | −0.81 | −0.81 | −0.62 | **−0.86** |
| OOV token rate (target) | 0.69 | 0.66 | 0.66 | 0.47 | 0.67 | 0.68 | **0.66** | 0.62 |
| OOV type rate (target) | 0.70 | 0.71 | 0.72 | 0.47 | 0.69 | 0.70 | **0.65** | 0.62 |
| TTR (source) | 0.67 | 0.71 | 0.71 | **0.60** | 0.81 | 0.81 | 0.56 | 0.82 |

The full list of metrics used for the correlation analysis is:

1. minimum length (source),
2. minimum length (target),
3. maximum length (source),
4. maximum length (target),
5. median length (source),
6. median length (target),
7. mean length (source),
8. mean length (target),
9. length std (source),
10. length std (target),
11. data size in lines,
12. number of parameters,
13. number of types ($|V|$) (source),
14. number of types ($|V|$) (target),
15. number of tokens (source),
16. number of tokens (target),
17. type-token-ratio (TTR) (source),
18. type-token-ratio (TTR) (target),
19. OOV type rate (source),
20. OOV type rate (target),
21. OOV token rate (source),
22. OOV token rate (target),
23. token ratio,
24. target type-to-parameter ratio,
25. target token-to-parameter ratio,
26. distance between the TTRs of source and target = $(1 - \text{TTR}_{src}/\text{TTR}_{trg})^2$,
27. token-to-parameter ratio (i) = (median length source * median length target * num_lines) / num_parameters,
28. token-to-parameter ratio (ii) = (num_source_tokens * num_target_tokens) / num_parameters.

# G ENLARGED FIGURES FOR ALL 30 LANGUAGE DIRECTIONS (AGGREGATE RESULTS FROM ALL RUNS)

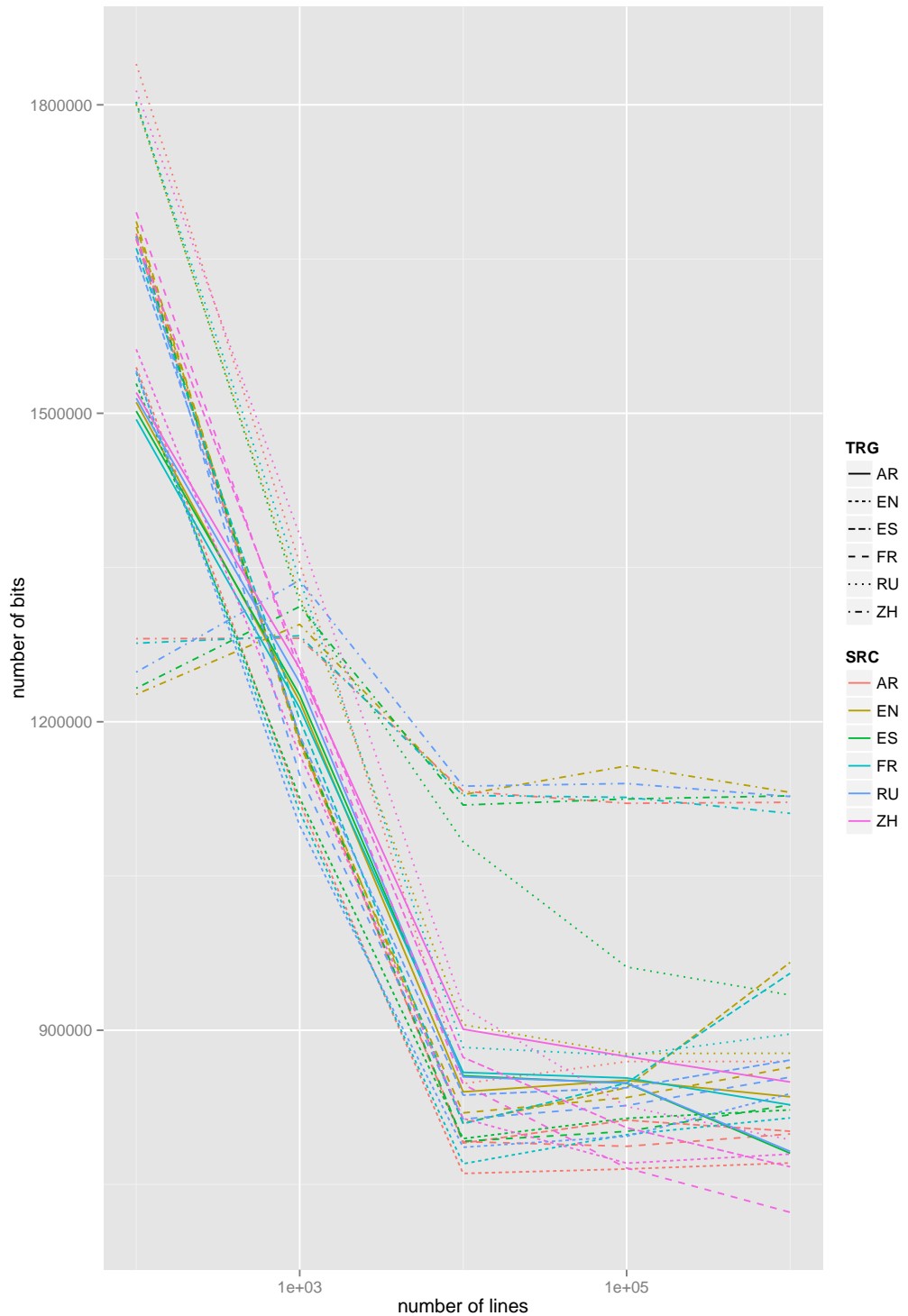

Figure 5: CHAR: character models

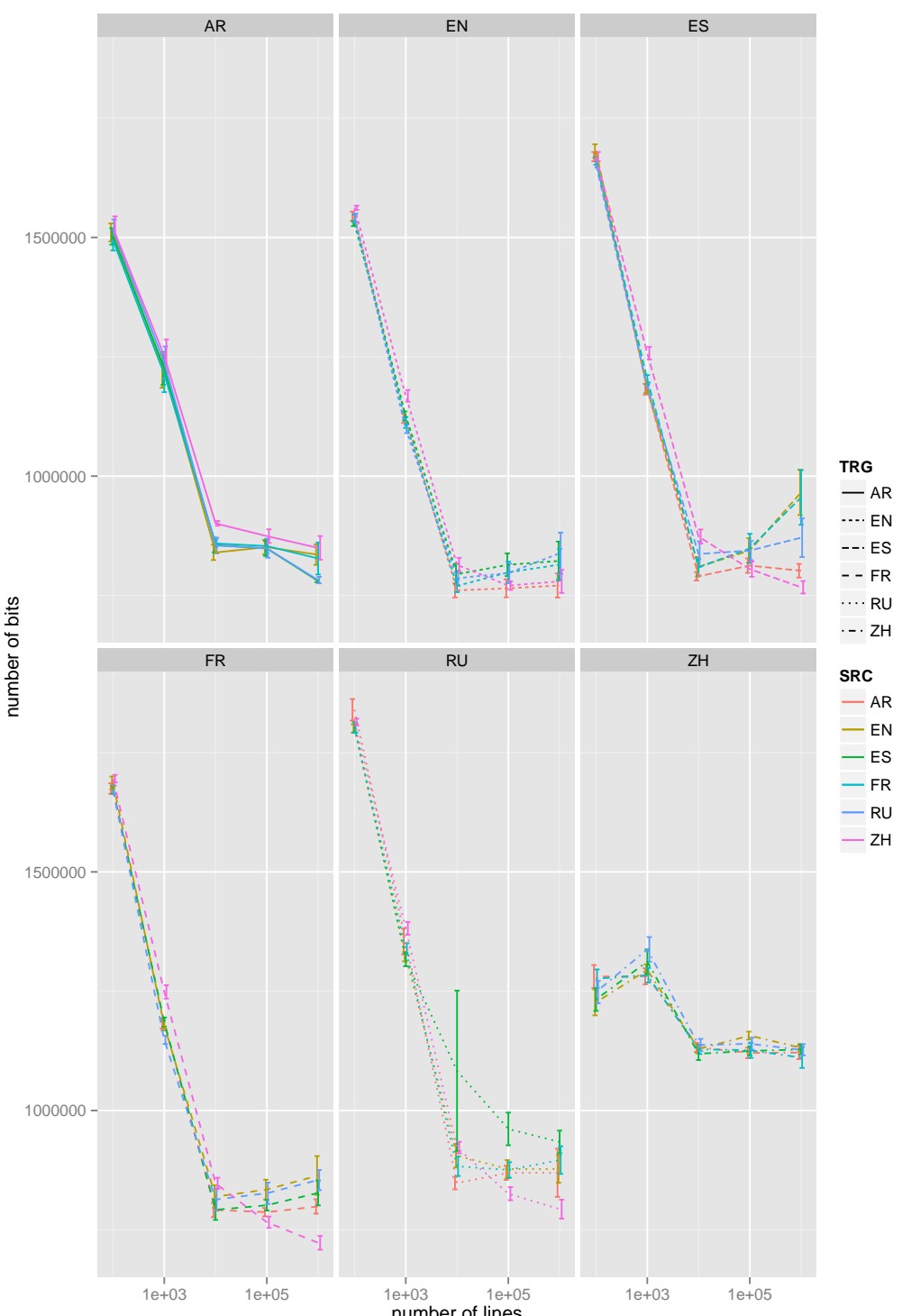

Figure 5: CHAR: character models (target language as facet)

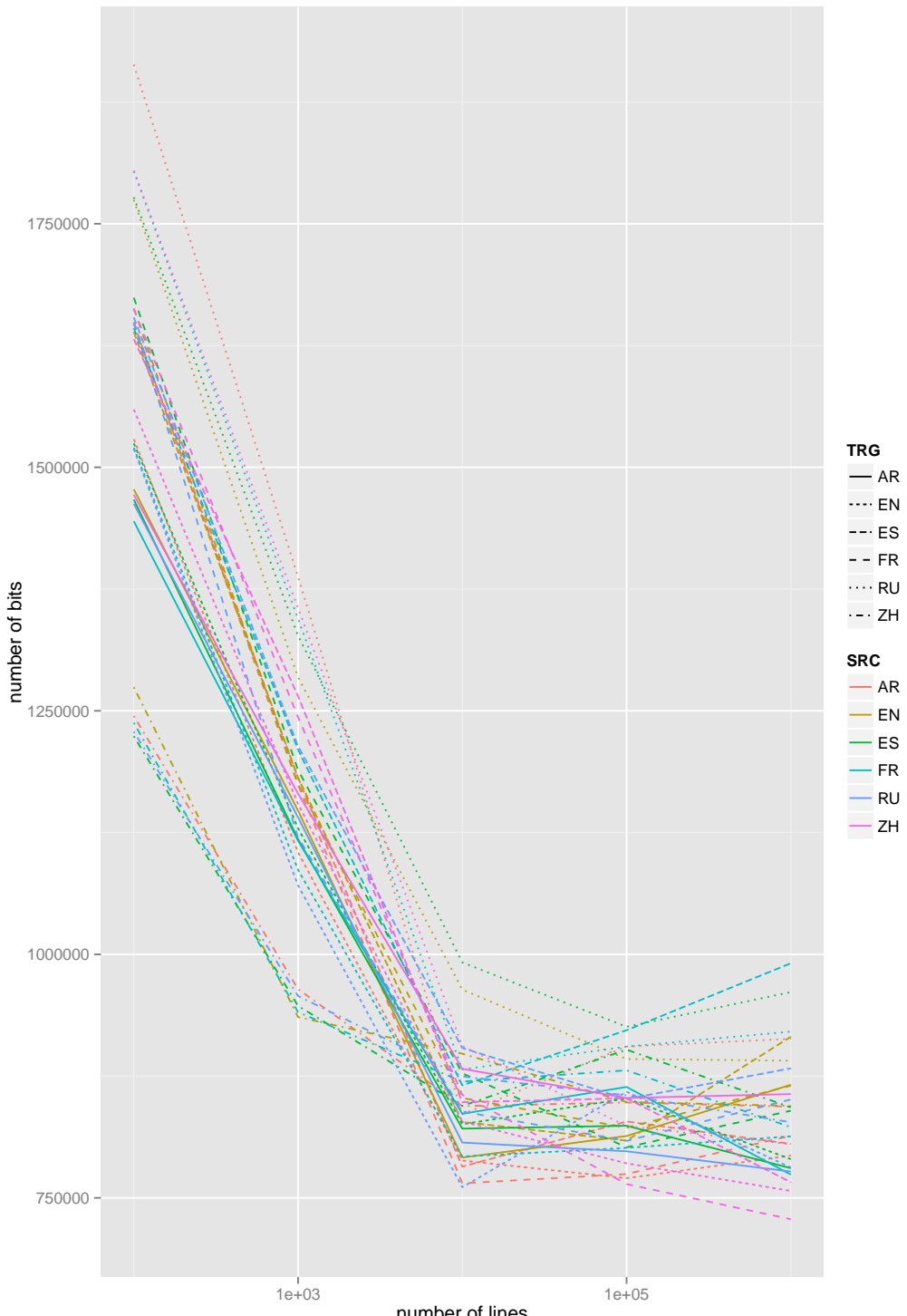

Figure 6: CHAR with Pinyin for $ZH_{trg}$

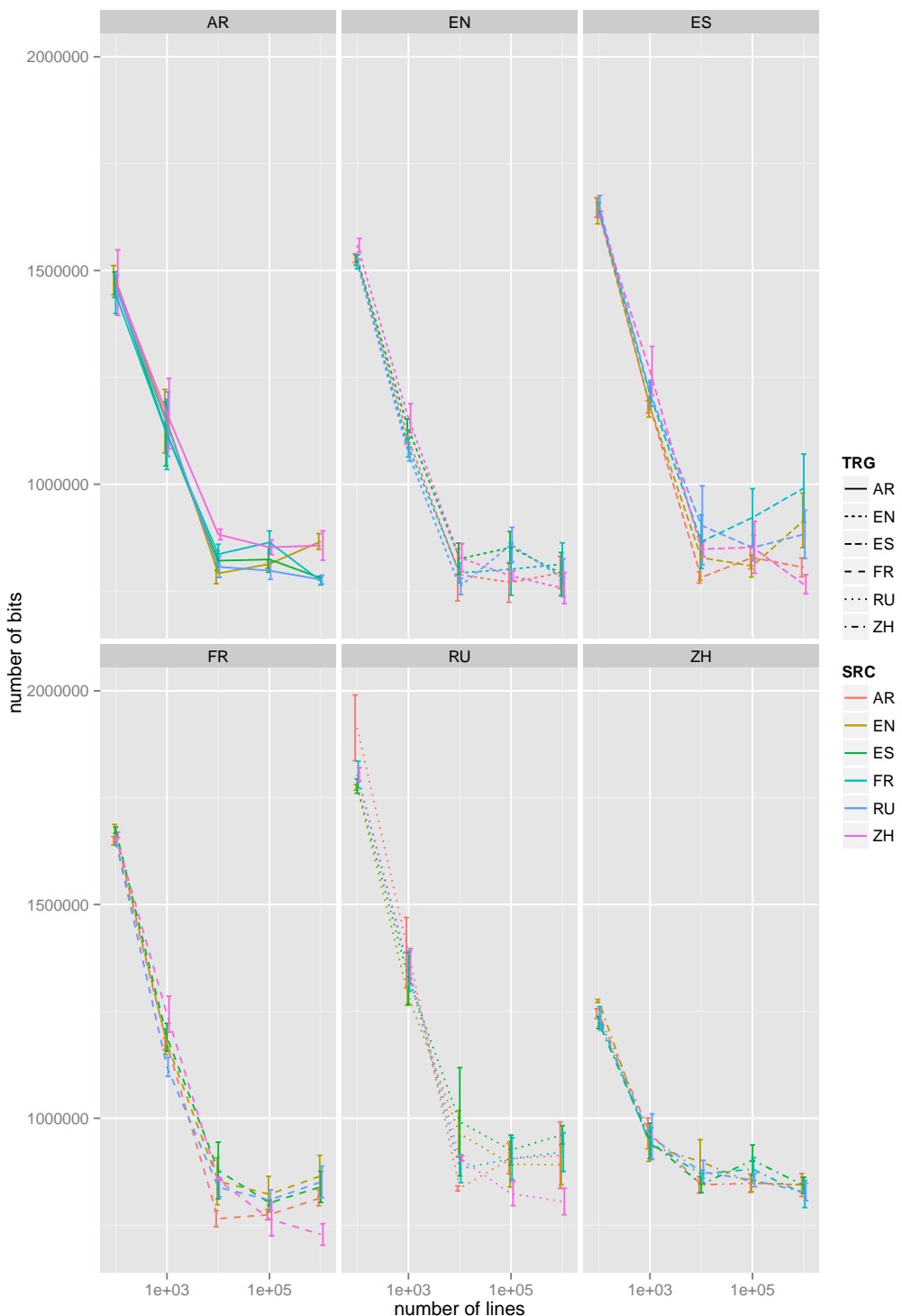

Figure 6: CHAR with Pinyin for $ZH_{trg}$ (target language as facet)

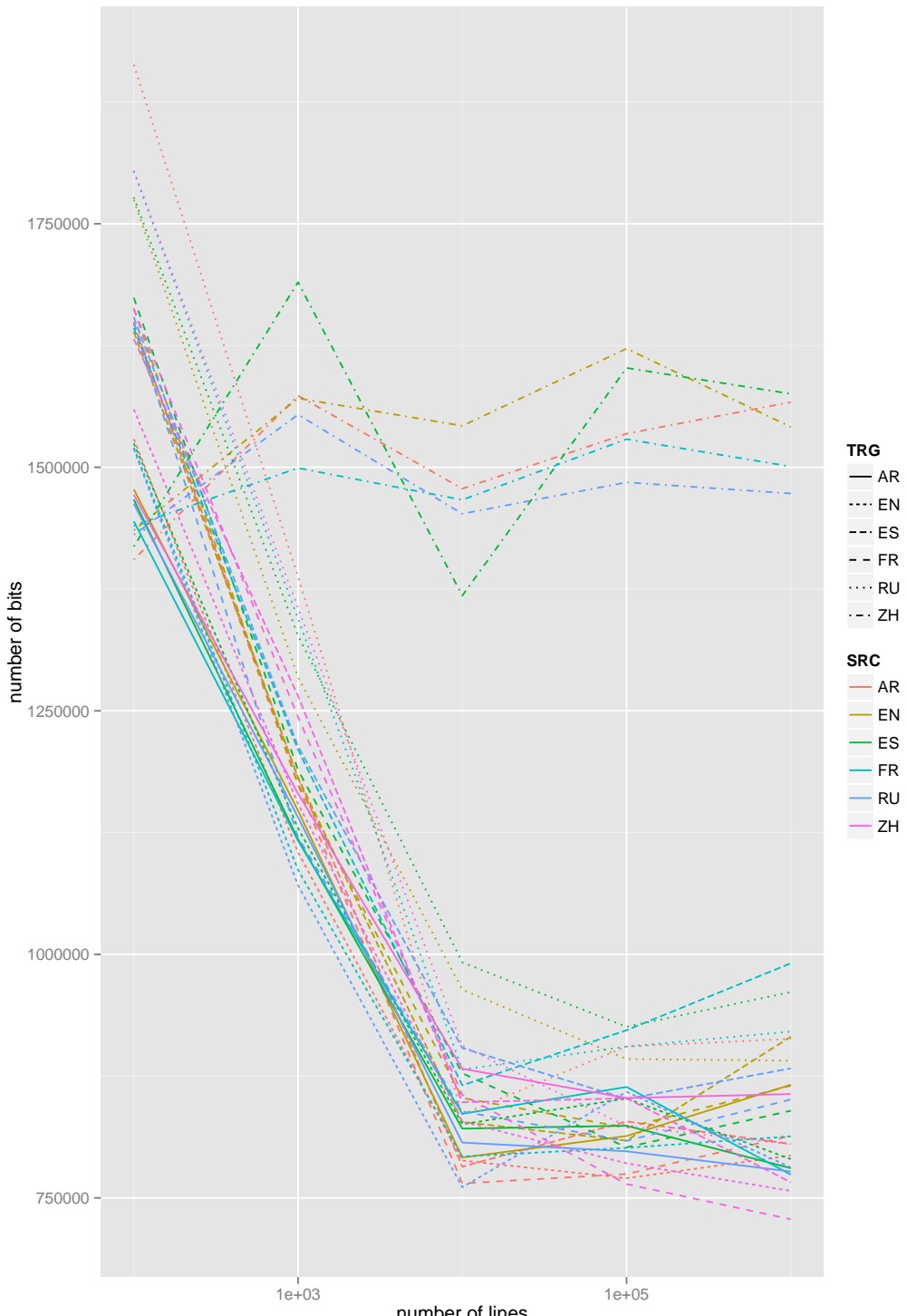

Figure 7: CHAR with Wubi for $ZH_{trg}$

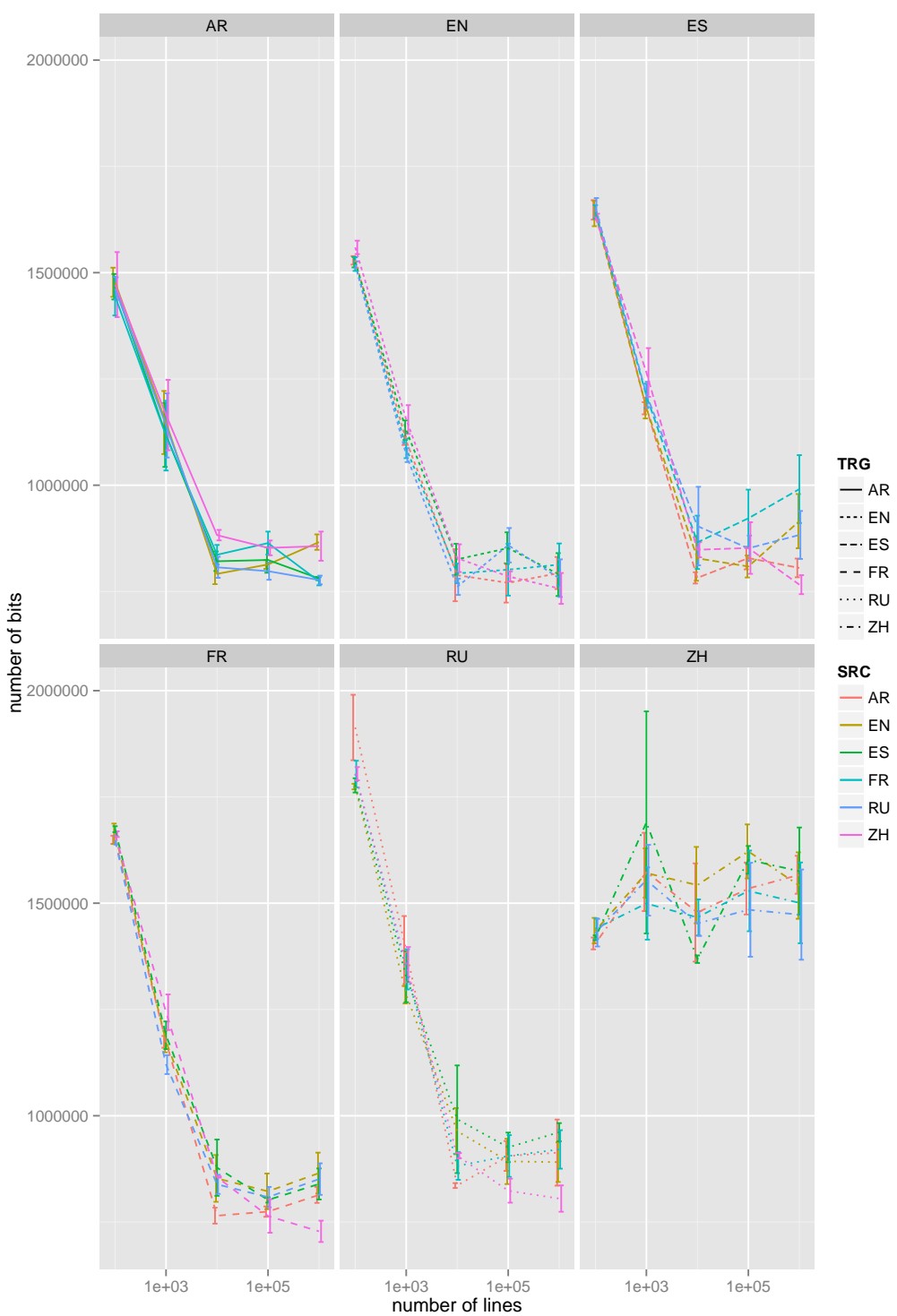

Figure 7: CHAR with Wubi for $ZH_{trg}$ (target language as facet)

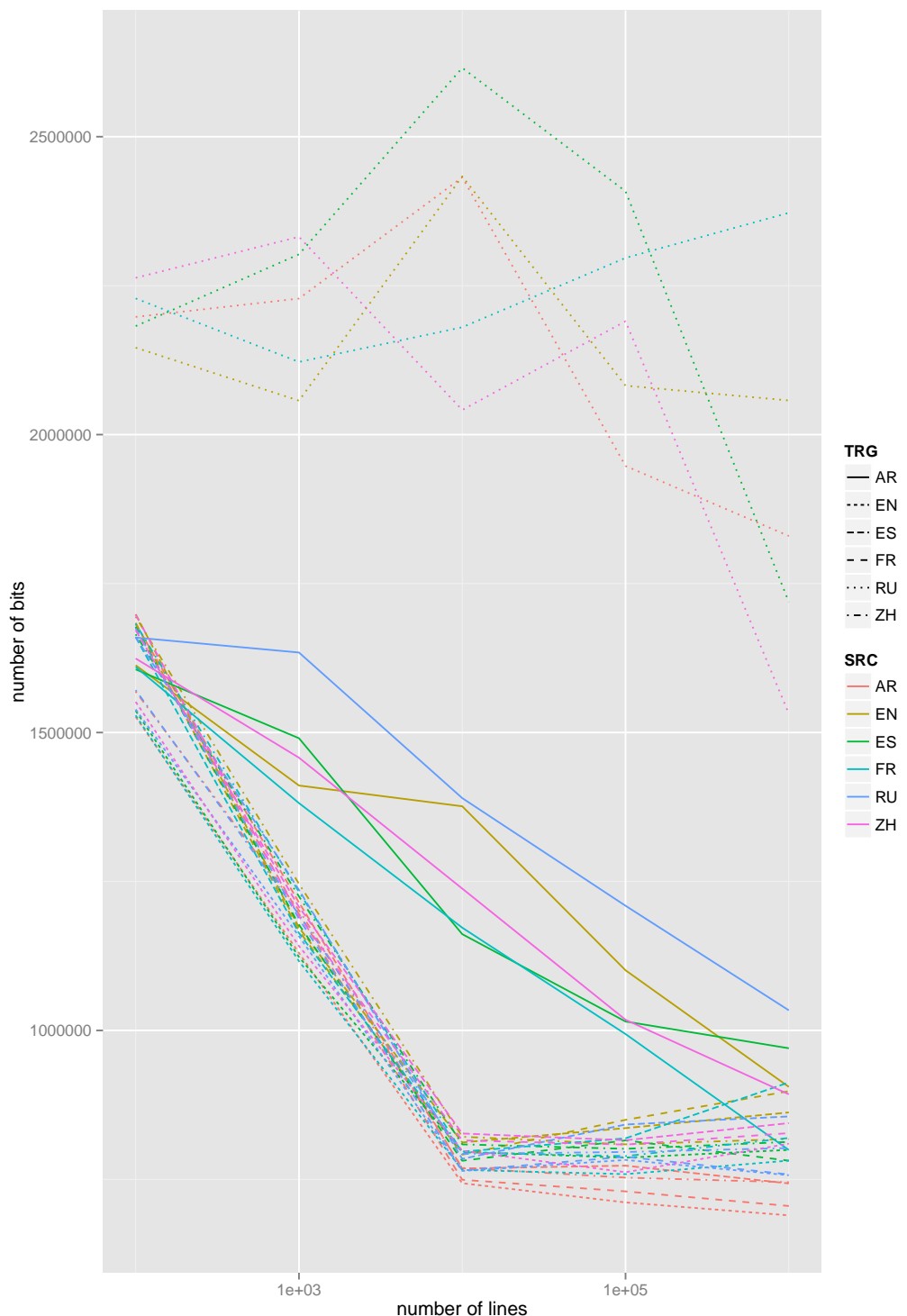

Figure 8: BYTE models with UTF-8 encoding

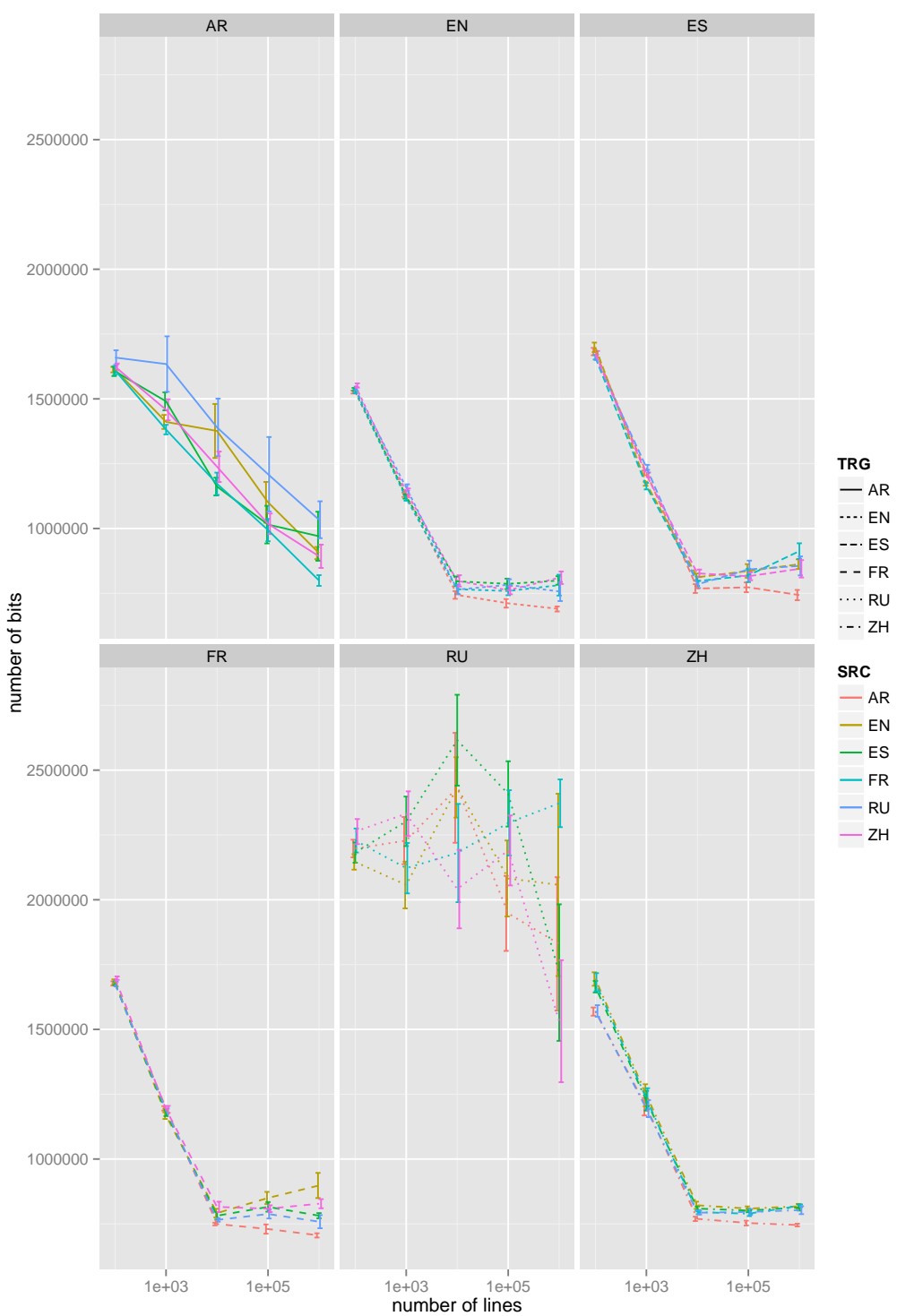

Figure 8: BYTE models with UTF-8 encoding (target language as facet)

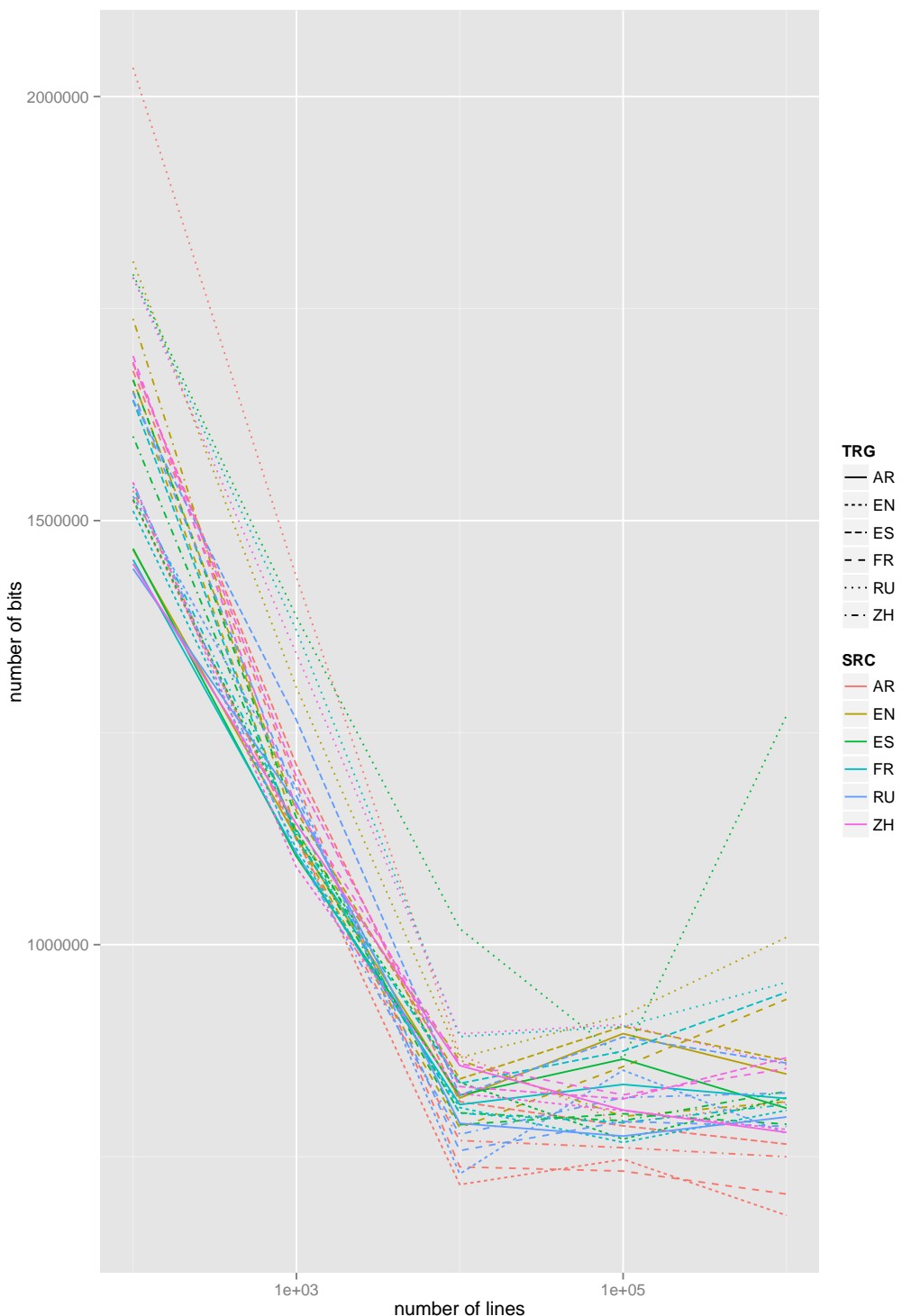

Figure 9: BYTE with $AR_{trg}$ & $RU_{trg}$ optimized with code pages 1256 & 1251 (ARRU$_{trg}$)

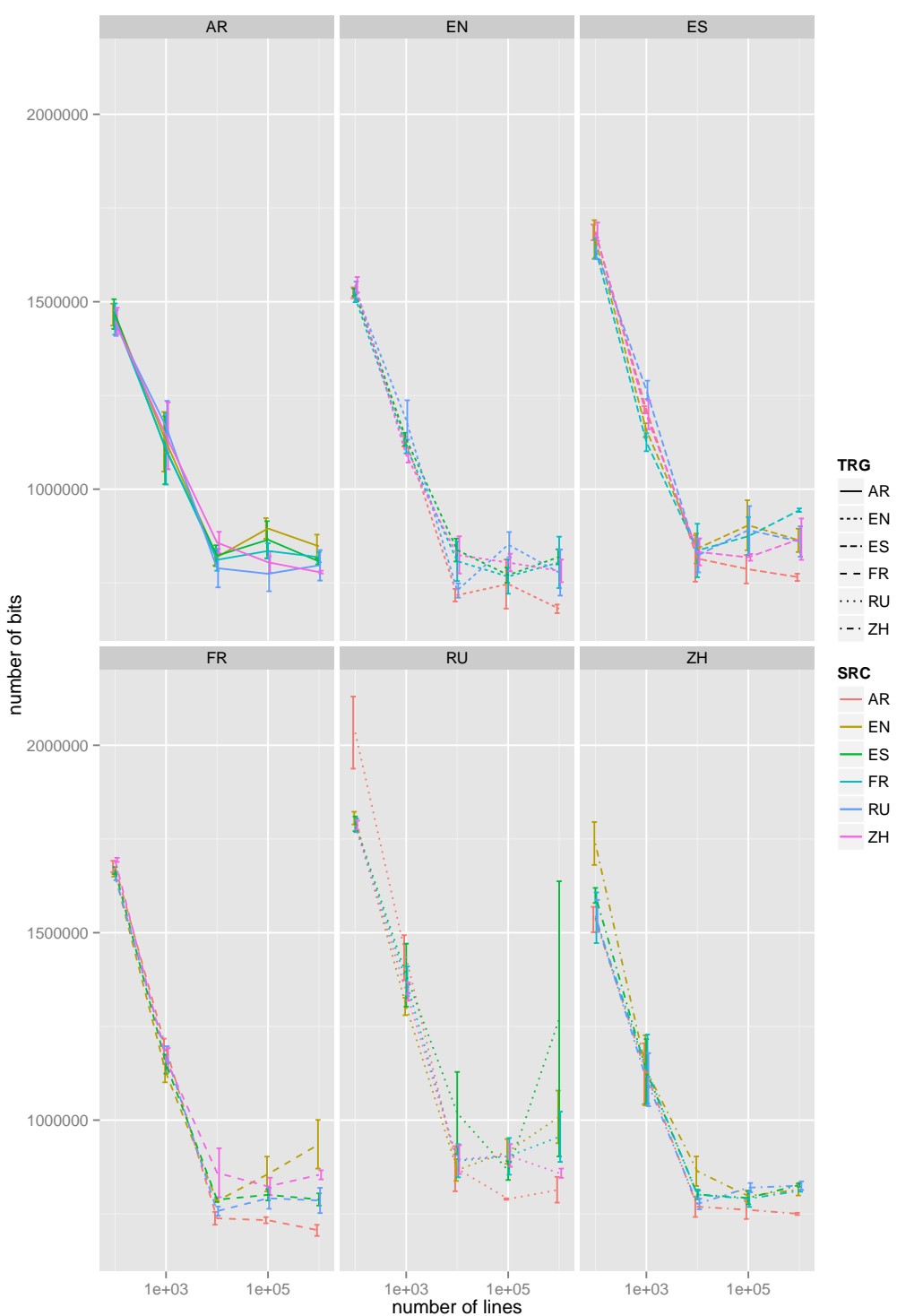

Figure 9: BYTE with $AR_{trg}$ & $RU_{trg}$ optimized with code pages 1256 & 1251 (target language as facet)

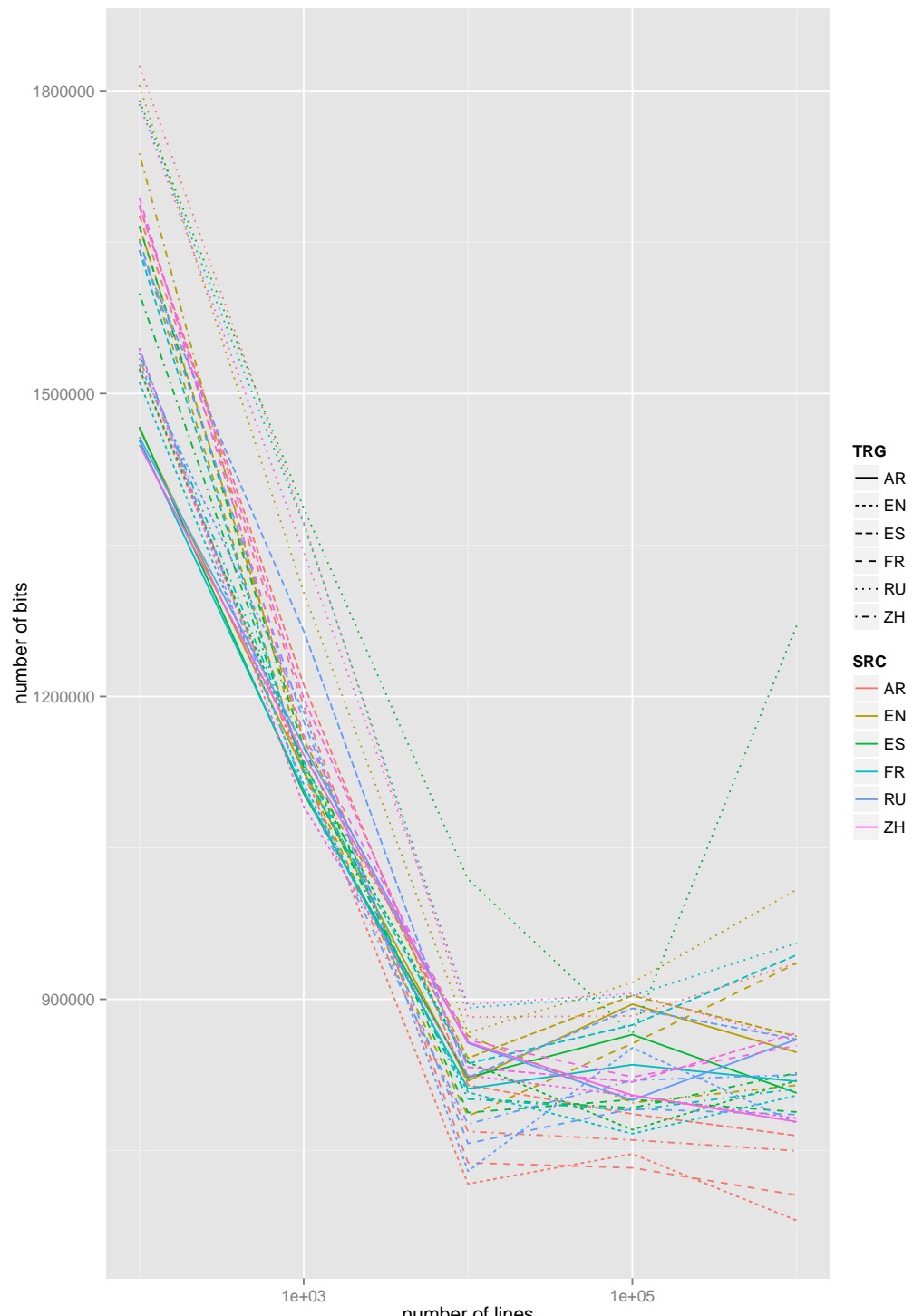

Figure 10: BYTE with directions AR-RU & RU-AR optimized on both source and target sides (ARRU$_{src,trg}$)

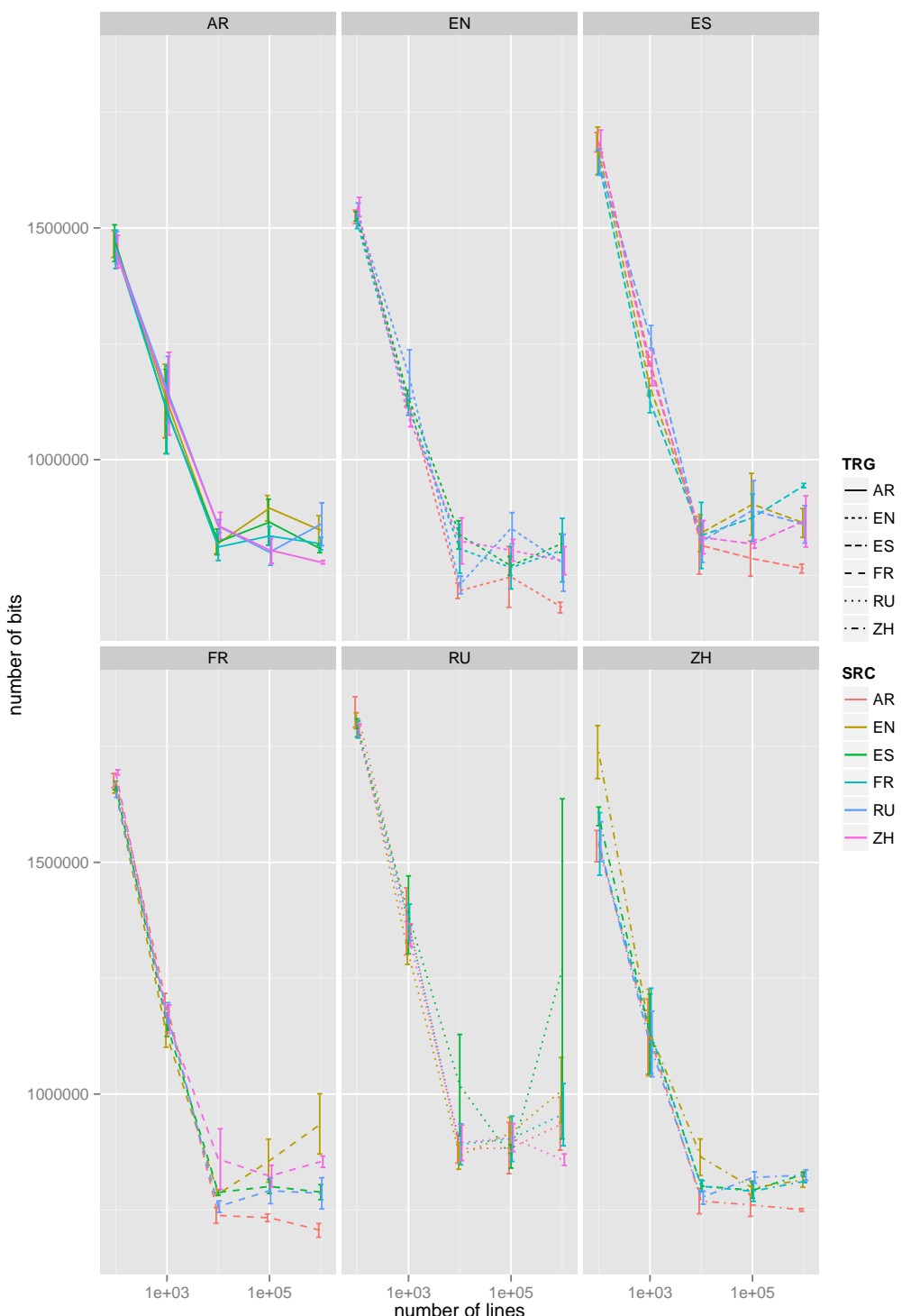

Figure 10: BYTE with directions AR-RU & RU-AR optimized on both source and target sides (target language as facet)

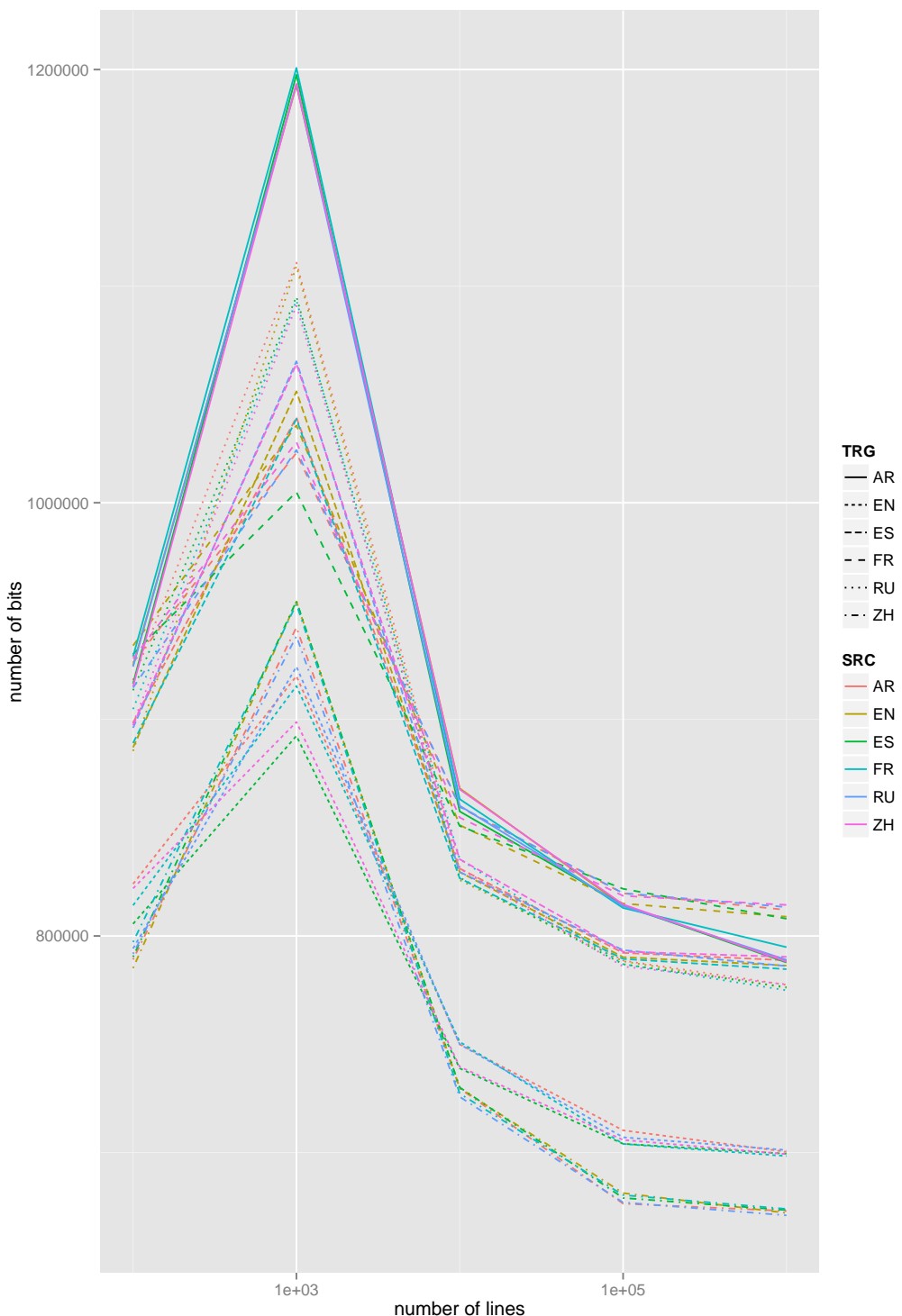

Figure 11: WORD models

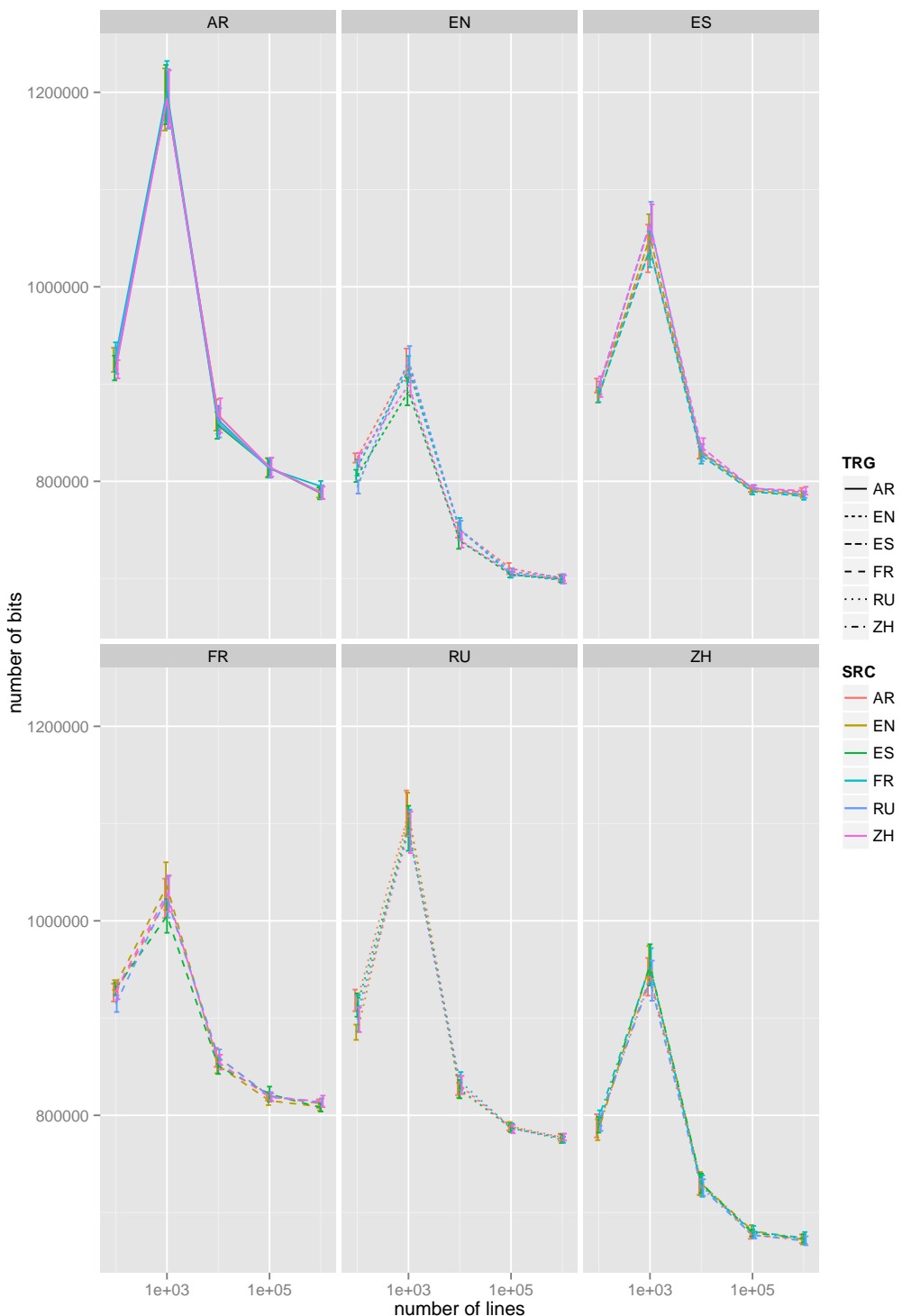

Figure 11: WORD models (target language as facet)

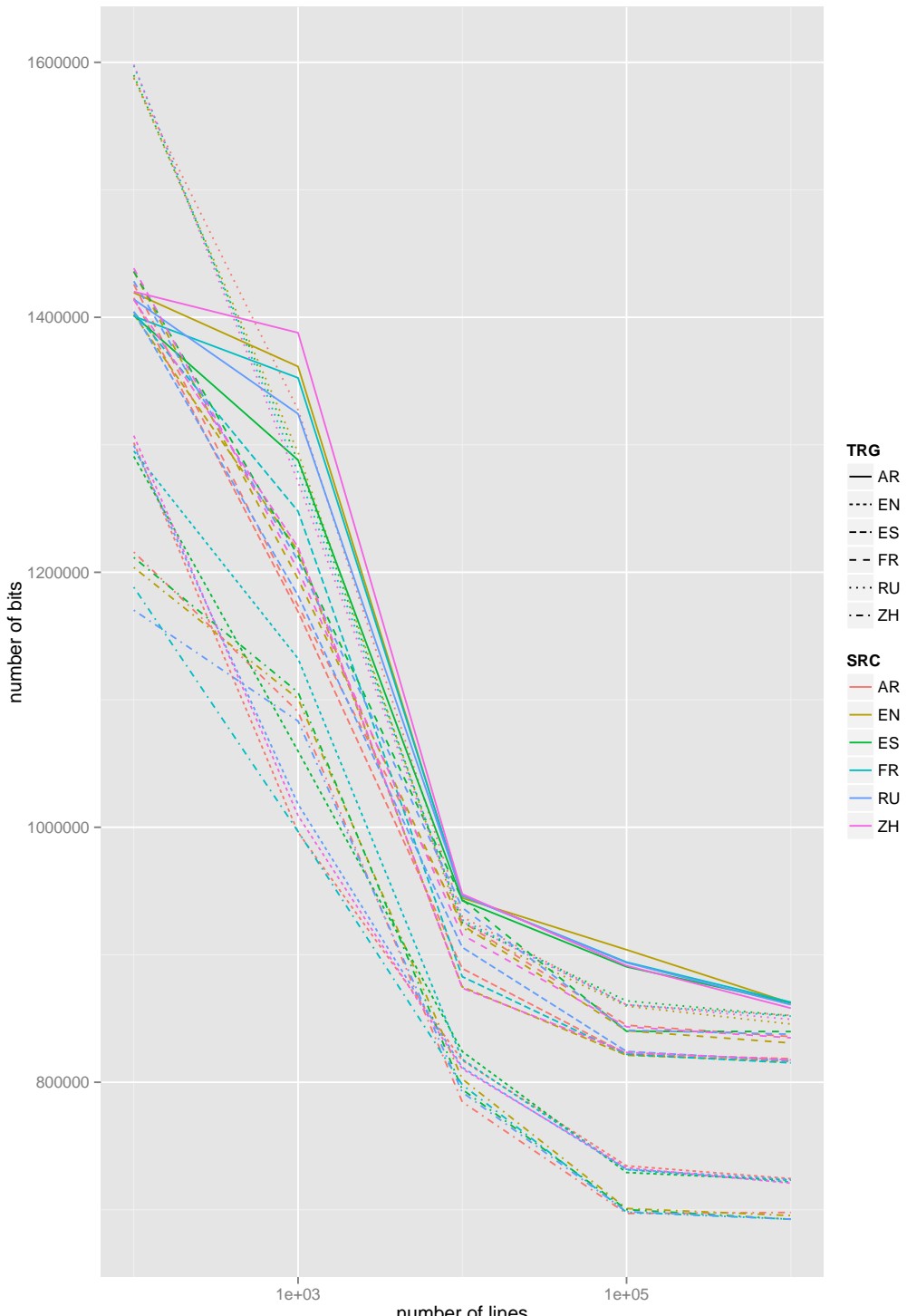

Figure 12: BPE models

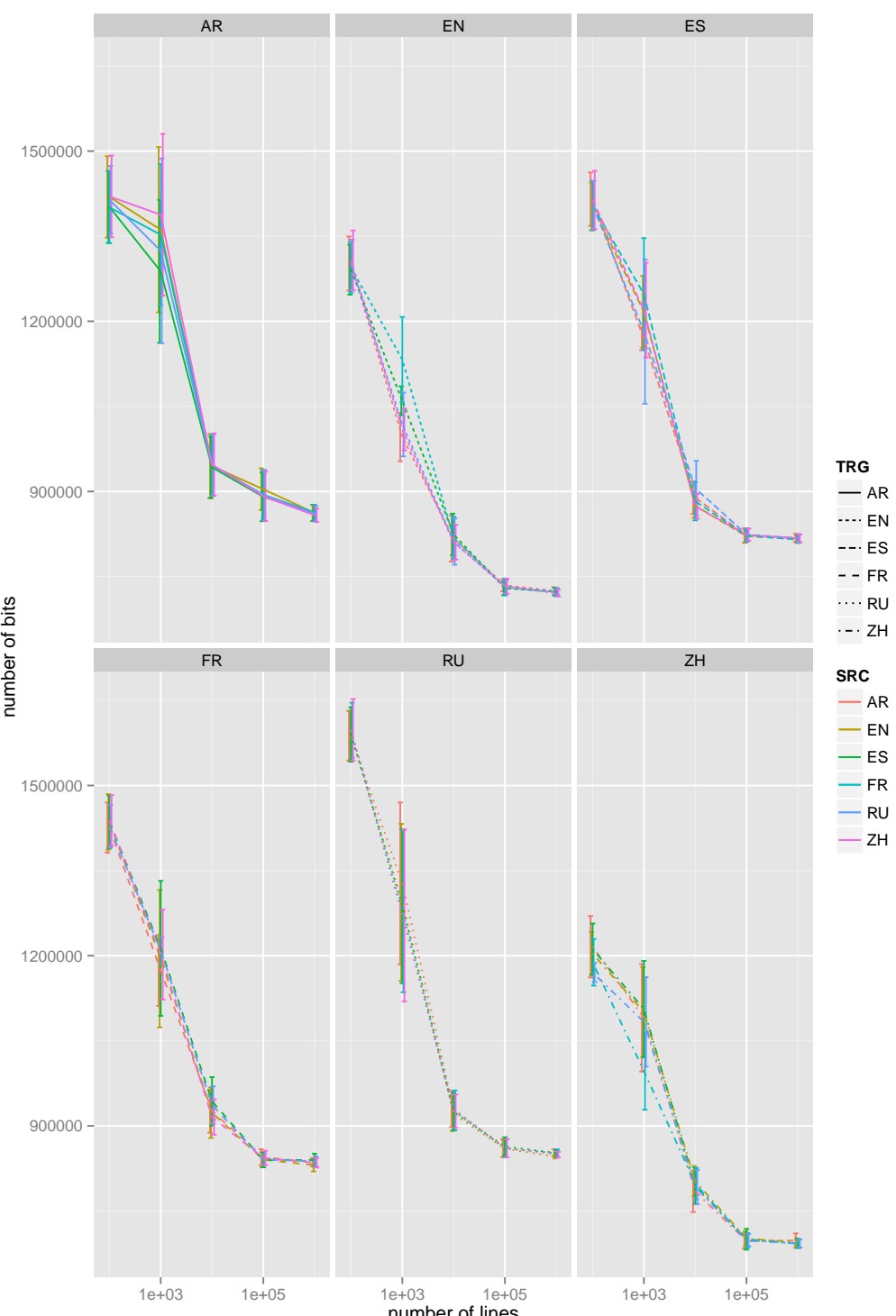

Figure 12: BPE models (target language as facet)

# H SAMPLE FIGURES FROM RUN A0, ALSO SORTED BY SOURCE LANGUAGE FOR CONTRAST

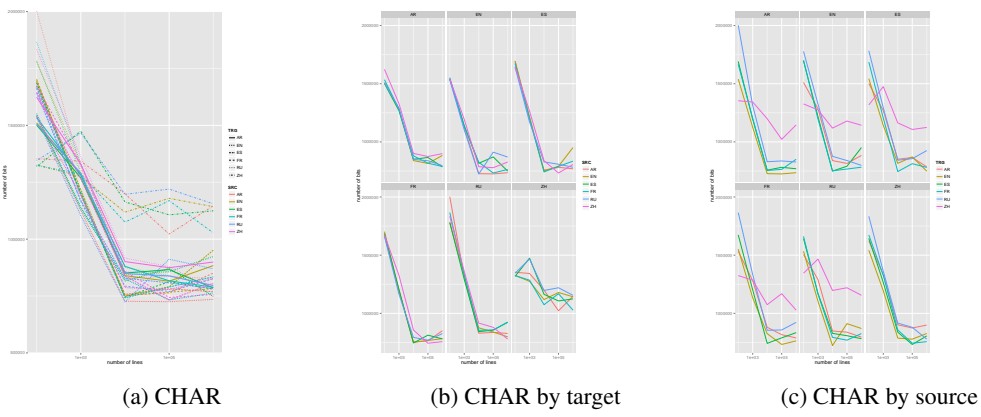

(a) CHAR        (b) CHAR by target        (c) CHAR by source

Figure 13: CHAR: character models from run A0

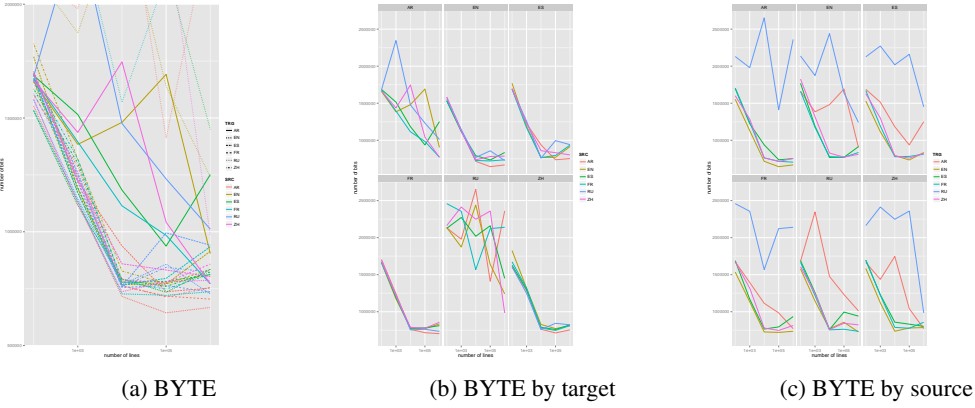

(a) BYTE        (b) BYTE by target        (c) BYTE by source

Figure 14: BYTE: byte models from run A0

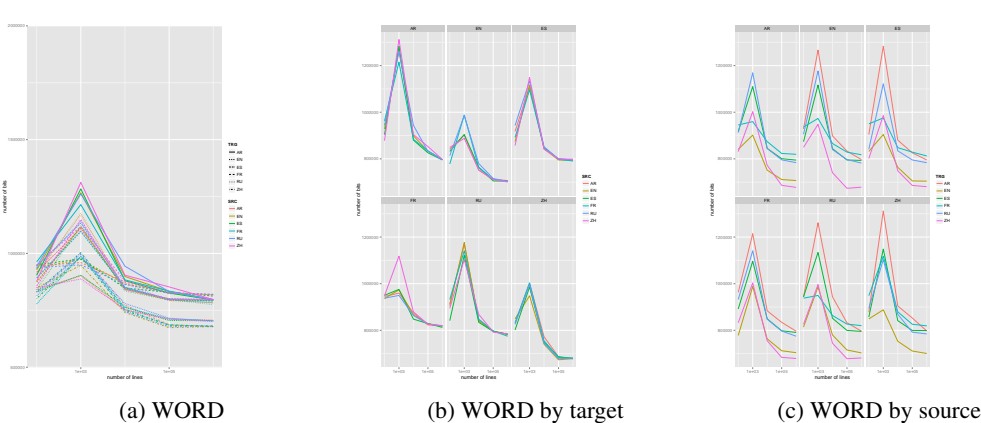

(a) WORD        (b) WORD by target        (c) WORD by source

Figure 15: WORD: word models from run A0

# I LANGUAGE PAIRS WITH SIGNIFICANT DIFFERENCES

15 (non-directional) language pairs total possible from 30 language directions, p=0.001.

| LANG PAIR | CHAR | Pinyin | Wubi | BYTE | ARRU$_t$ | ARRU$_{s,t}$ | WORD | BPE |
|---|---|---|---|---|---|---|---|---|
| AR-EN | | | | X | | | X | X |
| AR-ES | | | | | | | | |
| EN-ES | | | | | | | X | |
| AR-FR | | | | X | | | | |
| EN-FR | | | | | | | X | X |
| ES-FR | | | | | | | | |
| AR-RU | | | | X | | | | |
| EN-RU | | | | X | | X | X | X |
| ES-RU | | | | X | | | | |
| FR-RU | | | | X | | | | |
| AR-ZH | X | | X | X | | | X | X |
| EN-ZH | X | | X | | | | | |
| ES-ZH | | | X | | | | X | X |
| FR-ZH | X | | X | | | | X | X |
| RU-ZH | | | X | X | | X | X | X |

Language pairs with significant differences indicate that the 2 languages are *not* equally/similarly good or equally/similarly bad.

- Character models with ZH behave differently but the disparity can be eliminated with Pinyin.
- Byte models with AR and RU exhibit unstable performance due to length but this can be rectified with compression on the target side only (ARRU$_t$).
- Word-based models, including BPE, however, consistently favor EN and ZH (though it is more of a "mis-segmentation" for the latter, see § 3 and Appendix J) and disfavor AR and RU (as morphologically complex languages with higher OOV rates).

## J  LANGUAGE COMPLEXITY

In the words of Bentz et al. (2016):

> Languages are often compared with regard to their complexity from a computational, theoretical and learning perspective. In computational linguistics, it is generally known that methods mainly developed for the English language do not necessarily transfer well to other languages. The cross-linguistic variation in the amount of information encoded at the level of a word is, for instance, recognized as one of the main challenges for multilingual syntactic parsing (formulated as The Architectural Challenge (Tsarfaty et al., 2013)). Complexity of this kind is also found to influence machine translation: translating from morphologically rich languages into English is easier than the other way around (Koehn, 2005).

*****

Morphology is "the study of the formation and internal structure of words".
Morphemes are "the smallest meaningful units of language". (Bender, 2013)

*****

AR and RU are traditionally considered morphologically complex (see e.g. Minkov et al. (2007), Seddah et al. (2010) and proceedings of related workshops in subsequent years), and ZH lacking morphological richness (Koehn, 2005). But this definition of morphology is predicated on the notion of word, defined primarily from an alphabetic perspective. As pointed out by Zhang & Komachi (2018), "the important differences between logographic and alphabetic writing systems have long been overlooked". In logographic languages (i.e. languages with logographic scripts), there can be units within a character that carry semantic and phonetic information that have never been accounted for in the traditional practice of morphology or in the computation of morphological complexity. For example, in the comparison of different morphological complexity measures by Bentz et al. (2016), all measures studied are defined with the notion of word.[8] Yet, there is **no universally valid definition of a "word"** — the form/idea (as in, the philosophical concept) of a "word" may be there for most languages/cultures (though that is certainly also debatable), but its instantiations are different in different languages/cultures, as well as in different genres/settings within one language. The variability in the definition of word is evident in the variation in language-specific word tokenization algorithms, along with the "indeterminacy of word segmentation" or a work-in-progress status for the definition of "word" advocated by Haspelmath (2011), as well as the contested nature of wordhood, esp. for logographic languages such as ZH (see Duanmu (2017) and Li et al. (2019b) for how some ZH speakers do indeed consider a ZH character to be a word or how "word", as conventionally used in NLP, is not a native term or does not correspond with speakers' judgement).

Our results with the Transformer indicate that a notion of morphological complexity can be modeled given our word tokenization scheme, confirming that morphological complexity is only predicated on the notion of word and bounded within the word level, and orthogonal to the performance of character or byte models. That is, unless word-based segmentation has been applied, there is no reason to attribute crosslinguistic performance disparity to differences in morphological complexity. In fact, on the character and byte level, we were able to achieve performance without disparity. Hence **disparity is not a necessary condition but an expectation that has been in mutual reinforcement with our practice of word segmentation, while the definitions of "morphological complexity" and "word" are in a circular dependency with each other.**

In this paper, we *re*solve language complexity, more specifically that of morphological complexity, in the context of computing through CLMing with the Transformer, in that we *explain away* the representation granularities and criteria relevant for such calculation.

TLDR: Up to the point of our taking up the subject of language complexity in this paper, there has been not a rigorous definition of "language complexity". Conventionally, "language complexity" is synonymous to "linguistic complexity" (with the tradition of "linguistics" being primarily word-

---

[8]An exception could be that of the type/token ratio (TTR). One could imagine applying TTR on the character level for ZH, and that would be indicative of its morphological richness on the character level. However, that has thus far never been practiced or recognized in NLP.

based), and people just assume linguistic complexity, e.g. morphological/syntactic complexity, to be intrinsic and necessary in languages (across representation levels). Our findings show that linguistic complexity is relative to the representation granularity, i.e. since morphology is based on words, it is bounded to the word level.

*****

An alternative perspective, with finer prints:

We have also developed a more rigorous interpretation. We take on the definition of "language complexity" as one that is related to the statistical attributes of languages. We assume and define *solving* as the elimination of statistically significant performance disparity.

In larger (6-layer) models, and according to the conventional definition of "language" — i.e. language as a whole, we solved language complexity with compression of AR and RU in byte representations. In smaller (1-layer) models, one can think of the situation as: i) no complexity has been modeled by the Transformer hence there is nothing to solve, or ii) there is no complexity between these languages to begin with, or iii) the Transformer solved the complexity.

With respect to each representation level/granularity in the larger models:

- BYTE: one can think of us as having solved complexity with byte representations or with 1-layer models — for these 6 languages empirically. Theoretically, there could be languages with longer sequence lengths than RU and AR, in those cases, we don't claim to have solved the matter empirically but only resolved it conceptually. But this is the most that anyone could do at the moment, as there is no relevant parallel data available.

- CHARACTER: one can think of us as having solved it via bytes or 1-layer models. Whether we can be considered to have solved it via Pinyin for ZH depends on whether the evaluator accepts decomposition into a *phonetic representation only* qualifies as a solution for the ZH language.

- WORD: one can think of us as having solved it via bytes or 1-layer models. It is not possible to solve it strictly within the word level without creating word segmentation criteria that would be unrelatable to native speakers. And since "word" is exclusively a human concept, we must either claim that a universal solution is undefined or undefinable for computing, or retreat to a unit that is the greatest common factor crosslinguistically. Since some ZH speakers consider ZH characters as words, we return to the character-level solution.

It is beyond the scope of our paper to solve the qualitative disparity on the word level. However, we do advocate a more inclusive evaluation and critical reflection on the possibility of discontinuing the usage of "word" as such a non-technical term biases against both "morphologically complex" and "morphologically simple" languages. The world of languages in written form can be divided into those with logographic scripts and those with (phonetic) alphabetic ones, with the unit of character being the greatest common factor of them all, from the human perspective. For technical processing, esp. for fair multilingual sequence-to-sequence modeling with the Transformer, we recommend measures that are more standardized, such as those based on bytes or characters. There is room for improvement in the design of character encoding that complements the statistical profiles, e.g. with relative rank in sequence length, of different languages. We believe there is crosslinguistic systematicity on the character level to be leveraged.

One's readiness to accept this as a solution to language complexity can be a subjective matter. One may insist that language complexity be solved exclusively with monolingual LMing (which lies outside the scope of the present work), instead of being confounded with the logic of one language being conditional on another. One may also object to the idea of (re-)solving morphological complexity being equivalent to or leading to solving language complexity as a whole, for there could also be e.g. syntactic complexity (although as substantial "information concerning syntactic units and relations is expressed at word level" in morphologically rich languages (Tsarfaty et al., 2010), the boundary between morphology and syntax is less distinct for some languages than others (Haspelmath, 2011)). If, however, our results could be extended, we wonder if syntactic complexity could be due to our sentence segmentation or a combination of word and sentence segmentation practice. That we leave for future work for those who are interested in the topic.

## K  SAMPLE-WISE DOUBLE DESCENT (DD)

### K.1  OUR EXPERIMENTAL FRAMEWORK ON DD DATASETS FROM (NAKKIRAN ET AL., 2020)

Text experiments from previous work reporting sample-wise DD involved words (Belkin et al., 2019) and BPEs (Nakkiran et al., 2020).

We applied our experimental framework — by testing data points with $10^n$ lines — on the datasets reported in (Nakkiran et al., 2020) to exhibit DD. WMT'14[9] EN-FR was reported to demonstrate model-wise DD and IWSLT'14 (Cettolo et al., 2012) DE-EN model-wise and sample-wise DD. We downloaded and prepared the data with scripts[10] from the FAIRSEQ Toolkit (Ott et al., 2019). The WMT data was preprocessed with 40,000 BPE operations and IWSLT 10,000. Our focus is on sample-wise DD and hence our goal was to see if the spike at $10^3$ we observed with the UN data would apply also to these datasets. We used the same training regime[11] with the Transformer and Adam on SOCKEYE as before and tested both language directions on the entirety of both datasets, with no subsampling. For the IWSLT dataset, we tested data sizes with $10^2 - 10^5$ lines, then at $160,239$ as that is the total number of lines available. For the WMT dataset, we tested from $10^2$ to $10^7$, then at $35,762,532$.

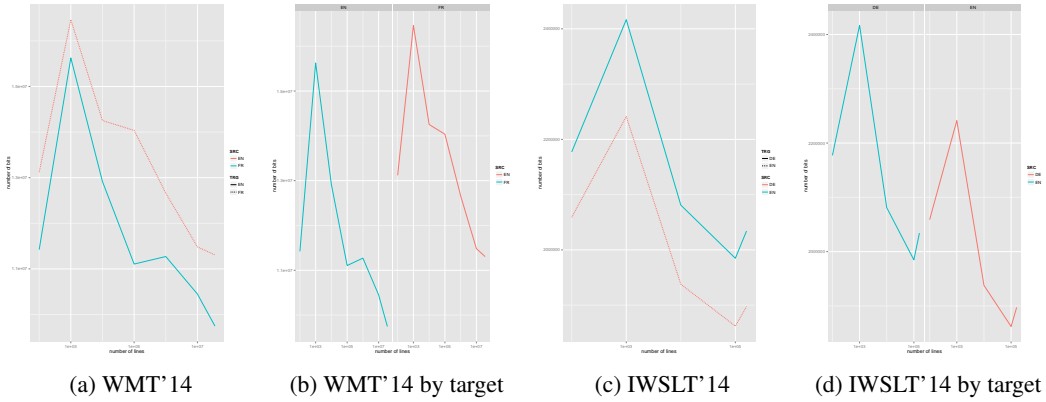

| (a) WMT'14 | (b) WMT'14 by target | (c) IWSLT'14 | (d) IWSLT'14 by target |

Figure 16: WMT'14 EN-FR and FR-EN and IWSLT'14 DE-EN and EN-DE: sample-wise DD shown at $10^3$

Table 2: Target-Train-Token-to-Parameter ratio (TTT2P ratio) for WMT'14 EN-FR and FR-EN

|  | Number of lines | | | | | | |
|---|---|---|---|---|---|---|---|
|  | 100 | 1,000 | 10,000 | 100,000 | 1,000,000 | 10,000,000 | 35,762,532 |
| EN: num train tokens | 3,248 | 33,768 | 313,154 | 3,123,129 | 30,852,455 | 308,640,462 | 1,174,344,513 |
| FR: num train tokens | 3,548 | 36,507 | 339,803 | 3,414,959 | 33,865,679 | 343,344,536 | 1,327,817,765 |
| EN-FR num params | 45,609,474 | 51,039,363 | 62,871,584 | 75,630,304 | 85,210,037 | 108,226,335 | 111,417,633 |
| TTT2P ratio | 0.000078 | 0.000715 | 0.005405 | 0.045153 | 0.397438 | 3.172468 | 11.917483 |
| FR-EN num params | 45,540,219 | 50,692,575 | 61,916,891 | 74,547,874 | 83,936,258 | 107,378,859 | 111,399,165 |
| TTT2P ratio | 0.000071 | 0.000666 | 0.005058 | 0.041894 | 0.367570 | 2.874313 | 10.541771 |

This shows that the effect we reported in § 5 also holds on these datasets: "the **ratio of target training token count to number of parameters** falls into O($10^{-4}$) for $10^2$ lines, O($10^{-3}$) at $10^3$, O($10^{-2}$) at $10^4$, and O($10^{-1}$) for $10^5$ lines and so on".

---

[9] http://www.statmt.org/wmt14/translation-task.html

[10] https://github.com/pytorch/fairseq/blob/master/examples/translation/prepare-wmt14en2fr.sh  and  https://github.com/pytorch/fairseq/blob/master/examples/translation/prepare-iwslt14.sh

[11] max-seq-len 300; checkpoint-frequency 4000 except for cases where 50 epochs would be reached before the first checkpoint: 400 for $10^2$ lines and 3450 for $10^3$ lines.

Table 3: Target-Train-Token-to-Parameter ratio (TTT2P ratio) for IWSLT'14 DE-EN and EN-DE

|  | Number of lines | | | | |
|---|---|---|---|---|---|
|  | 100 | 1,000 | 10,000 | 100,000 | 160,239 |
| DE: num train tokens | 2,874 | 27,675 | 253,757 | 2,519,534 | 4,035,591 |
| EN: num train tokens | 2,739 | 26,416 | 245,659 | 2,461,879 | 3,949,114 |
| DE-EN num params | 45,297,348 | 49,410,683 | 53,639,825 | 55,189,376 | 55,428,584 |
| TTT2P ratio | 0.000060 | 0.000535 | 0.004580 | 0.044608 | 0.071247 |
| EN-DE num params | 45,405,078 | 49,809,797 | 54,300,056 | 56,245,643 | 56,564,366 |
| TTT2P ratio | 0.000063 | 0.000556 | 0.004673 | 0.044795 | 0.071345 |

## K.2 Token-to-parameter ratio for non-neural monolingual LMs

We experimented also on KenLM (Heafield, 2011; Heafield et al., 2013), a non-neural LM with modified Kneser-Ney smoothing (Kneser & Ney, 1995; Chen & Goodman, 1999), on our dataset A and found that on the word level, such a spike (or a hump) is common across all languages, see Figure 17. The target-token-to-parameter ratio is under 1 for most of these smaller data sizes. This seems related to the analytical findings in Opper et al. (1990) where the pseudo-inverse solution to a simple learning problem was shown to exhibit non-monotonicity, with the peak exactly as the ratio of data to parameters ($\alpha$) approaches 1.

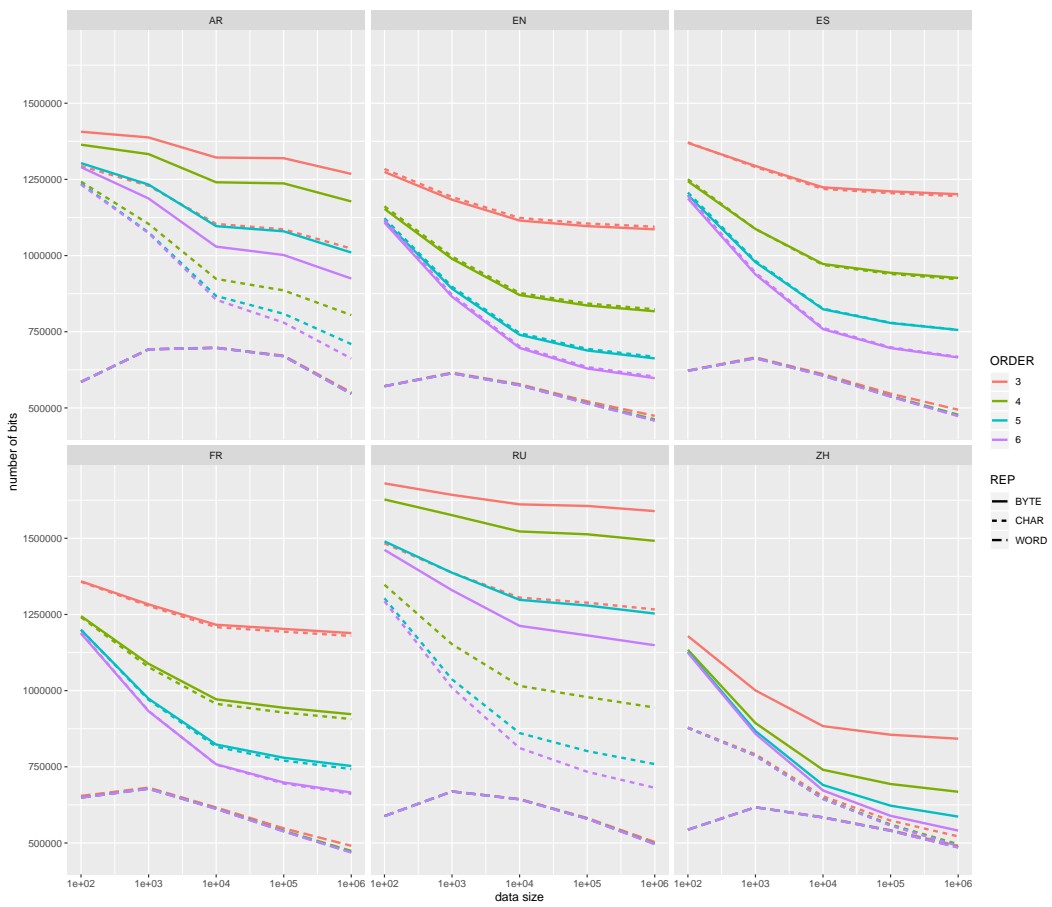

Figure 17: Kneser-Ney (monolingual) n-gram LMs on the same data (A) used for our neural CLMs

The number of parameters of a k-gram model is the number of unique n-grams, $1 \leq n \leq k$. Table 4 shows the ratios for our trigram model (all n-gram models of higher order exhibit the same effect).

On word level, where the function of number of bits to data size is not always monotonic, we observe less of a monotonic development whenever the token-to-parameter ratio is smaller than 1. This is more notably shown in the first 4 sizes in AR with a hump-like curve before the performance improves at $10^6$. This is different from the sharper descent for ES and FR, where only the first two data sizes have a non-monotonic relationship and a token-to-parameter ratio less than 1. Taking the token-to-parameter ratio as a rough proxy for over- ($< 1$) and under-parameterization ($> 1$), this can be seen as an instance of non-monotonicity with respect to data size in the "critical regime", i.e. when the model transitions from being (heavily) over- to under-parameterized (Belkin et al., 2019; Nakkiran, 2019).

**A remark on modeling with finer granularity**     Our KenLM results show the performance of bytes and characters is not on par with that of words with non-neural algorithms. NNs/DL has enabled much progress in this regard.

Table 4: Token-to-parameter ratios on non-neural monolingual trigram LMs

| | lang_numlines | num_tokens | \|unigrams\| | \|bigrams\| | \|trigrams\| | num_params | tokens/params |
|---|---|---|---|---|---|---|---|
| CHAR | AR_100 | 9079 | 85 | 925 | 2894 | 3904 | 2.325563525 |
| | AR_1000 | 123832 | 110 | 1577 | 8592 | 10279 | 12.04708629 |
| | AR_10000 | 1083517 | 152 | 3216 | 21479 | 24847 | 43.60755826 |
| | AR_100000 | 10625047 | 179 | 5114 | 44251 | 49544 | 214.4567859 |
| | AR_1000000 | 102064230 | 242 | 8517 | 90353 | 99112 | 1029.786807 |
| | EN_100 | 11730 | 78 | 806 | 2532 | 3416 | 3.433840749 |
| | EN_1000 | 159444 | 84 | 1215 | 5808 | 7107 | 22.43478261 |
| | EN_10000 | 1344001 | 125 | 2532 | 17181 | 19838 | 67.7488154 |
| | EN_100000 | 13132862 | 170 | 4231 | 36104 | 40505 | 324.2281694 |
| | EN_1000000 | 123491871 | 247 | 7126 | 70406 | 77779 | 1587.727677 |
| | ES_100 | 12374 | 87 | 781 | 2398 | 3266 | 3.788732394 |
| | ES_1000 | 171104 | 93 | 1210 | 5045 | 6348 | 26.95400126 |
| | ES_10000 | 1484804 | 117 | 2534 | 15462 | 18113 | 81.97449346 |
| | ES_100000 | 14549703 | 176 | 4261 | 33554 | 37991 | 382.9776263 |
| | ES_1000000 | 138596036 | 257 | 7217 | 67280 | 74754 | 1854.02836 |
| | FR_100 | 12456 | 89 | 836 | 2610 | 3535 | 3.523620934 |
| | FR_1000 | 179048 | 97 | 1259 | 5711 | 7067 | 25.33578605 |
| | FR_10000 | 1490983 | 133 | 2607 | 16282 | 19022 | 78.38203133 |
| | FR_100000 | 14528593 | 178 | 4390 | 35051 | 39619 | 366.707716 |
| | FR_1000000 | 138049189 | 259 | 7353 | 69522 | 77134 | 1789.732012 |
| | RU_100 | 11980 | 98 | 952 | 3051 | 4101 | 2.921238722 |
| | RU_1000 | 168156 | 111 | 1415 | 7106 | 8632 | 19.48053753 |
| | RU_10000 | 1436078 | 163 | 3506 | 20478 | 24147 | 59.4723154 |
| | RU_100000 | 14151728 | 190 | 5737 | 44071 | 49998 | 283.0458818 |
| | RU_1000000 | 134706120 | 263 | 10186 | 94975 | 105424 | 1277.755729 |
| | ZH_100 | 3318 | 605 | 2036 | 2634 | 5275 | 0.6290047393 |
| | ZH_1000 | 42572 | 1239 | 13266 | 24811 | 39316 | 1.082816156 |
| | ZH_10000 | 372003 | 2270 | 68178 | 175730 | 246178 | 1.511113909 |
| | ZH_100000 | 3659617 | 3403 | 241215 | 968852 | 1213470 | 3.015828162 |
| | ZH_1000000 | 34672612 | 4888 | 611213 | 3977112 | 4593213 | 7.548661906 |
| BYTE | AR_100 | 16655 | 76 | 320 | 1163 | 1559 | 10.68313021 |
| | AR_1000 | 227163 | 98 | 539 | 2070 | 2707 | 83.91688216 |
| | AR_10000 | 1985014 | 133 | 1616 | 5974 | 7723 | 257.0262851 |
| | AR_100000 | 19487689 | 148 | 2844 | 14274 | 17266 | 1128.674215 |
| | AR_1000000 | 186171180 | 165 | 5219 | 40507 | 45891 | 4056.812447 |
| | EN_100 | 11731 | 79 | 807 | 2533 | 3419 | 3.431120211 |
| | EN_1000 | 159449 | 85 | 1219 | 5812 | 7116 | 22.40711074 |
| | EN_10000 | 1345771 | 130 | 2527 | 17139 | 19796 | 67.98196605 |
| | EN_100000 | 13158948 | 154 | 3971 | 34985 | 39110 | 336.4599335 |
| | EN_1000000 | 123705128 | 169 | 6422 | 66606 | 73197 | 1690.030029 |
| | ES_100 | 12629 | 88 | 766 | 2414 | 3268 | 3.864443084 |
| | ES_1000 | 175286 | 94 | 1146 | 4901 | 6141 | 28.54355968 |
| | ES_10000 | 1513782 | 121 | 2409 | 14894 | 17424 | 86.87913223 |
| | ES_100000 | 14821495 | 154 | 3925 | 31905 | 35984 | 411.8912572 |
| | ES_1000000 | 141276766 | 169 | 6338 | 62199 | 68706 | 2056.250779 |
| | FR_100 | 12875 | 90 | 830 | 2560 | 3480 | 3.699712644 |
| | FR_1000 | 185227 | 99 | 1227 | 5497 | 6823 | 27.14744247 |
| | FR_10000 | 1542105 | 133 | 2492 | 15615 | 18240 | 84.54523026 |
| | FR_100000 | 15055657 | 156 | 4014 | 33105 | 37275 | 403.9076325 |
| | FR_1000000 | 143495667 | 175 | 6423 | 64044 | 70642 | 2031.308103 |
| | RU_100 | 21751 | 100 | 475 | 1365 | 1940 | 11.21185567 |
| | RU_1000 | 309279 | 113 | 694 | 2732 | 3539 | 87.39163606 |
| | RU_10000 | 2636591 | 151 | 1898 | 8430 | 10479 | 251.607119 |
| | RU_100000 | 25990263 | 160 | 3364 | 18321 | 21845 | 1189.757977 |
| | RU_1000000 | 247098758 | 169 | 6224 | 45935 | 52328 | 4722.113553 |
| | ZH_100 | 8559 | 140 | 1524 | 3532 | 5196 | 1.647228637 |
| | ZH_1000 | 116667 | 146 | 2706 | 12857 | 15709 | 7.426761729 |
| | ZH_10000 | 1019969 | 156 | 5596 | 36176 | 41928 | 24.32667907 |
| | ZH_100000 | 9990046 | 167 | 9228 | 81997 | 91392 | 109.3098521 |
| | ZH_1000000 | 94268840 | 196 | 13407 | 160359 | 173962 | 541.893287 |
| WORD | AR_100 | 1776 | 869 | 1534 | 1669 | 4072 | 0.4361493124 |
| | AR_1000 | 23460 | 5868 | 16064 | 20063 | 41995 | 0.5586379331 |
| | AR_10000 | 206549 | 26108 | 116814 | 164062 | 306984 | 0.6728331118 |
| | AR_100000 | 2035190 | 97997 | 776730 | 1383009 | 2257736 | 0.9014295737 |
| | AR_1000000 | 19410502 | 304978 | 4297319 | 10005650 | 14607947 | 1.328763173 |
| | EN_100 | 2071 | 682 | 1567 | 1869 | 4118 | 0.5029140359 |
| | EN_1000 | 27398 | 3292 | 13148 | 19834 | 36274 | 0.7553068313 |
| | EN_10000 | 236569 | 12014 | 83397 | 155493 | 250904 | 0.9428665944 |
| | EN_100000 | 2339109 | 37264 | 428249 | 1117802 | 1583315 | 1.477349106 |
| | EN_1000000 | 21943139 | 122457 | 1818166 | 6505850 | 8446473 | 2.59790554 |
| | ES_100 | 2232 | 710 | 1605 | 1974 | 4289 | 0.5204010259 |
| | ES_1000 | 29461 | 3839 | 13199 | 20634 | 37672 | 0.7820397112 |
| | ES_10000 | 263024 | 15116 | 83900 | 160078 | 259094 | 1.01516824 |
| | ES_100000 | 2588791 | 49499 | 439584 | 1116177 | 1605260 | 1.612692648 |
| | ES_1000000 | 24654449 | 142809 | 1840029 | 6268684 | 8251522 | 2.987866844 |
| | FR_100 | 2298 | 745 | 1737 | 2072 | 4554 | 0.5046113307 |
| | FR_1000 | 32011 | 3881 | 14535 | 22608 | 41024 | 0.780299337 |
| | FR_10000 | 273195 | 13998 | 86815 | 170729 | 271542 | 1.006087456 |
| | FR_100000 | 2684982 | 42870 | 428339 | 1150965 | 1622174 | 1.655175092 |
| | FR_1000000 | 25595487 | 118204 | 1703399 | 6171437 | 7993040 | 3.202221808 |
| | RU_100 | 1854 | 886 | 1589 | 1734 | 4209 | 0.4404846757 |
| | RU_1000 | 24746 | 5433 | 15511 | 20035 | 40979 | 0.603870275 |
| | RU_10000 | 216638 | 23403 | 108516 | 162401 | 294320 | 0.7360627888 |
| | RU_100000 | 2150746 | 81342 | 670857 | 1306351 | 2058550 | 1.044786865 |
| | RU_1000000 | 20421965 | 236088 | 3295028 | 8617195 | 12148311 | 1.681053852 |
| | ZH_100 | 1751 | 630 | 1434 | 1614 | 3678 | 0.4760739532 |
| | ZH_1000 | 23568 | 3181 | 13998 | 19341 | 36520 | 0.6453450164 |
| | ZH_10000 | 207714 | 13137 | 96829 | 160642 | 270608 | 0.7675826287 |
| | ZH_100000 | 2038639 | 46941 | 554739 | 1278188 | 1879868 | 1.08445859 |
| | ZH_1000000 | 19361101 | 134492 | 2527710 | 8401311 | 11063513 | 1.749995774 |

# L NUMBER OF MODEL PARAMETERS

Number of model parameters for dataset A

| Representation / Number of lines | CHAR 100 | CHAR 1,000 | CHAR 10,000 | CHAR 100,000 | CHAR 1,000,000 | BYTE 100 | BYTE 1,000 | BYTE 10,000 | BYTE 100,000 | BYTE 1,000,000 | WORD 100 | WORD 1,000 | WORD 10,000 | WORD 100,000 | WORD 1,000,000 | BPE 100 | BPE 1,000 | BPE 10,000 | BPE 100,000 | BPE 1,000,000 |
|---|---|---|---|---|---|---|---|---|---|---|---|---|---|---|---|---|---|---|---|---|
| **Number of PARAMS** | | | | | | | | | | | | | | | | | | | | |
| AR-EN | 44,226,639 | 44,245,589 | 44,309,118 | 44,369,067 | 44,480,248 | 44,223,056 | 44,240,470 | 44,304,515 | 44,336,795 | 44,360,874 | 45,247,147 | 50,481,885 | 69,784,815 | 132,473,233 | 325,770,330 | 45,127,305 | 49,170,732 | 63,907,829 | 86,155,852 | 89,242,512 |
| AR-ES | 44,235,864 | 44,254,814 | 44,300,918 | 44,375,217 | 44,490,498 | 44,232,281 | 44,249,695 | 44,295,290 | 44,336,795 | 44,360,874 | 45,275,847 | 51,042,560 | 72,964,365 | 145,014,108 | 346,631,130 | 45,143,705 | 49,556,132 | 65,812,279 | 87,022,952 | 89,628,037 |
| AR-FR | 44,237,914 | 44,258,914 | 44,317,318 | 44,387,367 | 44,492,548 | 44,234,331 | 44,254,820 | 44,307,590 | 44,342,945 | 44,367,024 | 45,311,722 | 51,085,610 | 71,818,415 | 138,219,383 | 321,411,005 | 45,207,255 | 49,516,157 | 65,451,479 | 87,305,077 | 89,514,137 |
| AR-RU | 44,247,139 | 44,273,264 | 44,348,068 | 44,389,567 | 44,496,648 | 44,244,581 | 44,269,170 | 44,326,040 | 44,342,945 | 44,367,024 | 45,456,247 | 52,676,410 | 81,458,540 | 177,653,183 | 442,242,105 | 45,329,230 | 50,670,307 | 71,872,079 | 89,287,227 | 89,999,987 |
| AR-ZH | 44,766,814 | 45,429,464 | 46,507,743 | 47,682,892 | 49,237,273 | 44,285,581 | 44,302,995 | 44,331,165 | 44,350,120 | 44,388,549 | 45,193,847 | 50,368,110 | 70,935,890 | 142,392,158 | 338,106,205 | 45,286,180 | 49,163,557 | 64,349,604 | 87,283,352 | 89,324,512 |
| EN-AR | 44,230,230 | 44,258,927 | 44,322,969 | 44,373,684 | 44,477,683 | 44,221,517 | 44,247,139 | 44,306,054 | 44,333,717 | 44,358,822 | 45,343,078 | 51,803,373 | 77,015,037 | 163,629,262 | 419,403,603 | 45,155,530 | 49,845,327 | 68,000,030 | 87,062,638 | 89,700,621 |
| EN-ES | 44,232,280 | 44,241,502 | 44,287,094 | 44,370,609 | 44,493,058 | 44,233,817 | 44,243,039 | 44,293,754 | 44,339,867 | 44,362,922 | 45,180,103 | 49,723,648 | 65,748,237 | 113,918,812 | 253,180,378 | 45,115,545 | 48,882,852 | 61,728,055 | 86,119,688 | 89,171,721 |
| EN-FR | 44,234,330 | 44,245,692 | 44,303,494 | 44,372,619 | 44,495,108 | 44,235,867 | 44,248,164 | 44,306,054 | 44,341,917 | 44,362,922 | 45,215,978 | 49,766,698 | 64,602,287 | 107,124,087 | 227,960,253 | 45,179,095 | 48,842,877 | 61,367,255 | 85,591,813 | 89,056,921 |
| EN-RU | 44,243,555 | 44,259,952 | 44,334,244 | 44,384,959 | 44,499,208 | 44,246,117 | 44,262,514 | 44,324,504 | 44,346,017 | 44,362,022 | 45,360,503 | 51,357,498 | 74,242,412 | 146,557,887 | 348,791,353 | 45,301,070 | 49,997,027 | 67,787,855 | 87,483,963 | 89,542,771 |
| EN-ZH | 44,763,230 | 45,416,152 | 46,493,919 | 47,678,284 | 49,239,833 | 44,287,117 | 44,296,339 | 44,329,629 | 44,353,192 | 44,390,597 | 45,098,103 | 49,049,198 | 63,719,762 | 111,296,862 | 244,655,453 | 45,258,020 | 48,490,277 | 60,265,380 | 85,480,088 | 88,867,296 |
| ES-AR | 44,234,838 | 44,263,535 | 44,318,873 | 44,376,736 | 44,482,803 | 44,226,125 | 44,251,747 | 44,301,446 | 44,333,717 | 44,358,822 | 45,357,414 | 52,083,437 | 78,603,261 | 169,893,582 | 429,823,827 | 45,163,712 | 50,037,839 | 68,951,326 | 88,845,326 | 89,893,645 |
| ES-EN | 44,227,663 | 44,236,885 | 44,291,198 | 44,367,531 | 44,487,928 | 44,229,200 | 44,238,422 | 44,298,371 | 44,339,867 | 44,362,922 | 45,165,739 | 49,443,037 | 64,156,911 | 107,642,257 | 242,739,802 | 45,107,337 | 48,689,964 | 60,774,901 | 85,235,276 | 88,978,320 |
| ES-FR | 44,238,938 | 44,250,210 | 44,299,398 | 44,375,731 | 44,500,228 | 44,240,475 | 44,252,772 | 44,301,446 | 44,341,917 | 44,369,072 | 45,230,314 | 50,046,762 | 66,190,511 | 113,388,407 | 238,380,477 | 45,187,287 | 49,035,389 | 62,318,551 | 86,474,501 | 89,249,945 |
| ES-RU | 44,248,163 | 44,264,560 | 44,330,148 | 44,388,031 | 44,504,328 | 44,250,725 | 44,267,122 | 44,319,896 | 44,346,017 | 44,362,922 | 45,374,839 | 51,637,562 | 75,830,636 | 152,822,207 | 359,211,577 | 45,309,262 | 50,189,539 | 68,739,151 | 88,366,651 | 89,735,795 |
| ES-ZH | 44,767,838 | 45,420,760 | 46,489,823 | 47,681,556 | 49,244,953 | 44,291,725 | 44,300,947 | 44,325,021 | 44,353,192 | 44,390,597 | 45,112,439 | 49,329,262 | 65,307,986 | 117,561,182 | 255,075,677 | 45,206,212 | 48,682,789 | 61,216,676 | 86,362,776 | 89,060,320 |
| FR-AR | 44,235,862 | 44,265,583 | 44,327,065 | 44,377,780 | 44,483,827 | 44,227,149 | 44,254,307 | 44,307,590 | 44,334,741 | 44,361,894 | 45,375,334 | 52,104,941 | 78,630,845 | 166,499,534 | 417,226,067 | 45,195,456 | 50,017,871 | 68,771,102 | 88,581,646 | 89,836,301 |
| FR-EN | 44,228,687 | 44,238,933 | 44,299,390 | 44,368,555 | 44,488,952 | 44,230,224 | 44,240,982 | 44,304,515 | 44,340,891 | 44,365,994 | 45,183,659 | 49,464,541 | 63,584,495 | 104,248,209 | 230,142,042 | 45,139,081 | 48,669,996 | 60,594,677 | 84,971,596 | 88,920,976 |
| FR-ES | 44,237,912 | 44,248,158 | 44,291,190 | 44,374,705 | 44,499,202 | 44,239,449 | 44,250,207 | 44,304,506 | 44,340,891 | 44,365,994 | 45,212,359 | 50,025,216 | 66,764,045 | 116,789,084 | 251,002,842 | 45,155,481 | 49,055,396 | 62,499,127 | 86,738,696 | 89,307,401 |
| FR-RU | 44,249,187 | 44,266,608 | 44,338,340 | 44,389,055 | 44,505,352 | 44,251,749 | 44,269,682 | 44,326,040 | 44,347,041 | 44,365,994 | 45,392,759 | 51,659,066 | 75,258,220 | 149,428,159 | 346,613,817 | 45,341,006 | 50,169,571 | 68,558,927 | 88,102,971 | 89,678,451 |
| FR-ZH | 44,768,862 | 45,422,808 | 46,498,015 | 47,682,380 | 49,245,977 | 44,292,749 | 44,303,507 | 44,331,165 | 44,354,216 | 44,393,669 | 45,130,359 | 49,350,766 | 64,735,570 | 114,167,134 | 242,477,917 | 45,297,956 | 48,662,821 | 61,036,452 | 86,099,096 | 89,002,976 |
| RU-AR | 44,240,470 | 44,272,751 | 44,342,425 | 44,383,924 | 44,485,875 | 44,232,269 | 44,261,475 | 44,316,806 | 44,336,789 | 44,358,822 | 45,447,526 | 52,899,565 | 82,846,205 | 186,197,198 | 477,582,675 | 45,256,384 | 50,594,383 | 71,978,270 | 89,526,798 | 90,078,989 |
| RU-EN | 44,233,295 | 44,246,101 | 44,314,750 | 44,374,699 | 44,491,000 | 44,235,344 | 44,248,150 | 44,313,731 | 44,342,939 | 44,362,922 | 45,255,851 | 50,259,165 | 68,399,855 | 123,945,873 | 290,498,650 | 45,200,009 | 49,246,508 | 63,801,845 | 85,916,748 | 89,163,664 |
| RU-ES | 44,242,520 | 44,255,326 | 44,306,550 | 44,380,849 | 44,501,250 | 44,244,569 | 44,257,375 | 44,304,506 | 44,342,939 | 44,362,922 | 45,284,551 | 50,819,840 | 71,579,405 | 136,486,748 | 311,359,450 | 45,216,409 | 49,631,908 | 65,706,295 | 87,683,848 | 89,550,089 |
| RU-FR | 44,244,570 | 44,259,426 | 44,322,950 | 44,382,899 | 44,503,300 | 44,246,619 | 44,262,500 | 44,316,806 | 44,344,989 | 44,369,072 | 45,320,426 | 50,862,890 | 70,433,455 | 129,692,023 | 286,139,325 | 45,279,959 | 49,591,933 | 65,345,495 | 87,155,973 | 89,435,289 |
| RU-ZH | 44,773,470 | 45,429,976 | 46,513,375 | 47,688,524 | 49,248,025 | 44,297,869 | 44,310,675 | 44,340,381 | 44,356,364 | 44,390,597 | 45,202,551 | 50,145,390 | 69,550,930 | 133,864,798 | 302,834,525 | 45,358,884 | 49,239,333 | 64,243,620 | 87,044,248 | 89,245,664 |
| ZH-AR | 44,500,054 | 44,850,287 | 45,421,209 | 46,028,980 | 46,853,875 | 44,252,749 | 44,278,371 | 44,319,366 | 44,340,373 | 44,372,646 | 45,316,454 | 51,746,541 | 77,590,013 | 168,583,886 | 425,565,523 | 45,234,880 | 49,841,743 | 68,220,702 | 88,525,838 | 89,741,581 |
| ZH-EN | 44,492,879 | 44,823,637 | 45,393,534 | 46,019,755 | 46,869,000 | 44,255,824 | 44,265,046 | 44,316,291 | 44,346,523 | 44,376,746 | 45,124,779 | 49,106,141 | 63,143,663 | 106,332,561 | 238,481,498 | 45,178,505 | 48,493,868 | 60,044,277 | 84,915,788 | 88,826,256 |
| ZH-ES | 44,502,104 | 44,832,862 | 45,385,334 | 46,025,905 | 46,860,800 | 44,265,049 | 44,274,271 | 44,307,066 | 44,346,523 | 44,376,746 | 45,153,479 | 49,666,816 | 66,323,213 | 118,873,436 | 259,342,298 | 45,194,905 | 48,879,268 | 61,948,727 | 86,682,888 | 89,212,681 |
| ZH-FR | 44,504,154 | 44,836,962 | 45,401,734 | 46,027,955 | 46,871,300 | 44,267,099 | 44,279,396 | 44,319,366 | 44,348,573 | 44,382,896 | 45,189,354 | 50,709,866 | 65,177,263 | 112,078,711 | 234,122,173 | 45,258,455 | 48,839,293 | 61,587,927 | 86,155,013 | 89,097,881 |
| ZH-RU | 44,513,379 | 44,851,312 | 45,432,484 | 46,040,255 | 46,875,400 | 44,277,349 | 44,293,746 | 44,337,816 | 44,352,673 | 44,376,746 | 45,333,879 | 51,300,666 | 74,817,388 | 151,512,511 | 354,953,273 | 45,380,430 | 49,993,443 | 68,008,527 | 88,047,163 | 89,583,731 |
| AR-ZH_pinyin | 44,230,739 | 44,254,814 | 44,290,668 | 44,355,742 | 44,808,248 | | | | | | | | | | | | | | | |
| EN-ZH_pinyin | 44,227,155 | 44,241,502 | 44,276,844 | 44,351,134 | 44,810,808 | | | | | | | | | | | | | | | |
| ES-ZH_pinyin | 44,231,763 | 44,246,110 | 44,272,748 | 44,354,206 | 44,815,928 | | | | | | | | | | | | | | | |
| FR-ZH_pinyin | 44,232,787 | 44,248,158 | 44,280,940 | 44,355,230 | 44,816,952 | | | | | | | | | | | | | | | |
| RU-ZH_pinyin | 44,237,395 | 44,255,326 | 44,296,300 | 44,361,374 | 44,819,000 | | | | | | | | | | | | | | | |
| AR-ZH_wubi | 44,252,264 | 44,271,214 | 44,310,143 | 44,372,142 | 44,785,698 | | | | | | | | | | | | | | | |
| EN-ZH_wubi | 44,248,680 | 44,257,902 | 44,296,319 | 44,367,534 | 44,788,258 | | | | | | | | | | | | | | | |
| ES-ZH_wubi | 44,253,288 | 44,262,510 | 44,292,223 | 44,370,606 | 44,793,378 | | | | | | | | | | | | | | | |
| FR-ZH_wubi | 44,254,312 | 44,264,558 | 44,300,415 | 44,371,630 | 44,794,402 | | | | | | | | | | | | | | | |
| RU-ZH_wubi | 44,258,920 | 44,271,726 | 44,315,775 | 44,377,774 | 44,796,450 | | | | | | | | | | | | | | | |
| EN-AR_cp1256 | | | | | | 44,230,742 | 44,259,439 | 44,325,529 | 44,365,492 | 44,437,747 | | | | | | | | | | |
| ES-AR_cp1256 | | | | | | 44,235,350 | 44,264,047 | 44,320,921 | 44,365,492 | 44,437,747 | | | | | | | | | | |
| FR-AR_cp1256 | | | | | | 44,236,374 | 44,266,607 | 44,327,065 | 44,366,516 | 44,440,819 | | | | | | | | | | |
| RU-AR_cp1256 | | | | | | 44,241,494 | 44,273,775 | 44,336,281 | 44,368,564 | 44,437,747 | | | | | | | | | | |
| ZH-AR_cp1256 | | | | | | 44,261,974 | 44,290,671 | 44,338,841 | 44,372,148 | 44,451,571 | | | | | | | | | | |
| AR-RU_cp1251 | | | | | | 44,242,531 | 44,267,120 | 44,338,340 | 44,373,695 | 44,457,224 | | | | | | | | | | |
| EN-RU_cp1251 | | | | | | 44,244,067 | 44,265,464 | 44,336,194 | 44,376,767 | 44,459,272 | | | | | | | | | | |
| ES-RU_cp1251 | | | | | | 44,248,675 | 44,265,072 | 44,332,196 | 44,376,767 | 44,459,272 | | | | | | | | | | |
| FR-RU_cp1251 | | | | | | 44,249,699 | 44,267,632 | 44,338,340 | 44,377,791 | 44,462,344 | | | | | | | | | | |
| ZH-RU_cp1251 | | | | | | 44,275,299 | 44,291,696 | 44,350,116 | 44,383,423 | 44,473,096 | | | | | | | | | | |

## Number of model parameters for dataset B

| Representation / Number of lines | CHAR 100 | CHAR 1,000 | CHAR 10,000 | CHAR 100,000 | CHAR 1,000,000 | BYTE 100 | BYTE 1,000 | BYTE 10,000 | BYTE 100,000 | BYTE 1,000,000 | WORD 100 | WORD 1,000 | WORD 10,000 | WORD 100,000 | WORD 1,000,000 | BPE 100 | BPE 1,000 | BPE 10,000 | BPE 100,000 | BPE 1,000,000 |
|---|---|---|---|---|---|---|---|---|---|---|---|---|---|---|---|---|---|---|---|---|
| **Number of PARAMS** | | | | | | | | | | | | | | | | | | | | |
| AR-EN | 44,230,224 | 44,277,863 | 44,328,071 | 44,430,017 | 44,680,027 | 44,233,811 | 44,276,843 | 44,321,419 | 44,356,263 | 44,385,977 | 45,583,728 | 51,799,608 | 72,558,944 | 137,925,984 | 347,956,340 | 45,326,100 | 49,857,876 | 64,758,280 | 86,794,440 | 89,354,238 |
| AR-ES | 44,240,474 | 44,282,988 | 44,331,146 | 44,430,017 | 44,703,602 | 44,244,061 | 44,281,968 | 44,319,369 | 44,354,213 | 44,390,077 | 45,680,078 | 52,442,283 | 75,710,819 | 146,661,034 | 361,865,590 | 45,356,850 | 50,162,301 | 66,575,605 | 88,073,640 | 89,654,563 |
| AR-FR | 44,243,549 | 44,291,188 | 44,344,471 | 44,443,342 | 44,689,252 | 44,246,111 | 44,294,268 | 44,328,594 | 44,355,238 | 44,390,077 | 45,664,703 | 52,299,808 | 74,587,419 | 140,492,584 | 339,724,565 | 45,354,800 | 50,126,426 | 66,297,830 | 87,741,540 | 89,551,038 |
| AR-RU | 44,264,049 | 44,321,938 | 44,371,121 | 44,445,392 | 44,656,452 | 44,267,636 | 44,315,793 | 44,346,019 | 44,354,213 | 44,384,952 | 45,856,378 | 54,230,908 | 85,165,419 | 180,046,309 | 461,160,415 | 45,540,325 | 51,314,401 | 73,041,305 | 89,296,465 | 90,003,063 |
| AR-ZH | 44,932,349 | 45,790,763 | 46,655,846 | 47,787,917 | 49,151,077 | 44,278,911 | 44,328,093 | 44,352,169 | 44,368,563 | 44,393,152 | 45,605,253 | 51,813,958 | 73,639,294 | 141,968,584 | 347,214,240 | 45,617,200 | 50,174,601 | 65,079,105 | 87,398,165 | 89,281,463 |
| EN-AR | 44,235,867 | 44,295,818 | 44,347,052 | 44,461,823 | 44,725,171 | 44,238,428 | 44,287,616 | 44,327,575 | 44,354,211 | 44,385,977 | 45,713,517 | 53,118,531 | 80,029,250 | 165,947,583 | 435,393,599 | 45,339,438 | 50,370,876 | 68,884,852 | 88,261,620 | 89,754,891 |
| EN-ES | 44,234,842 | 44,265,068 | 44,312,202 | 44,398,273 | 44,658,546 | 44,239,453 | 44,271,216 | 44,313,225 | 44,356,261 | 44,390,077 | 45,550,542 | 53,125,931 | 68,255,075 | 118,694,058 | 274,598,774 | 45,343,538 | 49,650,301 | 62,457,077 | 86,609,320 | 89,254,691 |
| EN-FR | 44,237,917 | 44,273,246 | 44,325,527 | 44,411,598 | 44,644,196 | 44,241,503 | 44,283,516 | 44,322,436 | 44,357,286 | 44,390,077 | 45,535,167 | 50,983,456 | 67,131,675 | 112,525,608 | 252,457,749 | 45,341,488 | 49,614,426 | 62,179,302 | 86,277,220 | 89,151,166 |
| EN-RU | 44,258,417 | 44,304,018 | 44,352,177 | 44,413,648 | 44,611,396 | 44,263,028 | 44,305,041 | 44,339,875 | 44,356,261 | 44,384,952 | 45,726,842 | 52,914,556 | 77,709,675 | 152,079,333 | 373,893,599 | 45,527,013 | 50,802,401 | 68,922,777 | 87,832,145 | 89,603,191 |
| EN-ZH | 44,926,717 | 45,772,843 | 46,636,902 | 47,756,173 | 49,106,021 | 44,274,303 | 44,317,341 | 44,346,025 | 44,370,611 | 44,393,152 | 45,475,717 | 50,497,606 | 66,183,550 | 114,001,608 | 259,947,424 | 45,603,888 | 49,662,601 | 60,960,577 | 85,933,845 | 88,881,591 |
| ES-AR | 44,240,987 | 44,298,378 | 44,348,588 | 44,461,823 | 44,736,947 | 44,243,548 | 44,290,176 | 44,326,551 | 44,353,187 | 44,388,025 | 45,761,645 | 53,439,555 | 81,603,650 | 170,310,847 | 442,341,439 | 45,354,798 | 50,522,940 | 69,792,628 | 88,900,596 | 89,904,907 |
| ES-EN | 44,229,712 | 44,262,503 | 44,310,663 | 44,398,273 | 44,646,747 | 44,234,323 | 44,268,651 | 44,314,251 | 44,357,287 | 44,388,025 | 45,502,320 | 50,804,280 | 66,677,600 | 114,322,272 | 267,637,364 | 45,328,148 | 49,497,940 | 61,547,528 | 85,969,096 | 89,104,382 |
| ES-FR | 44,243,037 | 44,275,828 | 44,327,063 | 44,411,598 | 44,655,972 | 44,246,623 | 44,286,076 | 44,321,426 | 44,356,262 | 44,392,125 | 45,583,295 | 51,304,480 | 68,706,075 | 116,888,872 | 259,405,589 | 45,356,848 | 49,766,490 | 63,087,078 | 86,491,196 | 89,301,182 |
| ES-RU | 44,263,537 | 44,306,578 | 44,353,713 | 44,413,648 | 44,623,172 | 44,268,148 | 44,307,601 | 44,338,851 | 44,355,237 | 44,387,000 | 45,774,970 | 53,235,580 | 79,284,075 | 156,442,597 | 380,841,439 | 45,542,373 | 50,954,465 | 69,830,553 | 88,471,121 | 89,753,207 |
| ES-ZH | 44,931,837 | 45,775,403 | 46,638,438 | 47,756,173 | 49,117,797 | 44,279,423 | 44,319,901 | 44,345,001 | 44,369,587 | 44,395,200 | 45,523,845 | 50,818,630 | 67,757,950 | 118,364,872 | 266,895,264 | 45,619,248 | 49,814,665 | 61,868,353 | 86,572,821 | 89,031,607 |
| FR-AR | 44,242,523 | 44,302,474 | 44,355,244 | 44,468,479 | 44,729,779 | 44,244,572 | 44,296,320 | 44,331,159 | 44,353,699 | 44,388,025 | 45,753,965 | 53,368,387 | 81,042,498 | 167,229,631 | 431,281,727 | 45,353,774 | 50,505,020 | 69,653,876 | 88,734,708 | 89,853,195 |
| FR-EN | 44,231,248 | 44,266,599 | 44,317,319 | 44,404,929 | 44,639,579 | 44,235,347 | 44,274,795 | 44,318,859 | 44,357,799 | 44,388,025 | 45,494,640 | 50,733,112 | 66,116,448 | 111,241,056 | 256,577,652 | 45,327,124 | 49,480,020 | 61,408,776 | 85,803,208 | 89,052,670 |
| FR-ES | 44,241,498 | 44,271,724 | 44,320,394 | 44,404,929 | 44,663,154 | 44,245,597 | 44,279,920 | 44,316,809 | 44,355,749 | 44,392,125 | 45,590,990 | 51,375,787 | 69,208,323 | 119,976,106 | 270,486,902 | 45,357,874 | 49,784,445 | 63,226,101 | 87,082,408 | 89,352,995 |
| FR-RU | 44,265,073 | 44,310,674 | 44,360,369 | 44,420,304 | 44,616,004 | 44,269,172 | 44,313,745 | 44,343,459 | 44,355,749 | 44,387,000 | 45,767,290 | 53,164,412 | 78,722,923 | 153,361,381 | 369,781,727 | 45,541,349 | 50,936,545 | 69,691,801 | 88,305,233 | 89,701,495 |
| FR-ZH | 44,933,373 | 45,779,499 | 46,645,094 | 47,762,829 | 49,110,629 | 44,280,447 | 44,326,045 | 44,349,609 | 44,370,099 | 44,395,200 | 45,516,165 | 50,747,462 | 67,196,798 | 115,283,656 | 255,835,552 | 45,618,224 | 49,796,745 | 61,729,601 | 86,406,933 | 88,979,895 |
| RU-AR | 44,252,763 | 44,317,834 | 44,368,556 | 44,469,503 | 44,713,395 | 44,255,324 | 44,307,072 | 44,339,863 | 44,353,187 | 44,385,465 | 45,849,709 | 54,332,995 | 86,326,338 | 186,987,199 | 491,940,415 | 45,446,446 | 51,098,428 | 73,022,324 | 89,511,412 | 90,078,987 |
| RU-EN | 44,241,488 | 44,281,959 | 44,330,631 | 44,405,953 | 44,623,195 | 44,246,099 | 44,285,547 | 44,327,563 | 44,357,287 | 44,385,465 | 45,590,384 | 51,697,720 | 71,400,288 | 130,998,624 | 317,236,340 | 45,419,786 | 50,073,428 | 64,777,224 | 86,579,912 | 89,278,462 |
| RU-ES | 44,251,738 | 44,287,084 | 44,333,706 | 44,405,953 | 44,646,770 | 44,256,349 | 44,290,672 | 44,325,513 | 44,355,237 | 44,389,565 | 45,686,734 | 52,340,395 | 74,552,163 | 139,733,674 | 331,145,590 | 45,450,546 | 50,377,853 | 66,594,549 | 87,859,112 | 89,578,787 |
| RU-FR | 44,254,813 | 44,296,284 | 44,347,031 | 44,419,278 | 44,632,420 | 44,258,399 | 44,302,972 | 44,334,738 | 44,356,262 | 44,389,565 | 45,671,359 | 52,197,920 | 73,428,763 | 133,565,224 | 309,004,565 | 45,448,496 | 50,341,978 | 66,316,774 | 87,527,012 | 89,475,262 |
| RU-ZH | 44,943,613 | 45,794,859 | 46,658,406 | 47,763,853 | 49,094,245 | 44,291,199 | 44,336,797 | 44,358,313 | 44,369,587 | 44,392,640 | 45,611,909 | 51,712,070 | 72,480,638 | 135,041,224 | 316,494,240 | 45,710,806 | 50,390,153 | 65,098,049 | 87,183,637 | 89,205,687 |
| ZH-AR | 44,586,587 | 45,051,530 | 45,509,804 | 46,139,135 | 46,958,515 | 44,260,956 | 44,313,216 | 44,342,935 | 44,360,355 | 44,389,561 | 45,724,269 | 53,125,699 | 80,568,898 | 167,966,911 | 435,022,911 | 45,484,846 | 50,529,084 | 69,045,108 | 88,563,188 | 89,718,539 |
| ZH-EN | 44,575,312 | 45,015,655 | 45,471,879 | 46,075,685 | 46,898,315 | 44,251,731 | 44,291,691 | 44,330,635 | 44,364,455 | 44,389,561 | 45,464,944 | 50,490,424 | 65,642,848 | 111,978,336 | 260,318,836 | 45,458,196 | 49,504,084 | 60,800,008 | 85,631,688 | 89,918,014 |
| ZH-ES | 44,585,562 | 45,020,780 | 45,474,954 | 46,075,685 | 46,891,880 | 44,261,981 | 44,296,816 | 44,328,585 | 44,362,405 | 44,393,661 | 45,561,294 | 51,133,099 | 68,794,723 | 120,713,386 | 274,228,886 | 45,488,946 | 49,808,509 | 62,617,333 | 86,010,888 | 89,218,339 |
| ZH-FR | 44,588,637 | 45,028,980 | 45,488,279 | 46,088,910 | 46,877,540 | 44,264,031 | 44,309,116 | 44,337,810 | 44,363,430 | 44,393,661 | 45,545,919 | 50,990,624 | 67,671,323 | 114,544,936 | 252,087,061 | 45,486,896 | 49,772,634 | 62,339,558 | 86,578,788 | 89,114,814 |
| ZH-RU | 44,609,137 | 45,059,730 | 45,514,929 | 46,090,960 | 46,844,740 | 44,285,556 | 44,330,641 | 44,355,235 | 44,362,405 | 44,388,536 | 45,737,594 | 52,921,724 | 78,249,323 | 154,098,661 | 373,522,011 | 45,672,421 | 50,960,609 | 69,083,033 | 88,133,713 | 89,566,839 |
| AR-ZH_pinyin | 44,225,099 | 44,282,988 | 44,337,296 | 44,483,317 | 44,700,202 | 44,237,403 | 44,297,866 | 44,349,100 | 44,448,511 | 44,642,227 | | | | | | | | | | |
| EN-ZH_pinyin | 44,219,467 | 44,265,068 | 44,318,352 | 44,451,573 | 44,724,146 | 44,242,523 | 44,300,426 | 44,348,076 | 44,447,487 | 44,644,275 | | | | | | | | | | |
| ES-ZH_pinyin | 44,224,587 | 44,267,628 | 44,319,888 | 44,451,573 | 44,735,922 | 44,243,547 | 44,306,570 | 44,361,388 | 44,447,999 | 44,644,275 | | | | | | | | | | |
| FR-ZH_pinyin | 44,226,123 | 44,271,724 | 44,326,544 | 44,458,229 | 44,728,724 | 44,254,299 | 44,317,322 | 44,352,684 | 44,447,487 | 44,641,715 | | | | | | | | | | |
| RU-ZH_pinyin | 44,236,363 | 44,287,084 | 44,339,856 | 44,459,253 | 44,712,370 | 44,259,931 | 44,323,466 | 44,364,460 | 44,454,655 | 44,645,811 | | | | | | | | | | |
| AR-ZH_wubi | 44,241,499 | 44,303,488 | 44,351,646 | 44,488,442 | 44,759,977 | 44,264,561 | 44,316,818 | 44,360,369 | 44,398,288 | 44,528,452 | | | | | | | | | | |
| EN-ZH_wubi | 44,235,867 | 44,285,568 | 44,332,702 | 44,456,698 | 44,714,921 | 44,259,953 | 44,306,066 | 44,354,225 | 44,400,336 | 44,528,452 | | | | | | | | | | |
| ES-ZH_wubi | 44,240,987 | 44,288,128 | 44,334,258 | 44,456,698 | 44,726,697 | 44,265,073 | 44,308,626 | 44,353,201 | 44,399,312 | 44,530,500 | | | | | | | | | | |
| FR-ZH_wubi | 44,242,523 | 44,292,224 | 44,340,894 | 44,463,354 | 44,719,529 | 44,266,097 | 44,314,770 | 44,357,809 | 44,399,824 | 44,530,500 | | | | | | | | | | |
| RU-ZH_wubi | 44,252,763 | 44,307,584 | 44,354,206 | 44,464,378 | 44,703,145 | 44,282,481 | 44,331,666 | 44,369,585 | 44,406,480 | 44,532,036 | | | | | | | | | | |
| EN-AR_cpl256 | | | | | | | | | | | | | | | | | | | | |
| ES-AR_cpl256 | | | | | | | | | | | | | | | | | | | | |
| FR-AR_cpl256 | | | | | | | | | | | | | | | | | | | | |
| RU-AR_cpl256 | | | | | | | | | | | | | | | | | | | | |
| ZH-AR_cpl256 | | | | | | | | | | | | | | | | | | | | |
| AR-RU_cpl251 | | | | | | | | | | | | | | | | | | | | |
| EN-RU_cpl251 | | | | | | | | | | | | | | | | | | | | |
| ES-RU_cpl251 | | | | | | | | | | | | | | | | | | | | |
| FR-RU_cpl251 | | | | | | | | | | | | | | | | | | | | |
| ZH-RU_cpl251 | | | | | | | | | | | | | | | | | | | | |

## Number of model parameters for dataset C

| Representation / Number of lines | CHAR 100 | CHAR 1,000 | CHAR 10,000 | CHAR 100,000 | CHAR 1,000,000 | BYTE 100 | BYTE 1,000 | BYTE 10,000 | BYTE 100,000 | BYTE 1,000,000 | WORD 100 | WORD 1,000 | WORD 10,000 | WORD 100,000 | WORD 1,000,000 | BPE 100 | BPE 1,000 | BPE 10,000 | BPE 100,000 | BPE 1,000,000 |
|---|---|---|---|---|---|---|---|---|---|---|---|---|---|---|---|---|---|---|---|---|
| **Number of PARAMS** | | | | | | | | | | | | | | | | | | | | |
| AR-EN | 44,237,394 | 44,277,862 | 44,319,874 | 44,432,062 | 44,644,154 | 44,237,909 | 44,276,331 | 44,315,783 | 44,352,675 | 44,380,342 | 45,632,910 | 51,878,481 | 72,594,782 | 137,710,265 | 347,843,439 | 45,372,198 | 49,998,240 | 64,770,100 | 86,735,506 | 89,389,603 |
| AR-ES | 44,243,544 | 44,288,112 | 44,327,049 | 44,432,062 | 44,640,054 | 44,242,009 | 44,285,556 | 44,317,833 | 44,353,700 | 44,377,267 | 45,698,510 | 52,453,506 | 75,866,582 | 146,529,365 | 361,666,589 | 45,373,223 | 50,205,290 | 66,481,850 | 88,137,706 | 89,650,978 |
| AR-FR | 44,246,619 | 44,293,237 | 44,333,199 | 44,441,287 | 44,664,654 | 44,248,159 | 44,290,681 | 44,318,858 | 44,352,675 | 44,381,367 | 45,720,035 | 52,319,231 | 74,570,982 | 140,332,215 | 338,956,089 | 45,424,473 | 50,212,465 | 66,174,350 | 87,724,631 | 89,533,103 |
| AR-RU | 44,257,894 | 44,321,937 | 44,361,899 | 44,457,687 | 44,620,579 | 44,257,384 | 44,314,256 | 44,339,358 | 44,359,850 | 44,379,317 | 45,898,385 | 54,253,406 | 85,360,132 | 180,206,765 | 460,198,814 | 45,577,198 | 51,406,590 | 73,204,825 | 89,355,406 | 90,003,578 |
| AR-ZH | 44,960,019 | 45,766,162 | 46,694,799 | 47,805,337 | 49,141,854 | 44,286,084 | 44,326,556 | 44,347,558 | 44,371,125 | 44,391,617 | 45,608,310 | 51,865,156 | 73,693,582 | 141,800,015 | 346,543,739 | 45,661,248 | 50,236,040 | 65,194,450 | 87,429,431 | 89,281,978 |
| EN-AR | 44,247,141 | 44,297,356 | 44,338,342 | 44,470,537 | 44,704,175 | 44,243,552 | 44,286,591 | 44,322,452 | 44,353,188 | 44,379,316 | 45,765,777 | 53,237,931 | 80,104,076 | 165,772,904 | 434,781,036 | 45,404,004 | 50,534,838 | 68,840,755 | 88,226,797 | 89,768,710 |
| EN-ES | 44,233,816 | 44,268,656 | 44,308,617 | 44,393,662 | 44,580,150 | 44,236,377 | 44,275,316 | 44,311,177 | 44,353,188 | 44,378,291 | 45,565,902 | 51,096,706 | 68,371,926 | 118,521,429 | 274,898,461 | 45,341,479 | 49,669,738 | 62,419,130 | 86,649,322 | 89,272,610 |
| EN-FR | 44,236,891 | 44,273,781 | 44,314,767 | 44,402,687 | 44,604,750 | 44,242,527 | 44,280,441 | 44,312,202 | 44,352,163 | 44,382,391 | 45,587,427 | 50,962,431 | 67,076,326 | 112,324,279 | 252,188,561 | 45,392,729 | 49,676,913 | 62,111,630 | 86,236,247 | 89,154,735 |
| EN-RU | 44,248,166 | 44,302,481 | 44,343,467 | 44,419,287 | 44,560,675 | 44,251,752 | 44,304,016 | 44,332,702 | 44,359,338 | 44,380,341 | 45,765,777 | 52,896,606 | 77,865,476 | 152,198,829 | 373,430,686 | 45,545,454 | 50,871,038 | 69,142,105 | 87,867,022 | 89,625,210 |
| EN-ZH | 44,950,291 | 45,746,706 | 46,676,367 | 47,766,937 | 49,081,950 | 44,280,452 | 44,316,316 | 44,340,902 | 44,370,613 | 44,392,641 | 45,475,702 | 50,508,356 | 66,198,926 | 113,792,079 | 259,775,611 | 45,629,504 | 49,700,488 | 61,131,730 | 85,941,047 | 88,903,610 |
| ES-AR | 44,250,213 | 44,302,476 | 44,341,926 | 44,470,537 | 44,702,127 | 44,245,600 | 44,291,199 | 44,323,476 | 44,353,700 | 44,377,780 | 45,798,545 | 53,525,163 | 81,738,380 | 170,178,152 | 441,685,968 | 45,404,516 | 50,638,262 | 69,695,795 | 88,927,213 | 89,899,270 |
| ES-EN | 44,230,738 | 44,263,526 | 44,305,026 | 44,393,662 | 44,582,202 | 44,234,325 | 44,270,699 | 44,310,151 | 44,352,675 | 44,379,830 | 45,533,070 | 50,808,913 | 66,734,430 | 114,107,577 | 267,980,143 | 45,340,966 | 49,566,112 | 61,562,420 | 85,947,538 | 89,141,795 |
| ES-FR | 44,239,963 | 44,278,901 | 44,318,351 | 44,402,887 | 44,602,702 | 44,244,575 | 44,285,049 | 44,313,226 | 44,352,675 | 44,380,855 | 45,620,195 | 51,249,663 | 68,710,630 | 116,729,527 | 259,093,393 | 45,393,241 | 49,780,337 | 62,966,670 | 86,936,663 | 89,285,295 |
| ES-RU | 44,251,238 | 44,307,601 | 44,347,051 | 44,419,287 | 44,558,627 | 44,253,800 | 44,308,624 | 44,333,726 | 44,359,850 | 44,378,805 | 45,798,545 | 53,183,838 | 79,499,780 | 156,604,077 | 380,335,518 | 45,545,966 | 50,974,462 | 60,997,145 | 88,567,438 | 89,755,770 |
| ES-ZH | 44,953,363 | 45,751,826 | 46,679,951 | 47,766,937 | 49,079,902 | 44,282,500 | 44,320,924 | 44,341,926 | 44,371,125 | 44,391,105 | 45,508,470 | 50,795,588 | 67,833,230 | 118,197,327 | 266,680,443 | 45,630,016 | 49,803,912 | 61,986,770 | 86,641,463 | 89,034,170 |
| FR-AR | 44,251,749 | 44,305,036 | 44,344,998 | 44,475,145 | 44,714,415 | 44,248,672 | 44,293,759 | 44,323,988 | 44,353,188 | 44,379,828 | 45,809,297 | 53,458,091 | 81,091,212 | 167,082,600 | 430,341,996 | 45,430,116 | 50,641,846 | 69,542,195 | 88,720,877 | 89,840,390 |
| FR-EN | 44,232,274 | 44,266,086 | 44,308,098 | 44,398,270 | 44,594,490 | 44,237,397 | 44,273,259 | 44,310,663 | 44,352,163 | 44,381,878 | 45,543,822 | 50,741,841 | 66,087,262 | 111,012,025 | 256,636,271 | 45,366,566 | 49,569,696 | 61,408,820 | 85,741,202 | 89,082,915 |
| FR-ES | 44,238,424 | 44,276,336 | 44,315,273 | 44,398,270 | 44,590,390 | 44,241,497 | 44,282,484 | 44,312,713 | 44,353,188 | 44,378,803 | 45,609,422 | 51,316,866 | 69,350,062 | 119,831,125 | 270,459,421 | 45,367,591 | 49,776,746 | 63,120,570 | 87,143,402 | 89,344,290 |
| FR-RU | 44,252,774 | 44,310,161 | 44,350,123 | 44,423,895 | 44,570,915 | 44,256,872 | 44,311,184 | 44,334,238 | 44,359,338 | 44,380,853 | 45,809,297 | 53,116,766 | 78,852,612 | 153,508,525 | 368,991,646 | 45,571,566 | 50,978,046 | 69,843,545 | 88,361,102 | 89,696,880 |
| FR-ZH | 44,954,899 | 45,754,386 | 46,683,023 | 47,771,545 | 49,092,190 | 44,285,572 | 44,323,484 | 44,342,438 | 44,370,613 | 44,393,153 | 45,519,222 | 50,728,516 | 67,186,062 | 115,101,775 | 255,336,571 | 45,655,616 | 49,807,496 | 61,833,170 | 86,435,127 | 88,975,290 |
| RU-AR | 44,257,381 | 44,319,372 | 44,359,334 | 44,483,337 | 44,692,399 | 44,253,280 | 44,305,535 | 44,334,228 | 44,356,772 | 44,378,894 | 45,808,385 | 54,424,235 | 86,480,524 | 187,000,424 | 490,903,916 | 45,506,404 | 51,238,326 | 73,054,003 | 89,535,469 | 90,075,398 |
| RU-EN | 44,237,906 | 44,280,422 | 44,322,434 | 44,406,462 | 44,572,474 | 44,242,005 | 44,285,035 | 44,320,903 | 44,355,747 | 44,380,854 | 45,632,910 | 51,707,985 | 71,474,574 | 130,929,849 | 317,198,191 | 45,442,854 | 50,166,176 | 64,920,628 | 86,555,794 | 89,317,923 |
| RU-ES | 44,244,056 | 44,290,672 | 44,329,609 | 44,406,462 | 44,568,374 | 44,246,105 | 44,294,260 | 44,322,953 | 44,356,772 | 44,377,779 | 45,698,510 | 52,283,010 | 74,748,374 | 139,748,949 | 331,021,341 | 45,443,879 | 50,373,226 | 66,632,378 | 87,957,994 | 89,579,298 |
| RU-FR | 44,247,131 | 44,295,797 | 44,335,759 | 44,415,687 | 44,592,974 | 44,252,255 | 44,299,385 | 44,323,978 | 44,355,747 | 44,381,879 | 45,720,035 | 52,148,735 | 73,452,774 | 133,551,799 | 308,311,441 | 45,495,129 | 50,380,401 | 66,324,878 | 87,544,919 | 89,461,423 |
| RU-ZH | 44,960,531 | 45,768,722 | 46,697,359 | 47,779,737 | 49,070,174 | 44,290,180 | 44,335,260 | 44,352,678 | 44,374,197 | 44,392,129 | 45,608,310 | 51,694,660 | 72,575,374 | 135,019,599 | 315,898,491 | 45,731,904 | 50,403,976 | 65,344,978 | 87,249,719 | 89,210,298 |
| ZH-AR | 44,608,101 | 45,040,780 | 45,524,646 | 46,155,529 | 46,950,831 | 44,267,616 | 44,311,679 | 44,328,324 | 44,362,404 | 44,384,948 | 45,753,489 | 53,231,275 | 80,652,940 | 167,815,784 | 434,131,820 | 45,548,388 | 50,653,622 | 69,052,723 | 88,573,421 | 89,714,050 |
| ZH-EN | 44,588,626 | 45,001,830 | 45,487,746 | 46,078,654 | 46,830,906 | 44,256,341 | 44,291,179 | 44,324,999 | 44,361,379 | 44,386,998 | 45,488,014 | 50,515,025 | 65,648,990 | 111,745,209 | 260,426,095 | 45,484,888 | 49,581,472 | 60,919,348 | 85,593,746 | 88,957,475 |
| ZH-ES | 44,594,776 | 45,012,080 | 45,494,921 | 46,078,654 | 46,826,806 | 44,260,441 | 44,300,404 | 44,324,049 | 44,362,404 | 44,383,023 | 45,553,614 | 51,090,050 | 68,920,790 | 120,564,309 | 274,249,245 | 45,485,863 | 49,788,522 | 62,631,098 | 86,995,946 | 89,218,850 |
| ZH-FR | 44,597,851 | 45,017,205 | 45,501,071 | 46,087,879 | 46,851,406 | 44,266,591 | 44,305,529 | 44,328,074 | 44,361,379 | 44,388,023 | 45,575,139 | 50,955,775 | 67,625,190 | 114,367,159 | 251,539,345 | 45,537,113 | 49,795,697 | 62,323,598 | 86,582,871 | 89,100,075 |
| ZH-RU | 44,609,126 | 45,045,905 | 45,529,771 | 46,104,279 | 46,807,331 | 44,275,816 | 44,329,104 | 44,348,574 | 44,368,554 | 44,385,973 | 45,753,489 | 52,889,950 | 78,414,340 | 154,241,709 | 372,781,470 | 45,689,838 | 50,989,822 | 69,354,073 | 88,213,646 | 89,571,450 |
| AR-ZH_pinyin | 44,233,294 | 44,287,087 | 44,333,199 | 44,487,412 | 44,749,729 | | | | | | | | | | | | | | | |
| EN-ZH_pinyin | 44,223,566 | 44,267,631 | 44,314,767 | 44,449,012 | 44,689,825 | | | | | | | | | | | | | | | |
| ES-ZH_pinyin | 44,226,638 | 44,272,751 | 44,318,351 | 44,449,012 | 44,687,777 | | | | | | | | | | | | | | | |
| FR-ZH_pinyin | 44,228,174 | 44,275,311 | 44,321,423 | 44,453,620 | 44,700,065 | | | | | | | | | | | | | | | |
| RU-ZH_pinyin | 44,233,806 | 44,289,647 | 44,335,759 | 44,461,812 | 44,678,049 | | | | | | | | | | | | | | | |
| AR-ZH_wubi | 44,254,819 | 44,309,637 | 44,348,574 | 44,489,462 | 44,744,604 | | | | | | | | | | | | | | | |
| EN-ZH_wubi | 44,245,091 | 44,290,181 | 44,330,142 | 44,451,062 | 44,684,700 | | | | | | | | | | | | | | | |
| ES-ZH_wubi | 44,248,163 | 44,295,301 | 44,333,726 | 44,451,062 | 44,682,652 | | | | | | | | | | | | | | | |
| FR-ZH_wubi | 44,249,699 | 44,297,861 | 44,336,798 | 44,455,670 | 44,694,940 | | | | | | | | | | | | | | | |
| RU-ZH_wubi | 44,255,331 | 44,312,197 | 44,351,134 | 44,463,862 | 44,672,924 | | | | | | | | | | | | | | | |
| EN-AR_cpl256 | | | | | | 44,248,677 | 44,299,916 | 44,340,902 | 44,456,713 | 44,636,591 | | | | | | | | | | |
| ES-AR_cpl256 | | | | | | 44,250,725 | 44,304,524 | 44,341,926 | 44,457,225 | 44,635,055 | | | | | | | | | | |
| FR-AR_cpl256 | | | | | | 44,258,405 | 44,307,084 | 44,342,438 | 44,456,713 | 44,637,103 | | | | | | | | | | |
| RU-AR_cpl256 | | | | | | 44,272,741 | 44,318,860 | 44,352,678 | 44,460,297 | 44,636,079 | | | | | | | | | | |
| ZH-AR_cpl256 | | | | | | | 44,325,004 | 44,356,774 | 44,465,929 | 44,642,223 | | | | | | | | | | |
| AR-RU_cpl251 | | | | | | 44,255,334 | 44,315,281 | 44,352,683 | 44,405,975 | 44,492,067 | | | | | | | | | | |
| EN-RU_cpl251 | | | | | | 44,249,702 | 44,305,041 | 44,346,027 | 44,405,463 | 44,493,091 | | | | | | | | | | |
| ES-RU_cpl251 | | | | | | 44,251,750 | 44,309,649 | 44,347,051 | 44,405,975 | 44,491,555 | | | | | | | | | | |
| FR-RU_cpl251 | | | | | | 44,254,822 | 44,312,209 | 44,347,563 | 44,405,463 | 44,493,603 | | | | | | | | | | |
| ZH-RU_cpl251 | | | | | | 44,273,766 | 44,330,129 | 44,361,899 | 44,414,679 | 44,498,723 | | | | | | | | | | |

## M    ERRATICITY

Length has been an issue since the dawn of the encoder-decoder approach for NMT (Cho et al., 2014). Most work on length bias, except for that by e.g. Sountsov & Sarawagi (2016), seems to have focused on the evaluation of generated translation output and monitored performance degradation with respect to sequence length, often arguing that beam size plays a role (Koehn & Knowles, 2017; Murray & Chiang, 2018). (Related work in Stahlberg & Byrne (2019) provides a good summary on this issue.) While there could also be confounds in search, our experiments show that a kind of length bias can surface already with CLMing, without generation taking place. To our knowledge, length bias has not been expressed as a sample-wise non-monotonicity across a large data size range as ours. While the connection between erraticity in CLMs and length bias in NMT models remains to be verified on a case-by-case basis, the knowledge of length also contributing to robustness (not just consistently poor/poorer performance) could support further experimentation/replication of any study. Failed attempts to reproduce results may be explainable by erraticity.

One may argue that erraticity may not be relevant when each model is more optimally trained (as opposed to being treated with our one-setting-for-all regime). But we do want to stress that this very stark contrast between erratic and non-erratic behavior is possible, prompting a question on fairness: is there a one-for-all setting under which the languages with non-erratic behavior shown in our study would demonstrate erraticity and vice versa?

To the best of our knowledge, the meta phenomenon of erraticity, as a sample-wise non-monotonicity measured intrinsically with cross-entropy and contributing to large variance across runs, is a novel and original discovery and contribution to research in robustness. We hope our work would inspire further evaluation on other models/architectures, reflection and theories on our assumption of unbounded computation (e.g. Xu et al. (2020)), as well as new understanding and solutions that take data statistics and realistic computational aspects into account. We defer a more comprehensive analysis of erraticity with further experiments to future work.

### M.1    ERRATICITY AS LARGE VARIANCE: EVIDENCE FROM DIFFERENT RUNS OF THE SAME DATA

To confirm that erraticity is not due to data-specific reasons, e.g. when certain data segments might be "easier" to model than others, we show figures from 2 runs (Figs. 18a and 18b) on the same dataset of wildly differing performance that only differ in seed. Note that changes in the y-direction can vary much, indicating large variance across runs.

By establishing that high variance holds across sample sizes, we showcased how it'd be possible to just test on 2 or 3 data points of smaller sizes to get a gauge on the robustness in higher order. It serves as a signal of when the system is being "stress-tested" and hyperparameters need re-tuning. Spot-testing on a couple of smaller data sizes can indeed save much time and energy. Take our run B0 byte models as an example: the training of the $10^2$-line model for EN-RU took 15 minutes, $10^3$ 40 minutes, $10^4$ 1 hour 50 minutes, and $10^5$ 3 hours 36 minutes. One can imagine how these would just be a fraction of training time for bigger models. (Likewise, for our ratio of target training token count to number of parameters — knowing when a representation might be prone to DD within a data size range could help prevent practitioners from prematurely declaring experimental results as negative or from unnecessarily rerunning an experiment because bigger data did not lead to better results.)

### M.2    ADDITIONAL EXPERIMENT WITH LENGTH FILTERING TO 300 BYTES

Figure 19a and 19b show results of additional experiment with subset of data in byte (UTF-8) representation length-filtered to 300, including dev data:

Erraticity remains for AR and RU. Scores are lower, though they cannot be compared with the experiments in the main paper due to difference in dev data size (3,077 lines vs. 1,804 lines here). Number of total lines for train is 5,533,672 lines for each language, from which we took the initial $10^2$-$10^6$. As in our main experiments, we filtered out only whole lines, i.e. not by discarding the tails of longer lines. 300 bytes aren't long sequences, but without data transform or hyperparameter tuning, things can look unfair. The EN translation of the longest RU line in this dataset is: "47. It is

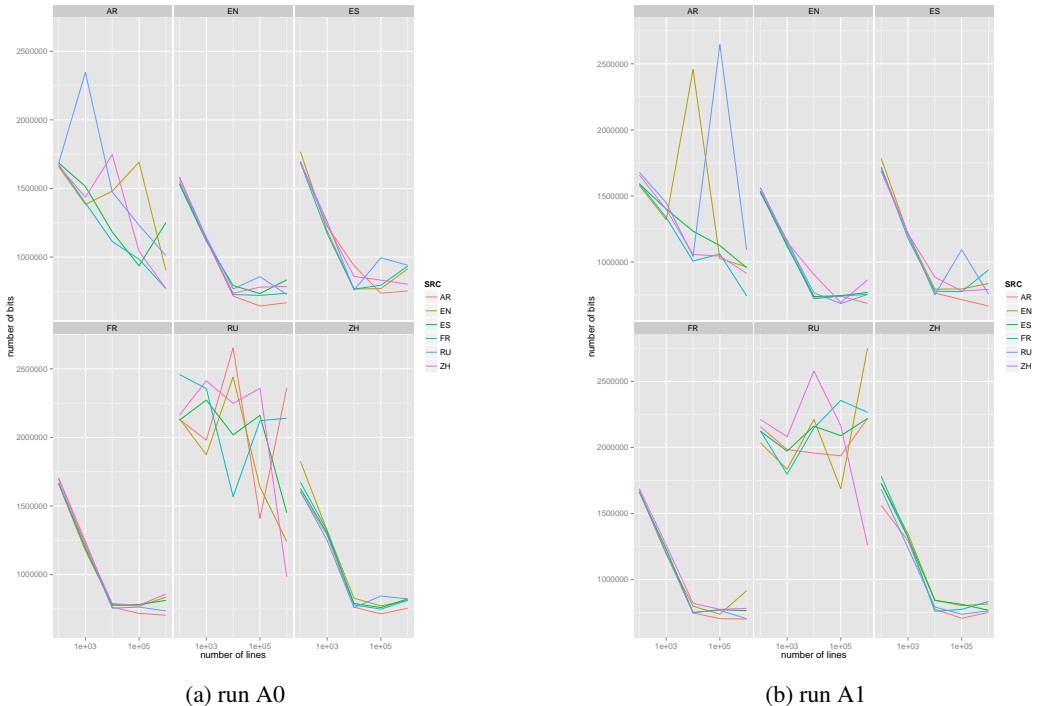

(a) run A0                                 (b) run A1

Figure 18: Same data with differing seeds

noted that there is a lack of information provided by the Government of Trinidad and Tobago with regard to the legal status of the Convention in the domestic legislation."

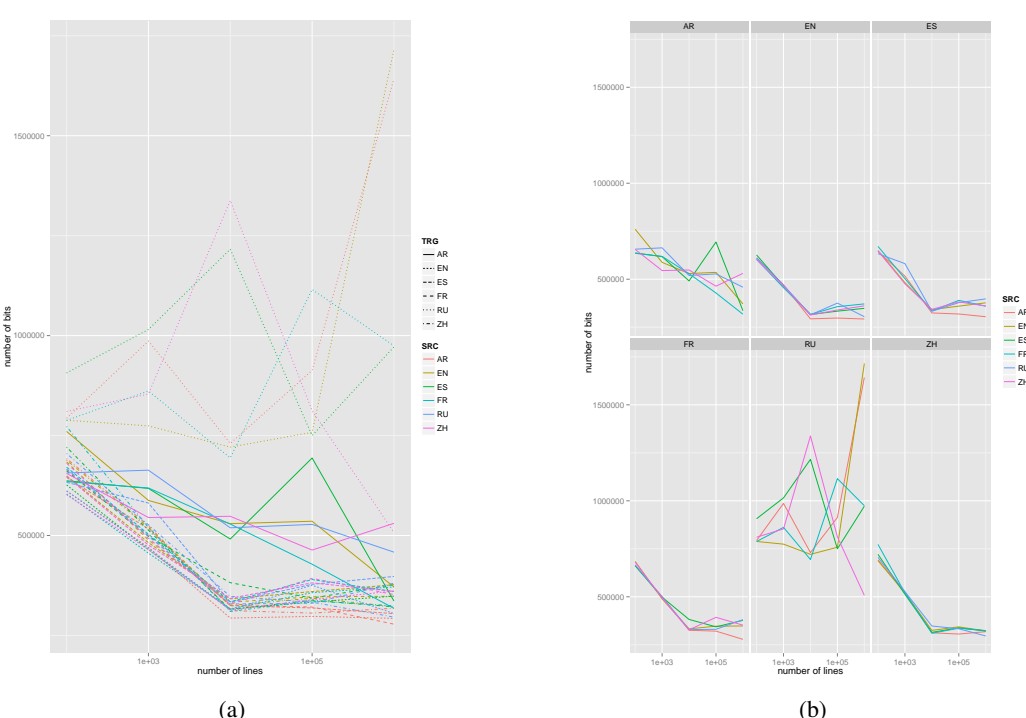

(a)                                                    (b)

Figure 19: Additional experiment with maximum length of 300 bytes (with no hyperparamter tuning, in our blind one-setting-for-all evaluation). Considering there are languages with much higher character sequence length than RU, there is food for thought for the design of next-generation Multilingual Plane.

## N    EXPERIMENTS WITH ONE-LAYER TRANSFORMER

We performed 1 run with dataset A in 4 sizes ($10^2$-$10^5$ lines, seed=13) with the primary representations of characters, bytes, and words, on 1-layer Transformers (num-layers 1:1, all other hyperparameters remain the same as for our main experiments). We compared this against run A0 in 4 sizes with the same seed. (Based on how our null hypothesis is set up, the higher the number of runs, the more likely it is for there to be disparity. Important is that we evaluate based on an equal number of runs and on the same data for all candidates.) Results are shown in Table 5 with no statistically significant disparity observed on the models trained with 1 layer across the board.

Many are under the impression that big data is the cause to the neutralization of language instances in DL/NNs. But, as this set of experiments shows, it is possible for there to be no statistically significant differences between them, with as little as our smallest data size of 100 lines.

Table 5: Number of language pairs out of 15 with significant differences, with respective p-values. $BYTE_{6layers}$ is the representation with erratic $AR_{trg}$ and $RU_{trg}$.

| p-value | $CHAR_{6layers}$ src | $CHAR_{6layers}$ trg | $BYTE_{6layers}$ src | $BYTE_{6layers}$ trg | $WORD_{6layers}$ src | $WORD_{6layers}$ trg | $CHAR_{1layer}$ src | $CHAR_{1layer}$ trg | $BYTE_{1layer}$ src | $BYTE_{1layer}$ trg | $WORD_{1layer}$ src | $WORD_{1layer}$ trg |
|---|---|---|---|---|---|---|---|---|---|---|---|---|
| 0.05 | 0 | 0 | 0 | 6 | 0 | 5 | 0 | 0 | 0 | 0 | 0 | 0 |
| 0.01 | 0 | 0 | 0 | 6 | 0 | 1 | 0 | 0 | 0 | 0 | 0 | 0 |
| 0.001 | 0 | 0 | 0 | 5 | 0 | 0 | 0 | 0 | 0 | 0 | 0 | 0 |

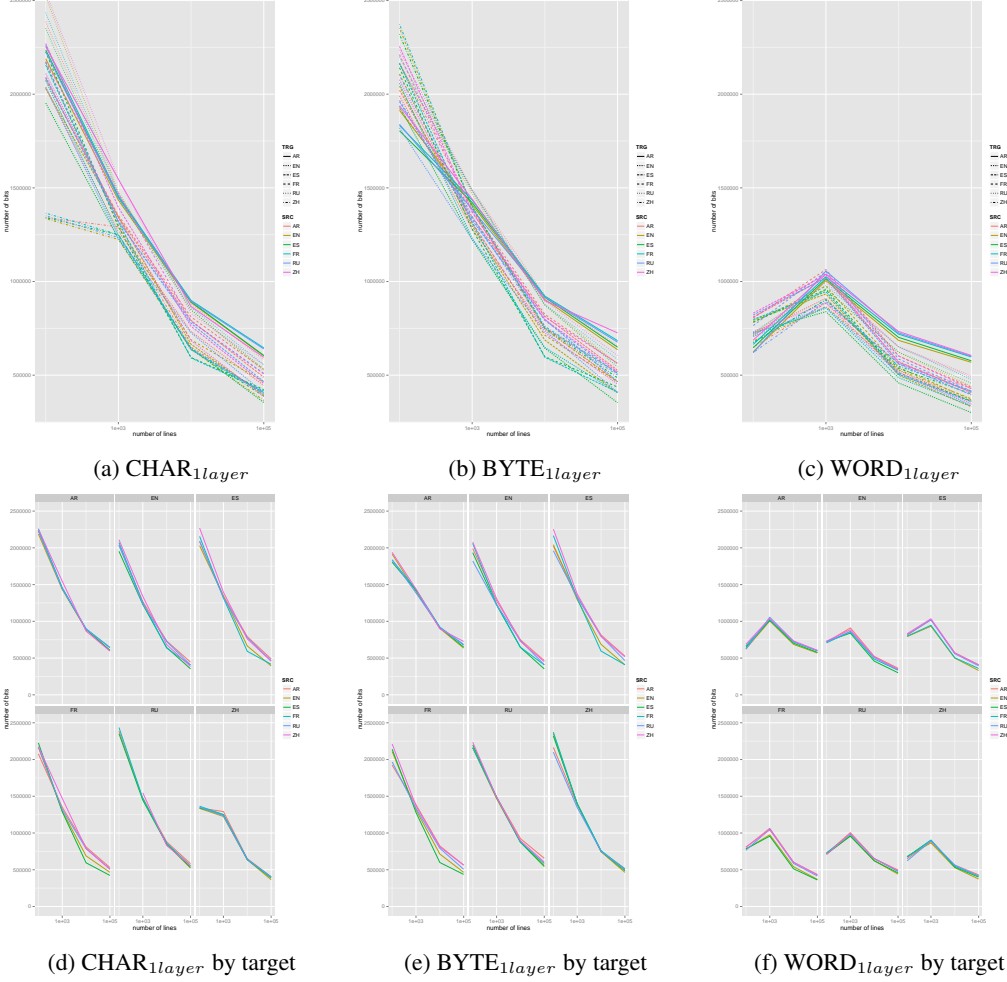

(a) $CHAR_{1layer}$          (b) $BYTE_{1layer}$          (c) $WORD_{1layer}$

(d) $CHAR_{1layer}$ by target    (e) $BYTE_{1layer}$ by target    (f) $WORD_{1layer}$ by target

Figure 20: One-layer Transformer models

## O   PAQs (Previously asked questions)

### O.1   One setting for all

**Q:** Normally, one trains a model with the objective of optimizing based on the training and evaluation data with hyperparameter tuning. The experiments here used one setting for all. Some model configurations might train better and converge close to their optima while other configurations might not reach their full potential. Can this not create a distortion in the results?

**A:** For conventional engineering practice, we agree that hyperparameter tuning would be a sine qua non. However, the evaluation objective is the relational distance between languages, hence we need to see it in a different light. Here is a loose analogy:

\*\*\*

Assume 3 objects in 3 different locations in space.

Relative evaluation from one setting allows one to capture the distance between these objects. It does not matter whether these three objects are in their "best" states.

For example, if one were to use a camera to capture these 3 objects and one does not adjust the setting (using just one random aperture, shutter speed, and focus), i.e. no tuning to capture any of these 3 specifically, nor does one try to model these 3 to their individual bests separately, what would result could be a picture that captures one of these 3 objects more favorably than the others, or it could be that all of these would be blurred. But either way, there is a degree of blurriness to be measured, giving us an idea of the relative distance between the objects. Such relative measurement is the evaluation strategy that our paper adopts.

Now, to add to the camera analogy, say one of the objects is running water, which was extra blurry [erraticity]: we suggest freezing the water, so even from the one arbitrary angle, it could be captured better. And it worked.

Also, while one might generally like to have a "pretty" photo, one that is e.g. taken with sub-optimal lighting, say, overexposure, can have a telling effect as it can bring out details in something dark, like a black box.

\*\*\*

Alternatively, one can tune hyperparameters for each model individually such that each model would be a more optimized one and then compare these models. In that case, one would be interpreting the differences between language in terms of hyperparameters, and the paper would be one that is algorithm-centric. That is of course also a possibility. Our approach, however, is a data-centric one. We would, first of all, like to understand the nature of language data, i.e. what it is about language, if there is anything at all, that makes it a different data type than other data, and what kind of structural constraints, if any, that we need to take into consideration. Then with findings from this data perspective, we try to relate back to the algorithm and make connections so to create a more holistic picture.

### O.2   Translationese / word order

**Q:** Multitexts are parallel texts or translations with the same meaning. There is little to no variation in word order, hence they are just "Translationese" (Gellerstam, 1986). That is why they turn out to be the same, with no performance disparity.

**A:** Our findings do show that when the semantics is properly controlled, such as in multitexts, the factors influencing performance are statistical properties related to sequence length and vocabulary, e.g. $|V|$ or TTR, and the languages tested can be different. Semantic equivalence is also not a reason why we should expect neutralization of source language instances, as that would mean we should expect equal results across target languages.

We agree that faithfulness is often a priority in producing good translations. Whether the translations are produced by humans or machines, only a single best translation can surface as the translation of choice. There may be many other competing hypotheses, but regardless of whether it is done through an automatic ranking algorithm by a machine or through a human expert, the purpose of

translation is the same. However, *styles* and preferences in translations can vary. While faithfulness is generally preferred in the translations of legal texts, more freedom with skillful rearrangement of and play on words (or rather, character or sub-character sequences) or sounds being a criterion for literary texts could be appreciated by certain readers. We agree that it could be very interesting and necessary to model these variations, and we understand that languages can surface in many multimodal forms beyond the confines of texts as well. But with a data-driven perspective, to model this broader variation in language, we need corresponding datasets — we suggest contrast sets where the difference in e.g. sequential order is explicit. And for evaluation, we would require an even more systematic meta evaluation, one that spans different datasets.

But the argument that language or data *could be* different beyond how it appears in one dataset is irrelevant in the evaluation of experiments involving said dataset.

## P  UNDERSTANDING THE PHENOMENA WITH ALTERNATE REPRESENTATIONS (EXTENDED VERSION)

**[Appendix P is an extended version of § 4.]**

To understand why some languages show different results than others, we carried out a secondary set of control experiments with representations targeting the problematic statistical properties of the corresponding target languages.

**Character level**    On the character level, it is well known that ZH differs from the other languages in its high $|V|$, in this study it has an averaged mean±std of 2550±1449[12] across all 5 data sizes from all 3 datasets compared to 170±87 from all other 5 languages combined, may these be in Latin or Cyrillic alphabet or the Abjad script. But what is often not known is that the character sequence length of logographic languages such as ZH is typically short (think and compare the sequence length of the Ancient Egyptian hieroglyphs or the Demotic script with that of the Greek script on the Rosetta Stone). Here in our case, the averaged mean sequence length in characters for ZH is 35±19, compared to 129±71 from the other 5 languages. Heuristics to mitigate high $|V|$ often involve decomposition, which automatically resolve the problem of short sequence length. We tried 2 methods to lower character $|V|$ with representations in ASCII characters — Pinyin and Wubi. The former is a romanization of ZH characters based on their pronunciations and the latter is an input algorithm that decomposes character-internal information into stroke shape and ordering and matches these to 5 classes of radicals (Lunde, 2008). We replaced the ZH data with these formats *only on the target side* and reran the experiments involving ZH as a target language ($ZH_{trg}$) on the character level.

Results in Figure 2 and Table 1 show that the elimination of disparity on character level is possible if ZH is represented through Pinyin (transliteration), as in Subfigure 2c. But Wubi exhibits erraticity (Subfigure 2a). Wubi in our data has a maximum sequence length of 688 characters. As we shall also show in our byte-level analysis below, there are reasons to attribute length as cause to erraticity.

Decomposition into strokes may seem like a natural remedy analogous to decomposing an EN word into character sequences, but one needs to be mindful of not exceeding an optimal length given finite computation. Considering the ZH in the UN data is represented in simplified characters, decomposing traditional characters would surely complicate the problem. As there are also sub-character semantic and phonetic units (Zhang & Komachi, 2018) that can be exploited for information and aligned with character sequences of other alphabets, qualitative advances in this area can indeed be a new state of the art.

**Byte level**    On the byte level, we observe irregularity for AR and RU. We find minimum sequence length of the target language to be one of the highest metrics correlating positively with the total number of bits ($\rho = 0.60$).[13] Our data is based on 300 characters as maximum length per line. While we wanted to retain at least 75% of the UN data after length filtering, this length still renders a maximum sequence length that exceeds 100 words (the default maximum length for the word alignment model, GIZA++ (Och & Ney, 2003), in the traditional SMT pipeline). Translated into bytes with UTF-8 encoding, data with 300 characters maximum gives us, e.g. for the $10^6$-line datasets, an averaged mean±std of 185±106 in length for AR and 246±142 for RU, considerably larger than that for ZH (94±53) and for EN/ES/FR ($\approx$145.41±77). With UTF-8 encoding, each character in AR, RU, and ZH contains 2 or more bytes. ZH typically has shorter line length in characters, compensating for the total byte sequence in length, even when most ZH characters are 3 bytes each. However, AR and RU generally have long line length in characters, so when converted to bytes, the sequence length remains long even when most of the characters might be just 2 bytes each. Results from our pairwise comparisons indicate 8 (non-directional) language pairs to be significantly different (see Table 1 under "BYTE"): ES-RU, EN-RU, FR-RU, RU-ZH, AR-RU, AR-EN, AR-ZH, and AR-FR — all involving AR or RU. (Appendix I lists also the language pairs with significant differences for other representations.)

---

[12]Figures are rounded to whole number. Complete tables of data statistics are provided in Appendix D.

[13]Top-3 correlates for each representation can be found in Appendix F.

Leveraging language-specific code pages can be a useful practical trick, a reminder that there are alternatives to UTF-8 for analyses and back-end processing if data is clean and homogeneous and if success of larger-scale prediction is not a concern. But one more sustainable alternative is to design a more adaptive and flexible character encoding scheme in general, taking into account the statistical profiles such as length (wrt characters and bytes) and sub-character (atomic/elementary/compound) information of all (or as many as possible) of the world's languages.

**Word level** The main difference between word and character/byte models is the absence of length as a top contributing factor correlating with performance. Instead, what matters more are metrics concerning word vocabulary, with top correlate being OOV token rate in the target language ($\rho = 0.66$). This is understandable as word segmentation neutralizes sequence lengths — the longer lengths in phonetic alphabetic scripts are shortened through multiple-character groupings, while the shorter lengths in logographic scripts (cf. difference in length for the 3 scripts on the Rosetta Stone, logographic scripts are typically shorter than phonetic ones) are lengthened by the insertion of whitespaces. To remedy the OOV problem, we use BPE, which learns a fixed vocabulary of variable-length character sequences (on word level, as it presupposes word segmentation) from the training data. It is more fine-grained than word segmentation and is known for its capability to model subword units for morphologically complex languages (e.g. AR and RU). We use the same vocabulary of 30,000 as specified in Junczys-Dowmunt et al. (2016). This reduced our averaged OOV token rate by 89-100% across the 5 sizes. The number of language pairs with significant differences ($p \leq 0.001$) reduced to 7 from 8 for word models, showing how finer-grained modeling has a positive effect on closing the disparity gap.

Version 1.1 (graphs to be updated, score tables added)

