# OpenReview forum: "Representation and Bias in Multilingual NLP: Insights from Controlled Experiments on Conditional Language Modeling"
_ICLR.cc/2021/Conference — Reject_

### Official Review · AnonReviewer2 · 2020-10-22
**An interesting and thorough exaimination of an important problem. Writing is sometimes too complicated and some more high level analysis could help.**

**Rating:** 6
**Confidence:** 4

**Review:**

The paper provides an empirical investigation of an important problem: the transferability of language modeling signals across languages in the transformer model. This is an important question because it can teach us both on the relations between languages and the properties of the transformer model (although it is not easy to tease the two effects apart).

This is a thorough paper with a large number of experiments and with interesting conclusions that nicely generalize the low level patterns observed in the experiments. These conclusions are likely to be useful for the research community as part of its on going investigation of language transfer and the transformer model.

I have several comments though:

1. The language of the paper is often very complicated. Just as a couple of examples: It was very hard for me to follow the abstract, the first paragraph, the (very long) sentence that start with "in order to eliminate" (1.1), item 3 in the list of contributions and this is just a partial list.  I ask that if the paper is accepted the authors will try to improve this aspect.

2. The writing is often over pedagogical and I often got the feeling that the authors try to educate their readers (but not in the positive sense of the word). I would try to avoid this style.

3. This work seems highly relevant to the following paper:

"Towards Zero-shot Language Modeling." Edoardo Maria Ponti, Ivan Vulic,Ryan Cotterell, Roi Reichart and Anna Korhonen . EMNLP 2019

I think the discussion parts can gain form comparing the conclusions of the two papers, when relevant.

---

> ### Author Response · Authors · 2020-11-13
> **first response to your review (1 of 2)**
>
> Thank you, Reviewer 2, for your appreciation of our efforts.
>
> 1. Re the language: thank you for your feedback. Part of our "language"/formulation is unintentional and we are in the process of reformulating some long sentences (e.g. in the abstract and in 1.1). But part of it has to do with the fact that we need(ed?) to be very careful with our wording, as though we are writing legalese (e.g. item 3 in Sec 1.2). Our findings can be viewed as in conflict of interests with linguistic typology (though that is not necessarily the case), which has become a dominant trend in multilingual NLP. In such context, we would be voicing a minority view that can be interpreted as "irreverent" by some, causing the paper to run into either disregard or potential rejection by those who reviewed out of self-interests instead of the interest of the community as a whole.
> (But we firmly believe in our cause to bring some fairness, diversity and inclusion into our multilingual practice. Some things need be said (e.g. Sec 1.2) and we are "singing our blues" --- providing representation --- for the "morphologically rich" as well as "morphologically poor".)
>
> What formulation would you suggest for our "quasi-legalese" for item 3 in Sec 1.2?
>
> Re the first paragraph of the paper: it is written to clarify that this is a bridge paper and to "defend". Some NLP practitioners have a habit to validate claims with downstream task results and with usefulness. And we are trying to move our community members away from that tradition, or at least create a space with a more scientific (as opposed to engineering) mindset. Scientific insights and contributions do not require topping the leaderboard or using SOTA techniques (or even being "useful"). A scientific contribution is to improve our understanding of a phenomenon in the world [1][2]. We take on this task, however, as use-inspired basic research, i.e. research in the Pasteur's quadrant [3] to bridge the gap between basic and applied research, as a quest for fundamental understanding with some consideration of use. (So if one gets the impression that this work is not really in the traditional language sciences or linguistics space, but a bit different from customary NLP work (except maybe as a blackbox paper?), yet less theoretical than a typical ML paper (perhaps like data science?), then they got it. That is exactly the space that we are targeting and DL/NNs help enable the bridging of all of these areas.)
>
> As we are trying to break new ground here, what would you do re the first paragraph if you were us?
>
> 2. Re tone being overly pedagogical: we were originally expecting a more diverse, interdisciplinary mix of readers/reviewers. We would like to improve on this tone issue if it bothers you. Would you please provide us with some (or at least one) example(s) and suggest an alternate formulation? Thank you.
>
> 3. Re Ponti et al. (2019) (hereafter [P19]): thank you for the reference. We will definitely mention this in related work. We were/are a bit hesitant when it comes to comparing with work that do not include ZH (or any logographic languages [4]). The authors of [P19] explicitly "exclude languages that are not written in the Latin script" even though ZH is available from the dataset they used (the Bible corpus from Christodouloupoulos and Steedman (2015)). And aware as we are of the qualitative biases (against ZH) in linguistics, we are more interested in methods that are fairer and more general. (ZH is a language for which not only is the notion of word a contested subject, as mentioned in the paper, the notion of parts of speech like nouns and verbs is also highly questionable [5,6]. Some word-based fundamental issues have been ignored/overlooked. And the only methods so far that put ZH on an equal footing with the other languages seem to be i. transliteration (but it is at the expense of a more native logographic representation), ii. byte-based processing, and iii. hyperparameter tuning.)
> Though saying all this or comparing with [P19] would seem like going off on a tangent a bit --- in the first place, we do not intend for this paper to be an application paper, nor do we want to make it like a paper comparing absolute scores [7] of systems with and without domain knowledge [8] for a narrower range of languages [9]. We would, instead, really like to set precedence in working in a new transdisciplinary space where we can relate concepts in an unsupervised or less supervised way, e.g. using Double Descent to do "new" science, making discoveries like erraticity.
>
> 4. Again, we appreciate your support. What can we do to get you to improve your score? We are in the process of revising. If there are any expressions that you find hard to read, please do not hesitate to let us know. Thanks.

---

> > ### Author Response · Authors · 2020-11-13
> > **first response to your review (2 of 2)**
> >
> > (cont'd)
> > [1] https://2020.emnlp.org/blog/2020-05-17-write-good-reviews
> >
> > [2] In our case: how can we help improve our understanding of the relationship between language, data representation, size, and performance in CLMing with the Transformer? As this can be a big question, we formulated the problem statement for this paper as "are all languages equally hard to CLM?".
> >
> > [3] https://en.wikipedia.org/wiki/Pasteur%27s_quadrant
> >
> > [4] "logographic languages" = "languages with logographic scripts"
> >
> > [5] Is it Harder to Parse Chinese, or the Chinese Treebank? Roger Levy, Christopher D. Manning. ACL 2003. https://www.aclweb.org/anthology/P03-1056.pdf
> >
> > [6] An Empirical Examination of Challenges in Chinese Parsing. Jonathan K. Kummerfeld, Daniel Tse, James R. Curran, Dan Klein. ACL 2013. https://www.aclweb.org/anthology/P13-2018.pdf
> >
> > [7] We adopted a novel relational evaluation method.
> >
> > [8] Cf. a related view from Rich Sutton, when asked about incorporating domain knowledge as soft priors in the Q&A after his talk at the ICML 2020 4th Lifelong Learning Workshop in video at 8:59:30-9:00:38 at https://icml.cc/virtual/2020/workshop/5735#collapse7535.
> > Our view is different from his, but we share his objective in contributing to a science of machine intelligence. That said, there is a caveat here. There are still scripts out in the world from less documented languages which have not been added to Unicode. So we do see the opportunity to optimize with some fair measures (as suggested in the paper: complementing character encoding with language statistical profiles), as that could also benefit non-neural algorithms as well as basic text processing. (This is a negligible concern for all of us here who are privileged enough to partake in an ML conference, but that may not be the case for many parts of the world where, e.g. one may have to struggle to get internet or may only have access to (older) digital devices.)
> >
> > [9] Although we "only" studied 6 languages in the paper, our data cover a broader range in diversity than that studied by [P19]. ZH and AR/RU are at the opposite ends of the spectrum* --- both on "word" level, in terms of traditional morphological analyses, and on the character level, with logographic languages being the outliers of the world's languages.
> > (* We agree that some agglutinative languages could be longer than RU/AR, hence at the further end of the spectrum beyond RU, i.e. with longer length or "morphological complexity". But there are no multi-way parallel data (multitexts) of larger sizes that would support our study.)

---

### Official Review · AnonReviewer3 · 2020-10-28
**Interesting paper and experiments, but lack of evidence to support its claims.**

**Rating:** 5
**Confidence:** 4

**Review:**

The paper investigates whether languages are equally hard to Conditional-Language-Model (CLM). To do this, the authors perform controlled experiments by modeling text from parallel data from 6 typologically diverse languages. They pair the languages and perform experiments in 30 directions with Transformers, and compare 3 different unit representations: characters, bytes, and word-level (BPE).

I appreciate the authors' effort for their systematic controlled experiments. However, I'm leaning towards rejecting this paper since I think some of the claims made in the paper are too strong and not really backed up by their experiments.

Some comments:
* The term "Conditional-Language-Model" can be misleading, since this paper model a target language conditioned on a source language, so more like in a machine translation setting rather than standard language modeling setting where you can also condition on the previous history.
* I'm also not sure if comparing perplexity by conditioning on another different language (source language) is correct. The experiments would be clearer if done with standard LM with Transformers encoder model like BERT for example.
* At the end of Section 2, the authors mention about "generalizations", but I couldn't really find any discussion about this in the paper. Maybe this can be clarified?
* I found that claiming script bias in character models is too strong, if the experiment only shows bias in ZH (and this is expected since its character-level has different notion with languages with Latin script).
* Byte-level: I found that there is still some "erraticity" in Figure 3(b) especially when the data size increases (which is more practical in real world application), so this is not entirely resolved?
* I also think the summary in Section 1.2 stating that linguistic typological information is not necessary given "statistical properties concerning sequence length and vocabulary" is not necessarily valid since these two properties are the results from linguistic typology information.


Missing references:
1. From characters to words to in between: Do we capture morphology? Clara Vania and Adam Lopez. ACL 2017.
2. Multilingual Part-of-Speech Tagging with Bidirectional Long Short-Term Memory Models and Auxiliary Loss. Barbara Plank, Anders Søgaard and Yoav Goldberg. ACL 2016.

---

> ### Author Response · Authors · 2020-11-12
> **first response to your review (1 of 2)**
>
> Thank you, Reviewer 3, for your review. We would love to have you swing towards (strongly) accepting and supporting our paper.
>
> To your concerns:
>
> 1. Re the term "CLM" being misleading: we adopt a definition of CLM that is rather standard in the DL tradition. For example, according to [1,2], CLM is to be differentiated from an unconditional LM in that unconditional LMing is the modeling of the probability of the next token, given the history of the preceding tokens, while CLMing is the modeling of the probability of the next token, given the history of the preceding tokens *and* conditioning context. In our case, such conditioning context is a line from the source language. But in, e.g. the case of [3], this conditioning context can be in other modalities. We used a standard neural sequence-to-sequence modeling setup, as you said, like for NMT. But different from NMT, we are not concerned with translation output as we would like to eliminate the confound in search for generation. We focus on an intrinsic evaluation in perplexity/cross-entropy of our CLMs.
>
> 2. Re "I'm also not sure if comparing perplexity by conditioning on another different language (source language) is correct": we are not sure why it would be correct or incorrect. We would like to do an intrinsic evaluation of CLMing, like in an NMT setup --- that *is* the definition of our task, of our investigation. Re "[t]he experiments would be clearer if done with standard LM with Transformers encoder model like BERT for example": we disagree. Our setup enables a better understanding of the encoder-decoder model as we are able to perform more systematic controls on data size, representation, and language on both source and target sides.
>
> 3. Re "generalizations": by that we mean the findings in Sec 1.2., e.g. representation relativity, source language neutralization. (To that we will also add, to the next revision, one obvious observation that we had mentioned in the main text but did not list in Sec 1.2: "[r]epresentational units of finer granularity can help close the gap in performance disparity".)
>
> 4. Re script bias being too strong of a claim: you are right, script bias is in the data. We should have made this more obvious.
>
> We stated already:
> i. in Sec 1.2 "[b]igger/overparameterized models can exacerbate the effect of data statistics. Biases that can be expressed quantitatively and lead to disparity are mitigable through hyperparameter tuning";
> ii. then we alluded to this again in Sec 7 (conclusion) "[i]t will take everyone’s effort to mitigate the bias in ourselves..."; and
> iii. we also provided an analysis of the data statistics in App. O.
>
> But we see that we should clarify "bias" more explicitly instead in the main text. (Some of our previewers did not feel comfortable with the concept of "human bias", but, we have already come this far... we will revise accordingly.)
>
> The bias that we are most interested in addressing in this paper (as apparent in the one-sentence summary for our paper) is word bias, which is a human bias. It is up to us to see and process languages with or without the concept of a "word", a concept that had given rise to a hierarchy that is not necessary. That is the primary intended reading of the "bias" in our title "Representation and Bias".
>
> (A second, more general, possible interpretation of "bias" is that we have a choice in adopting whichever perspective we wish to see languages in --- whether they are fundamentally different (main experiments results) or similar (when we "zoom out" and see no differences, as in results from App. M). It can all be just a matter of perspectives.)
>
> 5. Despite how things looked still a bit non-monotonic in Fig. 3b, disparity was eliminated. In larger/overparameterized settings for our main experiment, we are already stretching it with the length --- when we do not further tune our hyperparameters. As can be seen in App. L2, even 300 bytes for this setting would cause erraticity. But as mentioned in the last paragraph in Sec 5 and can also be seen in App. M, erraticity can be resolved and a monotonic development is possible when we adjust our hyperparameter setting, e.g. by decreasing the depth of the models to one layer.

---

> > ### Author Response · Authors · 2020-11-12
> > **first response to your review (2 of 2)**
> >
> > (cont'd)
> > 6. Re statistical properties concerning sequence length and vocabulary being the results from linguistic typology information: no, these properties do not result from linguistic typology. They are from the language data themselves (see App. D). And linguistic typology itself is also a result of language data, though with word bias from our academic tradition.
> >
> > As to whether it is possible for there to be correlations between statistical properties and symbolic linguistic typological concepts on the "word" level, our paper does not comment on that at all. That is beyond the scope of the present paper.
> > We are pleased to have been able to sort out when linguistic concepts do apply and to have reached the conclusion we did in point 3 in Sec 1.2. That is sufficient to us and for this paper.
> >
> > 7. Re the additional references: thank you and we will add these to our related work in either the main text or App. P, as space allows.
> >
> > We will be uploading a revision in the coming days.
> > If you have any further questions or feedback, or input as to whether we should move certain information from the main paper to the appendices (or vice versa), please do not hesitate to let us know.
> >
> > [1] https://github.com/oxford-cs-deepnlp-2017/lectures/blob/master/Lecture%207%20-%20Conditional%20Language%20Modeling.pdf
> > [2] A summary can also be found on p. 1 of Jamie Ryan Kiros' PhD thesis on Conditional Neural Language Models for Multimodal Learning and Natural Language Understanding (2018).
> > [3] Multimodal Neural Lnaguage Models. Kiros et al. ICML 2014. http://proceedings.mlr.press/v32/kiros14.pdf

---

> ### Author Response · Authors · 2020-11-18
> **v1.0 and your review**
>
> Dear Reviewer 3:
>
> v1.0 has just been uploaded.
>
> - We believe our formulations are now "cleaner", and we clarified and discussed bias more explicitly in this version.
> - Item 3 in §1.1 (previously §1.2) has also been revised.
> - Unfortunately, we did not have enough space for the additional references you recommended as it seemed like we might have to go off course a bit to talk about the explicit modeling of linguistic concepts and the evaluation thereof (vs the evaluation of a more implicit modeling through LMs/CLMs). There is a difference between these 2 objectives (which could make for a separate paper on its own). But we did consider, only to realize that we might end up confusing readers if we provide too much information. However, if you should think that including those references would be important and relevant for our objective, or if there is anything that you think we could improve on, please do not hesitate to let us know.
>
> We look forward to hearing from you.
>
> Thank you and best regards,
>
> authors

---

### Official Review · AnonReviewer1 · 2020-10-28
**Interesting but raises questions**

**Rating:** 4
**Confidence:** 4

**Review:**

This paper is trying to answer an important question: How does representation play a role in carrying meanings? In doing so, the authors experimented with 6 languages in  3 + 5 kinds of representations. The authors concluded that the different performances among language pairs maybe be the result of word segmentation in different ways.

This is an interesting step towards understanding meaning representations, especially in languages that do not have an alphabet. However,  as much as I agree with some of the final conclusions, the soundness of the experiments appears to be in question.

My main concerns are about the additional experiments with Chinese.

The authors claimed that "On the character level, target language ZH (ZHtrg) shows a different learning pattern throughout." There are two types of character-level representations used: Wubi and Pinyin. Wubi was originally invented for professional typesetters so that they can type fast. The segmentation may not be correlated with the meaning of the word at all, as claimed by the papers cited by the authors. Pinyin, on the other hand, is highly ambiguous. One pinyin may representation dozens of words and the authors did not take tones into considerations at all.

The author also mentioned that "After filtering length to 300 characters maximum per line in parallel for the 6 languages, we made 3 subsets of the data with 1 million lines each". Each language carries meaning differently and the information density is drastically different. 300 characters in Chinese carry much more information than 300 characters in English. This is an unfair comparison.

It would make a lot more sense if the authors treat each language differently because of their orthographic differences.

---

> ### Author Response · Authors · 2020-11-11
> **first response to your review**
>
> Thank you, Reviewer 1, for your review.
>
> We did not mention "meaning" or "meaning representations" in our paper, nor was it our intent to model meaning explicitly. We investigate the relation between language, data representation, size, and performance in the context of Transformer CLMs through the research question of whether languages are equally hard to CLM. One could think of our effort as estimating conditional probabilities with the Transformer.
>
> In fact, when we filtered our data for the 6 languages, we did so in lockstep/parallel (hence it is a fair comparison) [1]. Semantic content has been held constant. Meaning is an independent variable of our experiments. Re "300 characters in Chinese carry much more information than 300 characters in English": we understand, which is why we see the disparity on the character level between ZH and the other languages. A ZH character carries more information than an EN character does, hence sequence length in ZH is generally shorter than that in EN. Important for our comparisons is that we are evaluating parallel sets of lines.
>
> To your concerns re the additional experiments with ZH:
>
> Re Wubi --- "[t]he segmentation may not be correlated with the meaning of the word at all, as claimed by the papers cited by the authors": we do not dispute that one stroke can be meaningless, but a unit made up of several strokes can be meaningful. One might argue that there is a limit to how much semantic information one can obtain through alignment [2] of sub-character (or subword) units, because not all sub-character/subword units (in any language) are meaningful units. We agree, there is a limit to any kind of morphological analyses or to a mapping between form and meaning. Yet it can provide an additional source of information and can be a good pedagogical exercise for students.
>
> Re Pinyin: the implementation of Pinyin we used is with lexical tones (see App. B). Re the ambiguity in Pinyin: yes, we are aware of the limits of symbolic approaches for ZH being a real-world problem. Even if lexical tones were given, the lack of account of tone sandhi (tone changes in sequences) can pretty much only be effectively overcome by continuous representations at scale (by processing raw audio data when modeling sounds). This is also one reason why we try to advocate, in general, a more statistical approach to handling multilingual data, as opposed to relying on traditional linguistic typological resources.
>
> We recognize that, despite how ZH may be viewed as a high-resource language, it lacks a more native/flexible account in traditional language science. And we are working towards a more diverse and inclusive science, with this work being part of such initiative. We appreciate your support.
>
> If there is any part of our paper that still seems unclear to you or that you could help us improve, we'd appreciate it if you could be more specific in pointing out the relevant sections/sentences. Thank you.
>
> [1] When we filtered our data for length, we ensured that it remained fully parallel across all 6 languages. Because of this, every ZH line in characters has a translation in each of the other 5 languages whose line length does not exceed 300 characters.
>
> [2] Alignment can be useful for good translation and essential for the automatic compilation of lexicographic resources --- with neural and non-neural algorithms. For finer-grained alignment of a sub-character component in a logographic language and sub-"word" component (i.e. character or sub-character strings, depending on the script) in a non-logographic language, we cannot do so without breaking away from the traditional/convention notion of a "word" that tends to be centered on EN or a notion that relies on whitespace tokenization only. Examples: 1. in aligning the 鵝 in 企鵝 meaning 'penguin' in ZH (literal: 'stand'+'goose') with the gans 'goose' in vetgans (the Dutch word for 'penguin', lit. 'fat'+'goose'); 2. in aligning the ева part in королева (RU for 'queen') with the 女 (designating 'female') in 女王 (王 means 'king'); 3. in extracting the 鳥 'bird' part from 鵝 'goose'.

---

> > ### Comment · AnonReviewer1 · 2020-11-12
> > **A clarification question**
> >
> > * When I said "meaning", I meant any information that is carried and conveyed through the different representations. This is probably a loose definition, but do you think this is something you are trying to address?

---

> > > ### Author Response · Authors · 2020-11-12
> > > **Meaning vs. information**
> > >
> > > If by "meaning" you meant "information", then that should indeed, if we understand your formulation correctly, be more fitting.

---

> > > > ### Author Response · Authors · 2020-11-19
> > > > **Meaning vs. information (cont'd)**
> > > >
> > > > Dear Reviewer 1:
> > > >
> > > > When we first replied to your review, we did so assuming you meant "meaning", not "information", hence there is a need to adjust our answer:
> > > > re Wubi "we do not dispute that one stroke can be meaningless, but a unit made up of several strokes can be meaningful": we think that, for information, every stroke counts. So even when a Wubi segment does not seem meaningful, it can carry / carries information content.
> > > >
> > > > We have uploaded v1.0 and would be pleased if you wouldn't mind letting us know whether you're satisfied with it.
> > > >
> > > > Thank you and best regards,
> > > > authors

---

### Official Review · AnonReviewer4 · 2020-11-03
**Good premise; Unclear Paper Focus**

**Rating:** 3
**Confidence:** 4

**Review:**

Summary: The authors attempt to investigate to what extent languages are hard to conditionally language-model. They do this by using some information theoretic measures. Claims:
- There are no statistically significant differences between source language representations, but there are significant difference between pairs of target language representations.
- There is no complexity that intrinsic to a language except its statistical properties concerning sequence length and vocabulary (unless word-based methods are used).
- They also observe phenomena such as Double Descent and erraticity.

----
Strengths:
- The Experiments are extensive.
- The relative similarity of source language representations is interesting and worth exploring further.

Weaknesses:
- The diagrams are difficult to read
- The paper is hard to follow and would benefit from a clearer focus rather than the broad range of topics covered here. For example:
    - It is difficult to understand what the methods/terms (the information theoretic measure used, double descent) are - little time is spent explaining these.
    - Double descent is discussed in the paper but it is still made not clear why this is relevant in the paper.
    - Several portions of text are repeated - with some editing, space can be made to discuss concepts important to the paper
- The authors make recommendations for modeling (Eg. using char level or byte level models for certain models - which have been extensively studied for this): this is not followed up with any concrete results on translation/downstream tasks or pointing out relevant work.

---

> ### Author Response · Authors · 2020-11-11
> **some quick answers for now, to be followed up with more comprehensive ones (depending on your responses) and/or revisions**
>
> Thank you for your review.
>
> 1. Re diagrams: do you find the diagrams in App. F still too small or difficult to read? As stated in the 1st paragraph of Section 3 (v0.1): "[w]hat should be considered relevant results for our investigation is the number of language pairs with significant differences reported in Table 1, the general patterns of (non-)monotonicity and disparity in the figures, and the corresponding analyses", one does not need not be concerned with the absolute scores of the experiments directly. The results are summarized in the disparity tables (Table 1 (and Table 5 in App. M)). That said, we can further enlarge the diagrams upon your feedback.
>
> 2. Re "a clearer focus rather than the broad range of topics covered here": this paper can be viewed as a "bridge paper", connecting ideas in disparate fields [1]. Namely, we would like to not only "bridge an understanding between language science and engineering" (Sec. 2), but also connect concepts in language or language data with those in DL/NNs (Sec 1, and also in our keywords "science for NLP", "fundamental science in the era of AI/DL"). This latter keyword was a workshop title at ICLR 2020, though language science was not one of sciences being discussed. Just as phonologists have described and drawn parallels between the interaction of symbolic representations of sounds and cellular processes in biology, we think that there are plenty of opportunities to further language science and a statistical science for NLP with DL/NNs.
>
> 2i. Re "[i]t is difficult to understand what the methods/terms (the information theoretic measure used, double descent) are - little time is spent explaining these": the information-theoretic measure used is cross entropy (Sec 2.1 and App. B). We will add a brief description of DD in the main text. Sample-wise double descent (coined by [2]) is, in short, when performance gets worse with increasing data size and then gets better. Double Descent (DD) has been a rather popular topic this past year at ML conferences, including many submissions for ICLR 2021 as well. Most work, however, concentrate on the theoretical aspects, but there is another parallel submission advocating that the emergence of DD "is due to the interaction between the properties of the data and the inductive biases of learning algorithms" [3]. This seems to be a timely corroboration of our findings.
>
> 2ii. DD is relevant because:
> a. our (more general) goal is to improve our understanding and expectation of the relationship between language, data representation, size, and performance (abstract); and
> b. we also made the connection between words and ZH_trg in characters, both as "non-atomic" units that can be further decomposed (Sec. 5, DD) and as representations that are prone to exhibit DD. This can affirm the interpretation of wordhood for those ZH speakers who identify ZH characters as words (Sec. 3 last paragraph and App. I).
>
> (Similarly, erraticity is relevant due to [a] above and also because it affects performance disparity --- if a language has erratic performance and another doesn't, the difference is likely to show as statistically significant.)
>
> 2iii. Re "[s]everal portions of text are repeated - with some editing, space can be made to discuss concepts important to the paper": the initial version was written rather "defensively". There are many novelties (concepts, styles, approaches) being introduced in this paper, hence we repeated some things that seemed newer or more important. What are some of the repeated items that bothered you?
>
> 3. Re related work: do you find the related application references cited in our App. P sufficient? If not, what do you think is missing? (We are not sure if your comment was reflective of your including/excluding the information from App. P.)
> Re downstream task results: as stated in the beginning of the paper, this is not an application paper. There are quite a few substantial scientific contributions made in this paper already. We would like to first focus on building bridges with this paper, sorting out representations and what holds when.
>
> 4. Last but not least, we would like to understand the reasons behind your score assignment. If there is anything else that we could do to help us win your support of our work, please do point that out to us. Thank you.
>
> [1] from https://nips.cc/Conferences/2020/PaperInformation/ReviewerGuidelines (under Review content 1)
> [2] Nakkiran et al., ICLR 2020: https://openreview.net/forum?id=B1g5sA4twr
> [3] https://openreview.net/pdf?id=nQxCYIFk7Rz

---

> > ### Comment · AnonReviewer4 · 2020-11-24
> > **Response**
> >
> > 1. The grey background with the lighter colors is difficult to read:  I'd recommend (1) more contrast (Eg. white and darker colors) (2) larger axes and thicker lines. I appreciate that you have used line styles and color to help readability
> > 2. Reg. i. An more detailed description of Double Descent would be useful, even if in the Appendix
> > ii.
> > 3. I'm not sure which section you're referring to:  in the current version the App O is the last section. Are you referring to the changes in related work?

---

> > > ### Author Response · Authors · 2020-11-24
> > > **Response from 24Nov2020 from Reviewer4**
> > >
> > > Thanks for your feedback.
> > >
> > > Re 1: we will adjust the graphs.
> > >
> > > Re 2: we had in a previous non-published draft the sentence "Other related work and more recent development on DD can also be found in their (Chen et al. 2020 [1]) work" as the last sentence of the first paragraph under Sample-wise DD in §5. We had taken that out due to space. We can fit the sentence back in, as they provide a rather succinct summary in their App. A and their version has just been revised yesterday. Would this proposal of ours suffice (if not, would you please specify what else on DD you would like to see addressed)? Only sample-wise DD is relevant for our paper and nobody, to the best of our knowledge, has examined/analyzed this from the data perspective.
> > >
> > > Re 3: yes, in version 1.0, App. O is the last appendix. We took App. P (extended version of related work) from v0.1 out (the one imported on 10Nov2020 [2]) out because we did not want to overwhelm reviewers and readers. The related work section in v.1.0 on contains a more succinct summary of the most relevant related work.
> > >
> > > The sentence in the now-obsolete App. P on related application references that we were referring to in our previous reply was:
> > > "Although end-to-end, sequence-to-sequence application papers leveraging full subword representations have been plenty — with characters (e.g Lee et al. (2017)), bytes (Gillick et al., 2016; Costa-jussà et al., 2017; Li et al., 2019a), and BPEs (most NMT papers from 2016 on), there has been no systematic or statistical evaluation of these representations with an explicit effort to relate these to the statistical profiles (not traditional word-based linguistic categories) of diverse languages and to complement the statistical profiles with equitable measures in representation for fairness."
> > >
> > > In v1.0 and up, it has been paraphrased and put into §3 3rd paragraph (at the bottom of p. 4): "A practical takeaway from this set of experiments: in order to obtain more robust training results, use bytes for ZH (as suggested in Li et al. (2019a)) and characters for AR and RU (e.g. Lee et al. (2017))...".
> > >
> > > Also, we were about to re-inquire with you about your expectation of downstream task results because you had asked about these. We already clarified in our initial reply that this is not an application or NMT paper (as stated explicitly in the paper itself).
> > > One of our main objectives is to get people to examine their models and foundational assumptions carried over from traditional science(s) more critically and to look at data statistics more closely, instead of just focusing on quantitative results or calling neural networks a black box. And we'd appreciate it if you would please let us know if your expectation of downstream results is still relevant or if it's been updated given our elaboration on the main objective of this paper. One of our keywords is "science for NLP" --- we wanted to advocate a *science* for NLP for understanding, as NLP has been primarily *engineering*-focused.
> > >
> > > We'd appreciate your confirmation. Thank you again.
> > >
> > > [1] https://arxiv.org/pdf/2008.01036.pdf (App. A)
> > > [2] https://openreview.net/references/pdf?id=55tHd7KuIU

---

### Author Response · Authors · 2020-11-10
**v0.1 uploaded: combined main paper and appendices (from the original submission, not yet revised)**

Dear Reviewers:

Thank you for your reviews.

We have uploaded the originally submitted supplementary material (the Appendix section) together with the original main paper as a combined PDF (for future reference, we will refer, if necessary, to this combined version as v0.1) for reading convenience. Please note that this version is not yet revised.
Some of the concerns raised in the original reviews were already addressed in the appendices. We kindly invite and encourage all our readers (reviewers and general readers alike) to read and review the paper in its entirety.

We will reply to each of your concerns as expressed in this first set of reviews in the coming day(s) and upload revisions accordingly. We are confident about the findings of this paper providing valuable and significant insights to our current practice in language science and engineering, and also DL/NN evaluation and model interpretation. We are committed to improving our formulation to a version that would satisfy our reviewers and readers as much as possible, while remaining true to scientific integrity. We thank you in advance for your evaluation and input and look forward to a fruitful discussion period.

Thank you again and best regards,
Authors of "R&B in Multilingual NLP"

---

### Author Response · Authors · 2020-11-18
**v1.0 uploaded**

Dear Reviewers:

We have uploaded a new version of the paper (v1.0), in which we hope to have addressed your actionable feedback.
We'd appreciate it if you would please take a look and give us your comments. Please let me know if you have any questions or if there is anything that you'd like to discuss.

If you are satisfied with it, please do consider arguing for our acceptance.

Thank you and best regards,
Authors of "R&B in Multilingual NLP"

---

### Author Response · Authors · 2020-11-18
**v1.0.1 uploaded**

Changes:

- fixed minor typos/formulation in main paper.
- §6 (Related Work) re Bender (2009) "language-universal" --> "language independent"
- edited language in App. O (extended version §4) and App. I (language complexity).
(For App. I: "no valid or universal" --> "no universally valid". The latter is a more conservative formulation, also supported more concretely in Haspelmath (2011). The original formulation was meant to be understood with the premise "in the context of computing").

Version number is indicated at the end of the document.

---

### Comment · ~Ada_Wan1 · 2024-04-28
**Note re subsequent publication**

Part of this work, "Representation and Bias in Multilingual NLP: Insights from Controlled Experiments on Conditional Language Modeling", was re-formulated as "Fairness in Representation for Multilingual NLP: Insights from Controlled Experiments on Conditional Language Modeling" and published at ICLR 2022, see https://openreview.net/forum?id=-llS6TiOew.

---

### Decision · Program_Chairs · 2021-01-07
**Final Decision**

**Decision:**

Reject

**Comment:**

The paper study to what extent languages are hard to model by a conditional language-model based on information-theoretic measurements.

Overall, the reviewers value the systematic and extensive controlled experiments present in the paper. However, the presentation of the paper makes it very hard to follow and reviewers all still complain that it is hard to understand the take-home message of the paper.

Despite the reviewers also appreciate the authors' effort in improving the paper, submitting the revision, responding to the feedback, they still conclude that significant reorganizing and revising of the paper is needed before it can be published.

In particular, the paper may be able to improve by backing up the empirical study with some linguistic phenomena or by a more careful rewriting in explaining and discussing the empirical results.

Some other strong arguments such as "Our application of statistical comparisons as a fairness measure also serves as a novel rigorous method for the intrinsic evaluation of languages, resolving a decades-long debate on language complexity." may need to be carefully revised. In this particular example, it is unclear how this paper "resolve" the debate on language complexity by demonstrating a few experiments.  Several sentences like this one should be revised.

---

> ### Author Response · Authors · 2021-01-22
> **Response and announcement**
>
> We appreciate all comments received thus far and will be uploading a revision.
>
> Linguistic complexity (as we conventionally know it) or morphological/morphosyntactic complexity has been decomposed into statistical criteria in length and vocabulary. The argument is sound. In the context of computation, there is **no such complexity necessary** in Transformer CLMs. We will provide a more analytical solution to this in separate work.
>
> (In the case of deeper models, where disparity could arise, one could also understand it as the Transformer being able to solve/learn the ...V+C\*V+C\*V+... pattern (C for consonant and V for vowel), may the script be Latin, Cyrillic, or Abjad. The issue with the ZH logography was remedied using byte representation. Other linguistic complexities are not relevant in the models as demonstrated, unless such complexities are being modeled via word tokenization.)
>
> As noted in the title, the paper is about _insights_ from conditional language modeling experiments.
>
> The findings are clearly enumerated in the summary of findings in §1.1, and summarized again in the conclusion.
>
> The basic take-home message to the question "are languages equally hard to conditional language model" is yes-and-no. The paper walks one through why.
>
> That said, there are some "more profound meta interpretations" possible. So the bigger "take-home messages" can be different depending on where one's "home" is. For example,
>
> - a seasoned domain specialist (a linguist) may find comfort in seeing that neural networks could unite the two disparate schools of thought in linguistics --- general linguistics and comparative linguistics, whose underlying assumptions are, respectively, that languages are fundamentally similar and that languages are fundamentally different.
> - Linguists and/or computational linguists may be inspired to reflect on how our current operation and common conception of language is only word-bound. The field has not looked at language beyond the word-based/grammarian interpretation. This paper offers a new perspective, an opportunity for a new science, one of finer granularity and a more stable fundament, with characters and bytes as units. There is more to the structure of language beyond morphology, syntax, etc..
> - Algorithm-focused practitioners / most computer scientists could see that the role data plays in Double Descent, and how a holistic evaluation and consideration with the nature of data can be beneficial.
> - Someone who is new to NLP or someone who is interested in knowing how CLMing can be made fairer across different languages may be glad to find out data representation/preprocessing matters.
> - To those who see neural networks (more specifically, seq2seq models) as black boxes, we show that they are neither black nor boxes. They might have been quipped black because no one has done these very basic control experiments or looked at the data statistics. They are not boxes but more like _lenses_.
>
> So in this respect, it is ok for different people to not have the same take-home message from this paper.
>
> We hope our writing has offered those who already understood company, and helped those who don’t understand. If any reader should have any questions or comments on how they'd like a particular topic to be addressed more detailedly, or would like to be notified when the next revision becomes available, please email us or leave us a comment below. Thank you.